# Learning from outcomes shapes reliance on moral rules versus cost–benefit reasoning

**Maximilian Maier** [1,4] ✉, **Vanessa Cheung**[1,4] **& Falk Lieder** [2,3]

Many controversies arise from disagreements between moral rules and 'utilitarian' cost–benefit reasoning (CBR). Here we show how moral learning from consequences can produce individual differences in people's reliance on rules versus CBR. In a new paradigm, participants (total $N = 2{,}328$) faced realistic dilemmas between one choice prescribed by a moral rule and one by CBR. The participants observed the consequences of their decision before the next dilemma. Across four experiments, we found adaptive changes in decision-making over 13 choices: participants adjusted their decisions according to which decision strategy (rules or CBR) produced better consequences. Using computational modelling, we showed that many participants learned about decision strategies in general (metacognitive learning) rather than specific actions. Their learning transferred to incentive-compatible donation decisions and moral convictions beyond the experiment. We conclude that metacognitive learning from consequences shapes moral decision-making and that individual differences in morality may be surprisingly malleable to learning from experience.

In the courtroom drama *Terror*, the audience must judge the actions of Major Lars Koch, a fighter pilot accused of killing 164 people. Koch disobeyed orders and shot down a hijacked passenger jet headed towards a stadium filled with 70,000 people. Koch's decision to sacrifice the smaller group to save the larger one was based on utilitarian moral reasoning, but about 36% of the people who saw the play decided that he was guilty[1].

Although such life-and-death decisions are rare in everyday life, people often face analogous moral dilemmas between following moral rules (for example, telling a friend the truth about their bad cooking) and cost–benefit reasoning (CBR; for example, telling a white lie to avoid hurting the friend's feelings). CBR sometimes endorses violating rules for (what is perceived to be) the greater good. People's decisions in these moral dilemmas have consequences that they can learn from. Moral rules and CBR also clash on a number of important, controversial issues, including vaccination mandates and animal testing. These issues often become divisive and highly controversial because people vehemently disagree about whether moral rules take precedence over CBR or vice versa.

The question of whether to rely on moral rules or CBR is often conflated with the normative problem of whether morality consists in choosing actions with good consequences or whether the rightness of an action is inherent in the action itself[2–4]. Consequentialist theories, such as utilitarianism, state that the morality of actions depends on their consequences[5–7]. According to utilitarianism, actions should be judged by their expected combined effects on everyone's well-being. By contrast, deontological theories[8,9] state that actions should be judged only by whether they follow moral rules/norms.

People intuitively rely on both CBR and moral rules[10]. However, both are fallible: unquestioning adherence to moral rules can be harmful in some situations[11], and CBR is fallible when people overlook or misjudge relevant consequences[12]. Thus, ironically, trying to achieve the best possible outcomes through CBR can end up causing worse outcomes than following a moral rule[12–14]. For instance, in *Terror*, Koch's 'utilitarian' action may have prevented the passengers from stopping the terrorists and saving everyone, and it could also have weakened the crucial general norm against killing. If so, the consequences of following the rule would have been better. As such, although Koch's

[1]Department of Experimental Psychology, University College London, London, UK. [2]Department of Psychology, University of California, Los Angeles, Los Angeles, CA, USA. [3]Max Planck Institute for Intelligent Systems, Tübingen, Germany. [4]These authors contributed equally: Maximilian Maier, Vanessa Cheung. ✉e-mail: maximilianmaier0401@gmail.com

decision to commit sacrificial harm was based on CBR, it is unclear whether it met the utilitarian criterion to produce the best consequences. Therefore, in this Article, we delineate reliance on CBR versus moral rules from the endorsement of the ethical theories of deontology and consequentialism. The idea that relying on rules can lead to better consequences and is consistent with consequentialism is well founded in the philosophical literature on moral theories such as rule utilitarianism[15] and global consequentialism[16]. From a psychological perspective, deontological rules can be viewed as heuristics[11,13,14,17]. Some have argued that, even from a utilitarian perspective, using these heuristics in typical real-world situations leads to better consequences than CBR[12–14]. More generally, both reliance on moral rules and reliance on CBR can be considered decision mechanisms or decision strategies[18].

Previous research has conflated these decision mechanisms with the ethical theories of deontology and consequentialism by construing moral dilemmas as decisions between a utilitarian option and a deontological option. To avoid this conflation, we will analyse moral dilemmas as choices between an option that is consistent with a moral rule ('rule option') and an option that is inconsistent with that rule but appears preferable according to CBR ('CBR option'). Though we use these terms, it does not imply that people necessarily explicitly consider CBR or rules during the decision process. Some participants might, for instance, choose the rule option because of moral values or emotional reactions acquired through experiential learning. Moreover, we use CBR to refer to a 'naive' CBR, which considers the number of persons affected by one or more salient outcomes and the corresponding subjective probabilities. We do not assume that people engaging in CBR consider all possible consequences, including indirect and long-term consequences, because this kind of exhaustive cost–benefit analysis would be intractable in real-world situations[13].

What determines how much weight a person puts on moral rules versus CBR in moral dilemmas? One potential mechanism is learning from the consequences of their previous moral decisions. This mechanism is distinct from previous accounts of moral learning[19], including affective learning of moral intuitions[20–22] and moral rules[23], universalization[24], and social learning[25]. Unlike social learning, it involves neither imitation nor observational learning and does not require instruction or social feedback (for example, praise or criticism). Moral learning from consequences is crucial for moral development[21,26–28], yet it is comparatively understudied. This Article makes theoretical and empirical contributions to understanding moral learning from the consequences of previous decisions: we develop a formal theory and computational models of an overlooked mechanism of moral learning, provide an experimental demonstration of its existence and relevance, and introduce an experimental paradigm for studying it.

Our work builds on and extends the reinforcement learning (RL) perspective on moral decision-making developed by Cushman[29] and Crockett[22]. According to this view, people use two systems in moral decision-making: an intuitive model-free system that selects actions on the basis of their average consequences in the past, and a model-based system that builds a model of the world to reason about potential future consequences an action might have in a specific situation. The model-free system has been linked to rule-based decision-making, and the model-based system to CBR.

Both systems are fallible[12–14,20,30,31], but they can complement each other because they fail in different situations[32]. Therefore, people must learn when to use which system. Theories of strategy selection[18,33,34] and meta-control[35–38] postulate an overarching meta-control system that decides which decision mechanism to use in a given situation. On the basis of these theories, we propose that the meta-control system selects which moral decision mechanism to employ in a specific situation. Given the strong empirical evidence for the pervasive influence of RL on decision-making[39,40] and strategy selection[18,33,34,37,41], we postulate that meta-control over moral decision-making is also shaped by RL (metacognitive moral learning).

In the remainder of this Article, we formalize this hypothesis and test it in four experiments. In Experiment 1, we demonstrate the existence of adaptive metacognitive moral learning from the consequences of previous decisions. In Experiments 2 and 3, we examine the mechanisms of this learning; show that it transfers to real-life, incentive-compatible donation decisions; and find that metacognitive learning is a requirement for this transfer. Finally, in Experiment 4, we rule out the possibility that the findings are due to demand characteristics by demonstrating transfer for metacognitive learners to a different experiment, which participants thought was conducted by different researchers.

## Results
### A theory of metacognitive moral learning
Prior work has identified several mechanisms of moral learning[19]. According to one of these mechanisms, RL, moral values are learned from the consequences of previous actions. Each time the consequences of an action are better than expected, the probability of repeating this action is increased, and each time the action's consequences are worse than expected, the probability of repeating this action is reduced. While prior theories of moral learning[22,29] proposed that people learn on the level of more specific behaviours (for example, whether to punch someone), we propose that people also learn on the level of moral decision-making strategies (for example, whether to engage in rules or CBR). We refer to this mechanism as metacognitive moral learning.

According to this theory, the mechanisms of strategy selection learning[18,33,34,37] also operate on the mechanisms of moral decision-making. In strategy selection learning, the consequences of people's actions reinforce the decision strategies that selected them, unlike in operant conditioning[42], where consequences reinforce specific behaviours. We therefore propose that when a person concludes that one of their past decisions was morally wrong (right), this will teach them to decrease (increase) their reliance on the decision system or strategy that chose that action (such as rule-following or CBR; Fig. 1). For example, in *Terror*, the audience learns not only about the morality of shooting down airplanes but also about the morality of CBR more generally. Importantly, if people only learned about specific behaviours, moral learning would not generalize to different types of behaviours. By contrast, metacognitive moral learning should transfer to novel situations involving other behaviours.

Put simply, the mechanisms of metacognitive moral learning differ from standard RL in two key ways: (1) learning occurs in the meta-control system, whose 'actions' are our decision strategies (for example, CBR and rule-following), and (2) the reward signal is the person's moral evaluation of how good or bad their decision was. These moral evaluations are partly based on the consequences of the decision[43,44]. Therefore, learning from the consequences of past decisions could, in principle, adaptively increase people's reliance on decision strategies that produce good consequences and decrease reliance on those that produce bad consequences.

Given the coexistence of model-free and model-based RL[45], we postulate that metacognitive moral learning includes both model-based and model-free RL mechanisms. Model-free metacognitive moral learning consists of learning the expected moral values of relying on different decision strategies. By contrast, model-based metacognitive moral learning consists of learning conditional probability distributions over the possible outcomes of relying on different decision strategies (see 'Computational Models').

### Computational models
To test our theory, we developed RL models of metacognitive moral learning from the consequences of past decisions. As metacognitive learning could be model-based or model-free, we developed one computational model to represent each.

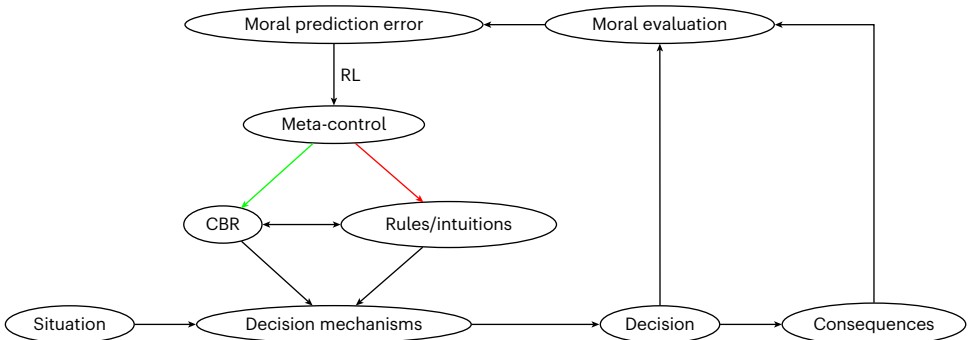

**Fig. 1 | Meta-control of moral decision-making is informed by learning from previous decisions.** The meta-control system determines which of multiple decision mechanisms, including moral rules and CBR, is employed in a particular situation (in this Article, we focus on rules versus CBR, though the model could be extended to accommodate other mechanisms). Depending on one's learning history, the meta-control system may temporarily override moral rules (red arrow) by allocating control over behaviour to CBR (green arrow) or vice versa. Whether moral learning increases (decreases) reliance on CBR or rules in subsequent decisions depends on how positively (negatively) one evaluates the previous decision. People's moral evaluations of past decisions are influenced by the consequences of these decisions. If the evaluation is more (less) positive than expected, this registers as a positive (negative) moral prediction error that causes the reliance on the mechanism that produced the decision to be turned up (down). We suggest that this strategy selection learning shapes moral decision-making.

Model-based learning uses an explicit model of the world to estimate the conditional probabilities of different outcomes[46] (see also ref. 31, p.159). We modelled model-based metacognitive moral learning as Bayesian learning of the conditional probabilities of good versus bad outcomes of decisions made using CBR versus following moral rules (for example, $P$(good outcome | CBR)). This model learns these probabilities by updating the parameters of two beta distributions: one for the probability that CBR will yield a good outcome and one for the probability that following rules will yield a good outcome. The probability of a bad outcome is simply one minus the probability of a good outcome. In other words, this model estimates the likelihoods of four different outcomes: following CBR leads to good versus bad outcomes, and following rules leads to good versus bad outcomes.

Model-free learning assigns values to actions directly rather than modelling the probabilities of different outcomes. Those values are based on the average reward each action produced in the past. To model model-free metacognitive moral learning, we adapted the most common model of model-free RL: $Q$-learning[47–49]. Our model assigns values directly to using moral decision-making strategies (that is, CBR versus following moral rules); those values are represented as $Q$ values. After each decision, the model updates the $Q$ value of relying on the decision strategy that produced that decision. This update is proportional to the experienced moral prediction error. The moral prediction error is the difference between the decision maker's moral evaluation of how morally right or wrong the decision was and the current $Q$ value of the decision strategy that produced it. The higher the $Q$ value assigned to a decision strategy, the more likely the model is to rely on it.

Unlike the model-based beta-Bernoulli model, the model-free $Q$-learning model does not learn about the probabilities of the four different outcomes; instead, it learns two $Q$ values: one for CBR and one for rule following. Our two computational models therefore capture the key distinction between model-based and model-free learning: model-based learning involves learning about the probabilities of the different outcomes of an action, whereas model-free learning assigns a value to the action itself. The models are described in more detail in the Methods 'Computational models of moral learning from consequences' section.

Our models of metacognitive moral learning attribute the outcome of each action to the decision strategy that selected it (that is, applying CBR versus moral rules). We compared these models to models of behavioural moral learning. Unlike metacognitive learning, behavioural learning attributes the outcome of each action to the action itself. For example, a child pushing their friend out of the sandbox may see that this action causes their friend to become upset, and learn not to repeat such actions. To model the generalization of behavioural learning across the different dilemmas of our experimental paradigm, we make the simplifying assumption that people generalize from the outcome of (not) taking the action under consideration in any one dilemma to the value of (not) taking the action under consideration in all other dilemmas. Our models of behavioural learning thus assume that each decision is represented as either performing the behaviour under consideration (action) or not (omission). Actions were a very salient behaviour-level representation on which the learning signal could operate, given that in each trial, participants were asked whether to act (for example, push the man) or not.

Behavioural learning can be either model-based or model-free. Apart from changing the learning signal to operate on the level of behaviours rather than strategies, our models of model-based versus model-free behavioural learning are therefore equivalent to our models of model-based versus model-free metacognitive learning. We deconfounded the action/omission learning from metacognitive learning as sometimes the action coincided with CBR and sometimes with rules.

**A new paradigm using realistic moral dilemmas with outcomes**

To test our theory and models of metacognitive moral learning, we developed an experimental paradigm for measuring the effect of learning from the consequences of previous moral decisions on subsequent moral decisions. Unlike previous moral decision-making paradigms, ours is a learning paradigm. Participants make decisions in a series of different moral dilemmas, where they see the outcomes of each decision before moving on to the next. In each trial, the participant reads a realistic moral dilemma and decides between two actions: one favoured by CBR (the 'CBR option') and one favoured by a moral rule (the 'rule option').

At the beginning of the paradigm, participants are randomly assigned to one of two conditions. In the 'CBR Success' condition, the CBR option always leads to overall good outcomes, and the rule option to overall bad outcomes. In the 'Rule Success' condition, the rule option always leads to overall good outcomes, and the CBR option to overall bad outcomes. We illustrate this paradigm in Fig. 2, and more details can be found in the Methods.

The moral dilemmas most widely used in experiments, which are based on the "trolley problem"[50,51], have been criticized as unrealistic and bizarre[12,52]. Furthermore, they assume that the outcomes are known with certainty and often confound CBR with taking action and rule-following with inaction (that is, omission). The moral dilemmas

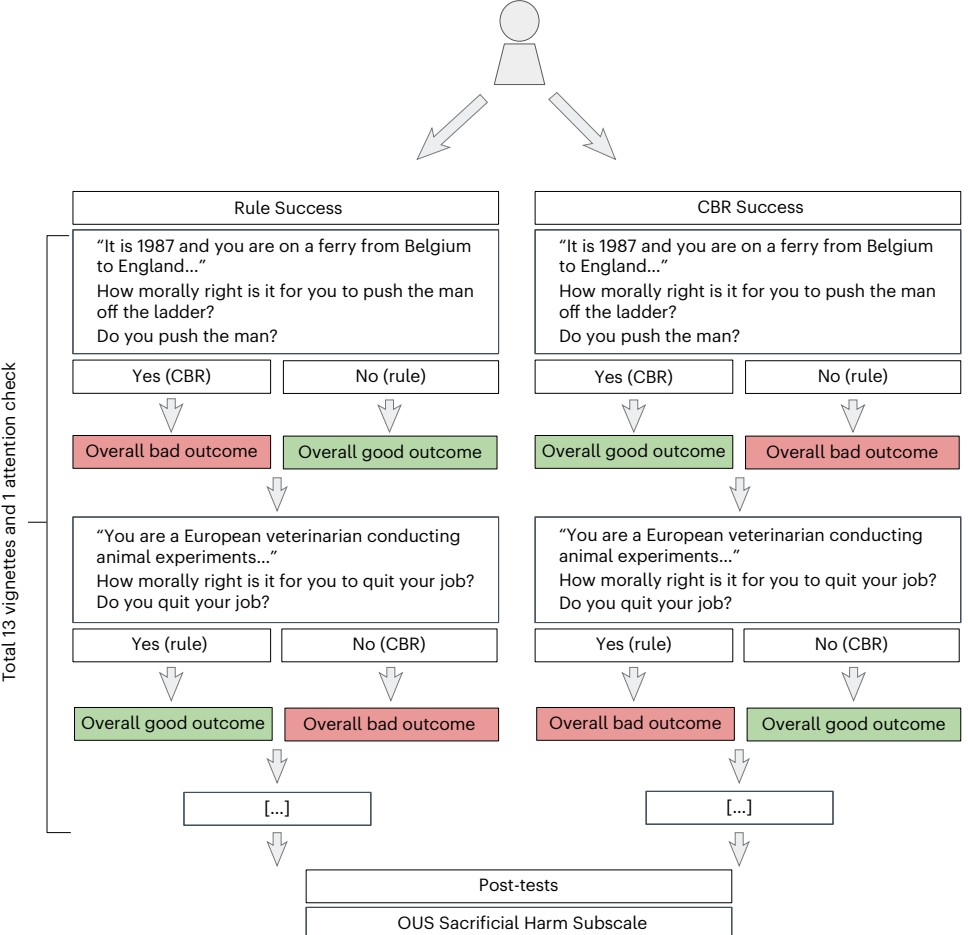

**Fig. 2 | The moral learning paradigm in Experiment 1.** Participants are randomized into one of two conditions: Rule Success and CBR Success. The experimental condition determines whether choosing the CBR option or the rule option leads to good or bad outcomes. In some vignettes, the action under consideration is the CBR option (for example, 'Do you push the man?'), and in other vignettes, it is the rule option (for example, 'Do you quit your job?'). This means that the 'yes' and 'no' responses do not always correspond to the same decision strategy (rules versus CBR) even in the same experimental condition. Because of this, from the participant's perspective, choosing 'yes' would sometimes lead to good outcomes and sometimes lead to bad outcomes. More details about the paradigm can be found in the Methods.

used in our paradigm mitigate all of these limitations (Methods, 'The moral learning paradigm').

Most participants are not trained in moral philosophy, meaning that the abstract moral theories of deontology and utilitarianism are probably less salient for them than the concrete choices between action and omission, the specific behaviours, and the specific moral rules that recommend or oppose them. Our experimental paradigm varied all of these salient features independently of which option each strategy recommended (see Fig. 2 and the Methods for more details). It is therefore not immediately obvious to participants what is being reinforced in our paradigm. This was also reflected in their responses to an open question in Experiment 1. (At the end of the study, we asked the participants whether they "used information about the outcomes of [their] choices when making decisions throughout the experiment" and, if so, how. Of those participants who reported taking outcomes into account, most appeared to be unaware of the specific manipulation—for example, "Yes I tried to worry more about the initial moral decision and less on the outcomes as it was clear the outcome could vary/was more unpredictable" and "I tried to anticipate what the likely outcome would be, but I wasn't right"; the full responses are available in the online repository.)

Furthermore, a majority of participants engaged strongly with the task and considered it informative about the real world: 90% of participants reported that they imagined the scenarios very vividly, and 90% reported that they felt good or bad after they saw good or bad outcomes. In addition, 67% of participants indicated that the decisions, situations and outcomes they encountered in the task were informative about the real world, and 50% indicated that the task gave them the opportunity to learn how to make better decisions in the real world. Finally, most participants indicated that the outcomes were plausible (83%) and a good reflection of whether they made the right decision (61%; see Supplementary Results, Experiment 4 for more details).

## Experiment 1

Experiment 1 investigated whether and, if so, what people learn from the outcomes of their previous moral decisions. We preregistered (https://osf.io/jtwvs) the following predictions: (1) when choosing the CBR option leads to good outcomes, participants learn to rely more on CBR; (2) when choosing the rule option leads to good outcomes, participants learn to rely more on moral rules; and (3) this learning transfers to people's general attitudes towards utilitarianism. Throughout this Article, we use one-sided tests only when we preregistered a one-sided prediction. We use two-sided tests either when we did not preregister a direction (mostly for interaction tests) or for tests that were not preregistered.

**Outcomes of past decisions influence choices and judgements.** *Choices.* Figure 3a shows that, depending on the experimental condition, participants learned to either increase or decrease their reliance

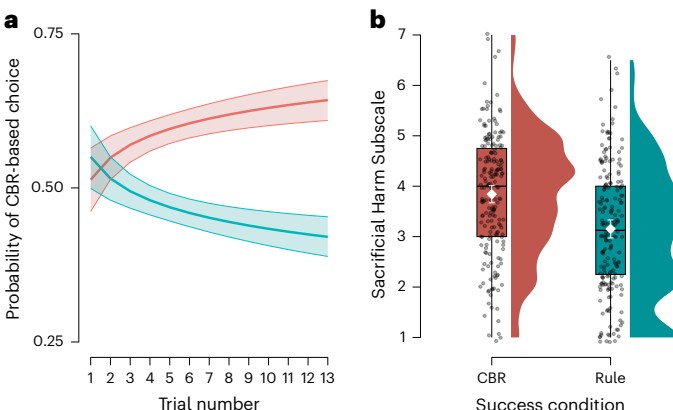

**Fig. 3 | Learning from consequences shapes reliance on moral rules versus CBR. a**, Probability of choosing the CBR option. The confidence bands indicate the 95% confidence level. **b**, Distribution of the OUS Sacrificial Harm Subscale scores in the CBR Success and Rule Success conditions. Means and error bars are indicated in white ($N = 387$). The error bars indicate 95% CIs. The box plots indicate the median with the interquartile range (IQR), and the whiskers extend to 1.5 times the IQR.

on CBR. In the CBR Success condition, the proportion of participants choosing the CBR option increased from 51.8% (95% confidence interval (CI), [44.5%, 59.0%]) on the first trial to 68.2% (95% CI, [61.2%, 74.7%]) on the last trial. In the Rule Success condition, the proportion of CBR choices decreased from 55.7% (95% CI, [48.4%, 62.9%]) to 44.3% (95% CI, [37.1%, 51.6%]).

As predicted, the logistic mixed-effects regression showed that participants in the CBR Success condition became increasingly more inclined to choose the CBR option with each decision ($b_{\log(\text{trial } N)} = 0.208$, $z = 3.66$, $P < 0.001$, one-sided). (Note that we report estimates for the log trial number for consistency with the remaining results. Following the preregistered model selection procedure, we obtained the same result using the model without the log transformation ($b_{\text{trial } N} = 0.044$, $z = 3.87$, $P < 0.001$).) Conversely, those in the Rule Success condition became increasingly more inclined to choose the rule option ($b_{\log(\text{trial } N)} = -0.203$, $z = -3.63$, $P < 0.001$, one-sided).

*Moral judgements*. Participants judged the moral rightness of the action under consideration on a scale from 0 ("Not at all morally right") to 100 ("Completely morally right") before making a decision. In some vignettes, this action is consistent with the CBR option, and in others, it is consistent with the rule option. We predicted that within each condition, additional experience would increase (decrease) the perceived moral rightness of actions that are consistent (inconsistent) with the rewarded decision strategy (that is, following CBR versus rules).

In the CBR Success condition, perceived moral rightness increased for actions endorsed by CBR ($b_{\log(\text{trial } N)} = 2.40$, 95% CI [0.56, 4.24]) and decreased for actions opposed by CBR (that is, actions endorsed by moral rules) ($b_{\log(\text{trial } N)} = -0.96$; 95% CI, [-3.33, 1.42]). In line with this, we found a significant interaction between trial number and whether the action coincided with rules or CBR on moral judgements ($b = -1.68$, $t(2340.73) = -2.20$, $p = 0.028$, two-sided). In other words, participants in the CBR Success condition became more likely to endorse CBR actions and to oppose rule actions during the experiment, which is in line with increased reliance on CBR as they proceeded through the task.

Similarly, in the Rule Success condition, perceived moral rightness increased for actions complying with moral rules ($b_{\log(\text{trial } N)} = 1.68$; 95% CI, [-0.74, 4.09]) and decreased for actions violating rules (that is, actions endorsed by CBR) ($b_{\log(\text{trial } N)} = -1.23$; 95% CI, [-3.01, 0.56]).

However, the interaction between trial number and whether the action coincided with rules or CBR was not statistically significant ($b = -1.45$, $t(2239.64) = -1.91$, $P = 0.057$, two-sided).

**Moral learning transfers to convictions about sacrificial harm.** To test for transfer beyond our experimental paradigm, we included the Oxford Utilitarianism Scale (OUS) Sacrificial Harm Subscale[53] as a post-test. Figure 3b shows that, as predicted, the mean utilitarianism scores were significantly higher in the CBR Success condition than in the Rule Success condition ($t(383.31) = 5.51$, $P < 0.001$, $d = 0.56$, one-sided).

In this and the following experiments, we included some exploratory (that is, not preregistered) measures after the learning paradigm (see Supplementary Results, Experiment 1 for the results).

## Experiment 2

In Experiment 2, we aimed to (1) replicate the results of Experiment 1, (2) show transfer to incentive-compatible donation decisions and additional self-report measures, and (3) understand the underlying learning mechanisms.

To achieve the second objective, we added two new measures of transfer: an incentive-compatible donation decision and a scale to measure deontological convictions. To achieve the third objective, we added new self-report measures designed to measure the mechanisms of decision-making and learning. The experiment was preregistered at https://osf.io/7ds8a.

**Replication of effect on choices and judgements.** Experiment 2 replicated the effect of learning from consequences on moral decision-making found in Experiment 1 (CBR Success: $b_{\log(\text{trial } N)} = 0.15$, s.e. = 0.06, $z = 2.61$, $P = 0.005$, one-sided; Rule Success: $b_{\log(\text{trial } N)} = -0.12$, s.e. = 0.06, $z = -2.16$, $P = 0.015$, one-sided). As in Experiment 1, the effect of learning was weaker for judgements than for choices; however, in Experiment 2, we found no significant interaction effect of trial number and experimental condition on judgements (CBR Success: $b = 0.35$, $t(909.10) = 0.45$, $P = 0.656$, two-sided; Rule Success: $b = -1.11$, $t(2224.32) = 1.39$, $P = 0.164$, two-sided).

**Self-report measures show model-based metacognitive learning.** We developed self-report measures for model-free and model-based learning in line with previous literature on these two types of RL in the moral domain[22,29,54]. Because model-based learning involves learning a probabilistic model of the anticipated outcomes of actions, we used a measure that asked the participants to rate the probabilities of good versus bad outcomes of choosing the CBR option and of choosing the rule option.

In contrast, model-free learning involves assigning values intrinsically to actions rather than building a model of their possible consequences. Therefore, to measure model-free learning, we adapted a task from Cushman et al.[54], where we asked the participants to imagine carrying out typically harmful actions, which would not cause negative consequences in this specific instance (for example, shooting a prop gun). If people show an aversion to these actions even though they cannot produce negative consequences, this suggests that they are assigning values intrinsically to actions (that is, model-free learning).

*Evidence for model-based metacognitive learning*. We showed the participants two new moral dilemmas (Methods). In one dilemma, the action under consideration was the rule option ('rule action'), and in the other dilemma, the action was the CBR option ('CBR action'). For both dilemmas, the participants predicted the probability of an overall good versus overall bad outcome for each action and each omission on a scale of 0 ("Bad outcome much more likely") to 100 ("Good outcome much more likely"). To assess the effect of learning, we calculated the probability of an action versus an omission leading

**Table 1 | Cognitive modelling results showing the proportions of participants that relied on each type of learning in Experiments 2–4**

| Model | | | $\mathbb{E}(f|Y)$ (%) | | | | | |
|---|---|---|---|---|---|---|---|---|
| | | | Experiment 2 | | Experiment 3 | | Experiment 4 | |
| | | | CBR | Rule | CBR | Rule | CBR | Rule |
| Model-based | Metacognitive | MB-M | **78.88** | 18.96 | **86.97** | 31.81 | **74.08** | 40.24 |
| Model-free | | MF-M | 7.78 | 10.34 | 3.32 | 4.06 | 8.26 | 1.26 |
| Constant | | C-M | 4.42 | 2.43 | 2.53 | 3.45 | 3.59 | 5.55 |
| Model-based | Behaviour | MB-B | 3.97 | **61.73** | 4.41 | **56.17** | 9.32 | **49.53** |
| Model-free | | MF-B | 2.52 | 2.90 | 2.18 | 3.82 | 2.90 | 2.59 |
| Constant | | C-B | 2.44 | 3.64 | 0.59 | 0.69 | 1.85 | 0.82 |

'CBR' denotes the CBR Success condition, and 'Rule' denotes the Rule Success condition. $\mathbb{E}(f|Y)$, expected frequency. The largest proportions in each condition are highlighted in bold.

**Table 2 | Cognitive modelling results showing the proportions of participants that relied on each learning mechanism in Experiments 2–4**

| Learning mechanism | Experiment 2 | | | | Experiment 3 | | | | Experiment 4 | | | |
|---|---|---|---|---|---|---|---|---|---|---|---|---|
| | CBR | | Rule | | CBR | | Rule | | CBR | | Rule | |
| | $\mathbb{E}(f|Y)$ (%) | $\varphi$ | $\mathbb{E}(f|Y)$ (%) | $\varphi$ | $\mathbb{E}(f|Y)$ (%) | $\varphi$ | $\mathbb{E}(f|Y)$ (%) | $\varphi$ | $\mathbb{E}(f|Y)$ (%) | $\varphi$ | $\mathbb{E}(f|Y)$ (%) | $\varphi$ |
| Metacognitive learning | **89.4** | **1** | 27.9 | 0.05 | **92.5** | **1** | 35.8 | 0.06 | **85.5** | **1** | 41.9 | 0.26 |
| Behavioural learning | 5.2 | 0 | **67.3** | **0.95** | 5 | 0 | **60.8** | **0.94** | 11.1 | 0 | **53.5** | **0.74** |
| No learning | 5.4 | 0 | 4.8 | 0 | 2.5 | 0 | 3.4 | 0 | 3.3 | 0 | 4.6 | 0 |

'CBR' denotes the CBR Success condition, and 'Rule' denotes the Rule Success condition. The exceedance probability $\varphi$ of a given model family is the probability that the proportion of participants best explained by a model from that family is greater than for any of the alternative model families. The largest proportions in each condition are highlighted in bold.

to good consequences for all participants in both the rule action vignette and the CBR action vignette (that is, $\Delta M$, or the $P(+|action) - P(+|omission)$ score).

Participants in the Rule Success condition rated rule actions to be more likely to lead to good outcomes relative to omissions ($\Delta M = 6.03$; 95% CI, [0.41, 11.65]) than participants in the CBR Success condition ($\Delta M = -3.41$; 95% CI, [−8.81, 1.99]). In contrast, participants in the CBR Success condition rated CBR actions to be more likely to lead to good outcomes relative to omissions ($\Delta M = 8.21$; 95% CI, [2.76, 13.67]) than participants in the Rule Success condition ($\Delta M = -5.47$; 95% CI, [−10.94, −0.01]). This interaction was significant ($F_{1,756} = 17.28, P < 0.001$) (Supplementary Fig. 1).

In other words, participants in the CBR Success condition were more positive about the expected outcomes of CBR actions (that is, they thought engaging in them would lead to better consequences relative to not doing anything) than about those of rule actions, while this pattern was reversed in the Rule Success condition. This suggests that (some) participants learned the conditional probabilities of good versus bad outcomes, given that the decision is reached using CBR or rules, a mechanism we refer to as model-based metacognitive learning.

*No evidence for model-free metacognitive learning.* The experimental manipulation had no significant effect on people's emotional reactions to violations of the moral rule to do no harm ($t(377.08) = 0.62$, $P = 0.269, d = 0.06$, one-sided), which would have been evidence for model-free learning[54] (Supplementary Results, Experiment 2 'Model-free learning').

For more details on the methodology, analytic approach and results for additional self-report measures, see Methods and Supplementary Results, Experiment 2.

**Computational modelling results support (model-based) metacognitive learning (exploratory).** We used the data from Experiment 2 to test our computational models of metacognitive moral learning against computational models of behavioural RL, which learned whether to perform the behaviour under consideration (action) or not (omission), and the equivalent models without any learning (Methods, 'Computational models of moral learning from consequences').

We found that in the CBR Success condition, most participants (78.9%) relied primarily on model-based metacognitive learning. In the Rule Success condition, most participants (61.7%) relied primarily on model-based behavioural learning, and only 19.0% engaged in model-based metacognitive moral learning (Table 1). When comparing families of models, in the CBR Success condition, the proportion of participants whose behaviour was best explained by either of the models of metacognitive learning (89.4%) was significantly larger than the proportions of participants best explained by models of behavioural learning or no learning (Table 2). By contrast, in the Rule Success condition, the two models of behavioural learning jointly provided the best explanation for the majority of participants (67.3%), while the two models of metacognitive learning provided the best explanation for only 27.9% of the participants (Table 2).

**Metacognitive learning transfers to a range of measures.** Learning in our experimental paradigm transferred to self-report measures of people's moral convictions and an incentive-compatible donation decision (Fig. 4). As predicted, compared with the Rule Success condition, participants in the CBR Success condition scored higher on the OUS Sacrificial Harm Subscale[53] ($t_{369.12} = 4.02, P < 0.001$, $d = 0.42$, one-sided) and lower on the Deontology Subscale of the Deontological-Consequentialist Scale (DCS)[55] ($t_{376.31} = 1.67, P = 0.048$, $d = 0.17$, one-sided).

In the incentive-compatible donation decision, participants allocated £200 between a charity promoting human challenge trials, in which healthy volunteers are infected with a virus to speed up the development of vaccines (CBR option), and a charity supporting conventional medical research (rule option; Section 3 shows that participants

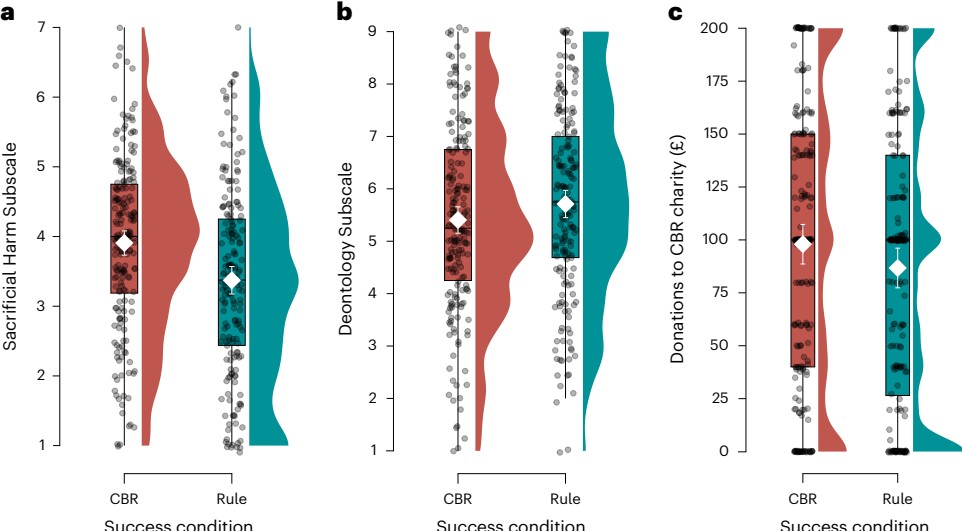

**Fig. 4 | Transfer results for Experiment 2. a**, Responses to the Sacrificial Harm Subscale from the OUS. **b**, Responses to the Deontology Subscale from the DCS. **c**, The amount of money participants donated to the charity supporting human challenge trials. Each panel compares responses in the CBR Success and Rule Success conditions. Means and 95% CIs are indicated in white. The box plots indicate the median with the IQR, and the whiskers extend to 1.5 times the IQR ($N$ = 380).

generally agreed with this categorization). According to a preregistered one-sided $t$-test, participants in the CBR Success condition donated significantly more money (mean, £97.70) to support human challenge trials than participants in the Rule Success condition (mean, £86.73) ($t_{377.54} = 1.67$, $P = 0.047$, $d = 0.17$, one-sided).

According to the theory outlined earlier, the transfer we observed should occur only for people who engage in metacognitive learning—that is, learning about decision mechanisms (CBR versus rules). In line with this, we found that the predicted transfer effects occurred for only those participants who showed evidence of metacognitive learning (Fig. 5). Evidence for metacognitive learning significantly moderated the effect of the experimental manipulation on the OUS Sacrificial Harm Subscale ($b = 0.74$, $t_{376} = 5.01$, $P < 0.001$) and the DCS Deontology Subscale ($b = -0.74$, $t_{376} = 3.52$, $P < 0.001$), but this moderation was not significant for the donation decision ($b = 14.36$, $t_{376} = 1.86$, $P = 0.064$) (all two-sided). Moreover, when including evidence for metacognitive learning as a covariate, we found a significant main effect of condition on the OUS Sacrificial Harm Subscale ($b = 2.15$, $t_{376} = 6.17$, $P < 0.001$), on the DCS Deontology Subscale ($b = -1.90$, $t_{376} = 3.87$, $P < 0.001$) and on donation decisions ($b = 42.23$, $t_{376} = 2.32$, $P = 0.010$) (all two-sided).

### Experiment 3
Exploratory analyses of Experiment 2 suggested that moral learning might transfer only for people engaging in metacognitive learning. Experiment 3 provides a well-powered, preregistered (https://osf.io/7guj6) replication and extension of these findings with two additional real-world donation decisions (Methods) and twice as many participants ($N$ = 834). All materials were identical to those from Experiment 2, except that we removed the self-report measures of metacognitive learning.

We found that the proportions of participants best explained by each model (Table 1) and each type of learning mechanism (Table 2) were similar to those in Experiment 2. As predicted, metacognitive learners exhibited strong evidence of transfer to self-report measures of moral convictions (OUS Sacrificial Harm Subscale: $b = 2.48$, $t_{830} = 10.62$, $P < 0.001$; DCS Deontology Scale: $b = -2.07$, $t_{830} = 6.36$, $P < 0.001$) and an overall main effect of the assigned condition in the moral learning paradigm on the three donation decisions ($b = 22.07$, $t_{830} = 5.34$, $P < 0.001$).

When averaging across all participants, we found evidence for transfer to people's moral convictions (OUS Sacrificial Harm Subscale: $t_{829.06} = 7.50$, $P < 0.001$, $d = 0.52$, one-sided; DCS Deontology Subscale: $t_{825.3} = 2.81$, $P = 0.003$, $d = 0.20$, one-sided) but were unable to detect the effect on donations ($t_{832} = 1.55$, $P = 0.061$, one-sided). This discrepancy underscores the importance of metacognitive learning for transfer. We report the full results, including effects on individual donation decisions, in Supplementary Results, Experiment 3.

### Experiment 4
Experiments 1–3 showed that the moral learning observed in our paradigm transfers to other measures within the context of the same experiment. In principle, this could be due to demand characteristics or very narrow, highly context-specific learning. Experiment 4 therefore aimed to demonstrate that the effects of moral learning from consequences transfer beyond the experiment in which the learning took place (that is, transfer to another study conducted by different experimenters). To achieve this, we used an innovative experimental design comprising two separate online studies run by different experimenters from different institutions. The first online study contained the learning paradigm, and the subsequent (seemingly unrelated) study measured people's moral convictions and donation behaviour. This allowed us to show that the learning transfers to a new experimental context, thus ruling out the alternative explanation that effects are driven by demand characteristics.

We preregistered the experiment at https://osf.io/dgsfb. Because our theoretical framework predicts transfer only for metacognitive learners and focusing on metacognitive learners had higher statistical power in previous experiments, we preregistered to test transfer only for metacognitive learners.

**Computational modelling results show (model-based) metacognitive learning.** Replicating the previous modelling results, we again found that similar proportions of participants were best explained by each model (Table 1) and each type of learning mechanism (Table 2). For additional model-based analyses, see Supplementary Results, Experiment 4.

**Metacognitive learning transfers to a different experiment.** As shown in Fig. 6, Experiment 4 replicated all transfer effects from Experiments 1–3. The effect was replicated across experiments and an average delay

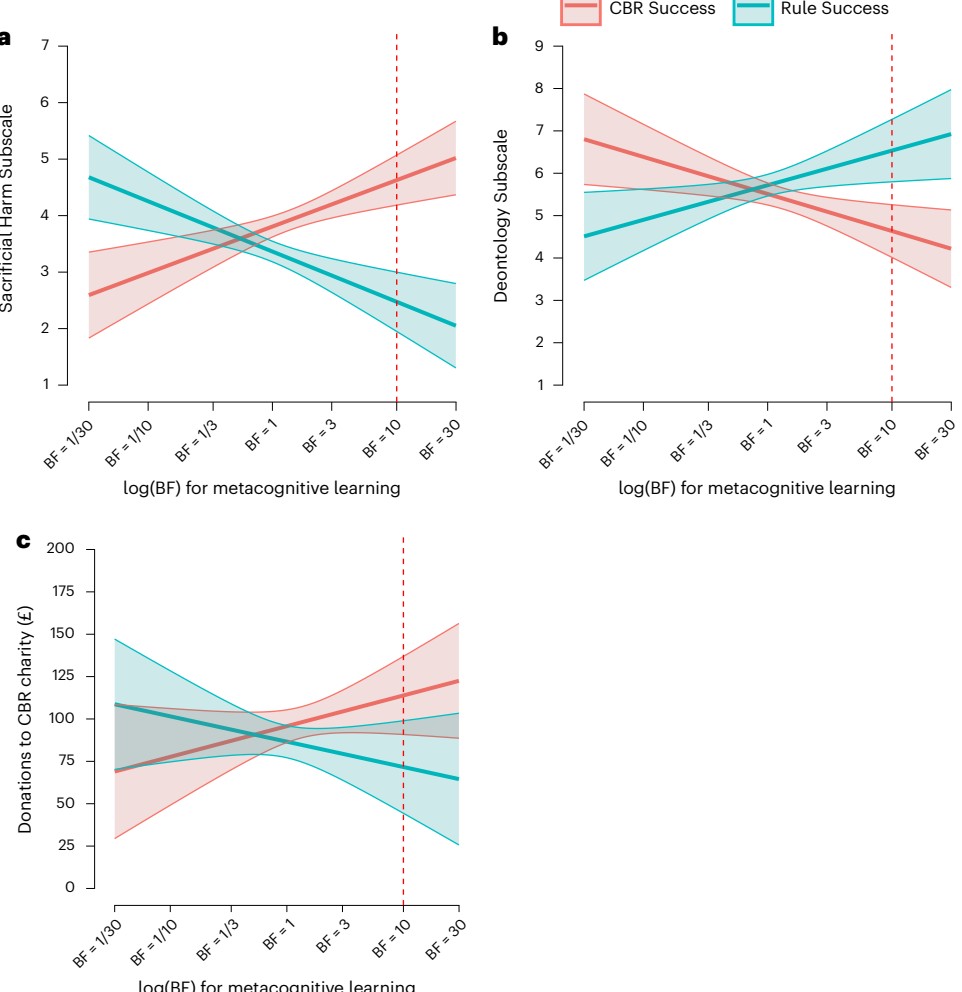

**Fig. 5 | Transfer is moderated by evidence for metacognitive learning in Experiment 2. a**, Responses to the Sacrificial Harm Subscale from the OUS. **b**, Responses to the Deontology Subscale from the DCS. **c**, How much money participants donated to the charity supporting human challenge trials. Each panel compares responses between the CBR Success and Rule Success conditions as a function of the amount of evidence the participants' responses in the moral learning paradigm provided for metacognitive learning quantified using Bayes factors (BF). BF values of >1 indicate evidence for metacognitive learning. The red dotted lines at BF = 10 indicate strong evidence of metacognitive learning; this is where the main effect of the experimental condition was tested. For all panels, the confidence bands indicate the 95% confidence level ($N = 380$). See Section 3 for a smoothed conditional means version of the plots.

of about two hours (mean, 121 minutes; median, 99 minutes; range, 0.25 to 561 minutes).

We found that evidence for metacognitive learning significantly moderated the effect of the experimental manipulation on all measures of transfer (OUS Sacrificial Harm Subscale: $b = 0.81$, $t_{723} = 7.79$, $P < 0.001$; DCS Deontology Subscale: $b = -0.62$, $t(723) = 4.82$, $P < 0.001$; donations: $b = 8.69$, $t(723) = 4.06$, $P < 0.001$). (These results are for the 'Human Challenge Trials' and 'Animal Testing' vignettes. As preregistered, we removed the vignette about sending doctors to crisis zones because a pilot study found that participants did not view this as a rules-versus-CBR conflict; see Methods, 'Transfer to new study'.) Furthermore, when including evidence for metacognitive learning as a covariate, we found a significant main effect of condition on all measures of transfer (OUS Sacrificial Harm Subscale: $b = 2.16$, $t(723) = 8.77$, $P < 0.001$; DCS Deontology Subscale: $b = -1.64$, $t(723) = 5.33$, $P < 0.001$; donations: $b = 20.54$, $t(723) = 4.04$, $P < 0.001$).

This evidence of transfer to a new experiment, which to participants appeared to be conducted by different researchers, rules out demand characteristics as an alternative explanation of the findings from Experiments 1–3. Moreover, as detailed in Supplementary Results, Experiment 4, Risk Aversion, Experiment 4 also ruled out the alternative explanation that the learning and transfer effects are due to changes in risk aversion.

**Individual differences in perceived real-world relevance and engagement predict metacognitive learning (exploratory).** To understand why some participants engaged in metacognitive learning whereas others did not, we measured how they perceived the moral learning paradigm. In brief, we found that evidence for metacognitive learning was predicted by participants taking the task seriously; believing that the task allowed them to learn how to make better decisions in the real world; experiencing an emotional response to the outcomes; and perceiving the outcomes as plausible, informative about the real world and a good reflection of whether they made the right decision (all $P < 0.02$). We found no evidence that any of these factors explained why metacognitive learning was more prevalent in the CBR Success condition than in the Rule Success condition (all $P > 0.10$). For details and additional results, see Supplementary Results, Experiment 4, Other Exploratory Measures.

## Discussion

Across four experiments, learning from the consequences of past decisions consistently guided participants to adopt moral decision

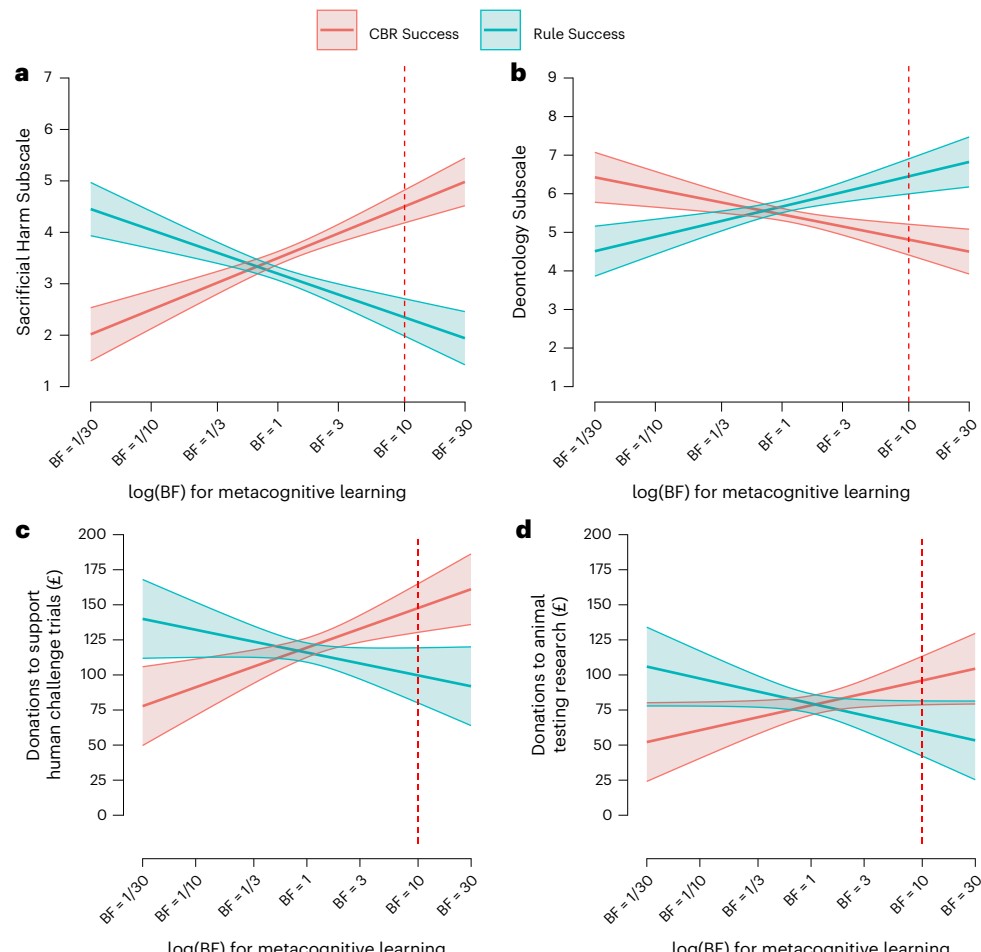

**Fig. 6 | Metacognitive learning transfers to measures of moral convictions and donation decisions in Experiment 4. a**, Responses to the Sacrificial Harm Subscale from the OUS. **b**, Responses to the Deontology Subscale from the DCS. **c**, The amount of money donated to the charity advocating for human challenge trials. **d**, The amount of money donated to a breast cancer research charity that uses animal research. Each plot compares responses between the CBR Success and Rule Success conditions as a function of the amount of evidence the participants' responses in the moral learning paradigm provided for metacognitive learning (BF). BF values >1 indicate evidence for metacognitive learning. The red dashed lines at BF = 10 indicate strong evidence of metacognitive learning; this is where the main effect of the experimental condition was tested. For all panels, the confidence bands indicate the 95% confidence level (*N* = 727). See Section 3 for a smoothed conditional means version of the plots.

strategies that benefited the greater good. In an environment where relying on CBR led to better outcomes, participants learned to override moral rules for what they perceived to be the greater good. In an environment where CBR led to worse outcomes, participants learned to follow moral rules instead. These findings suggest that meta-control over moral decision-making is shaped by fast, adaptive learning from the consequences of previous decisions.

Moreover, we found that metacognitive moral learning involves generalization: its effects transferred from decision-making in hypothetical moral dilemmas to scales developed to measure utilitarianism versus deontology, which are usually considered stable personality traits[53], and incentive-compatible, real-world donation decisions. These transfer effects occurred even when the transfer measures were administered in a different experiment, which participants thought was conducted by different experimenters. This rules out the possibility that effects were driven by demand characteristics. Finally, we observed transfer only for participants who showed evidence for metacognitive learning. These learning and transfer effects are consistent with the theorized mechanism illustrated in Fig. 1: people increase or decrease their reliance on following moral rules versus CBR according to the outcomes of previous decisions.

Even though moral learning was driven by consequences, it did not always direct people towards making their decisions by reasoning about consequences (CBR). Instead, learning from consequences increased reliance on moral rules when following them had previously led to better consequences. In other words, people who prioritize moral rules might do so to bring about good consequences, even though they may not necessarily be explicitly reasoning about this. Overall, our findings suggest that moral learning from consequences aligns people's decision-making with global consequentialism[16], according to which one should use whatever rule-based, reasoning-based or virtue-based decision mechanism yields the best consequences. This suggests that, ironically, some people who insist on following moral rules regardless of the consequences may have reached this conviction by learning from consequences. In this sense, everyone might be a consequentialist learner, regardless of which moral principles they endorse.

These findings offer a new perspective on human morality that connects two fundamental debates: the debate about whether people make moral decisions on the basis of (intuitive) moral rules (often equated with deontology) or CBR (often equated with utilitarianism)[11,56] and the debate about whether human morality is learned from experience (empiricism) or innate (nativism)[57]. Our findings suggest that the degree to which a person's moral decisions are driven by either utilitarian reasoning or intuitive moral rules depends partly on their learning history. Across all experiments, the overwhelming

majority of participants showed some form of learning from consequences (at least 95% showed metacognitive or behavioural learning; Table 2), suggesting that at most 5% of people are strictly deontological, in the sense that they would continue to base their decisions on moral rules even if the consequences of previously doing so had been predominantly bad. Instead, many people even show metacognitive learning: they flexibly adapt the degree to which they rely on (intuitive) moral rules by learning from the consequences of previous decisions. This suggests that people's moral decisions are not inevitably controlled by potentially innate intuitions. Instead, we found that experience can teach people to update their decision-making strategies on the basis of empirical observations. This supports the empiricist view that human morality is shaped by learning from experience. Consistent with this view, people might disagree about moral dilemmas partly because of differences in life experience. Some people may have experienced that blindly following the rules yields worse outcomes than occasionally overriding them with CBR. Many others have probably experienced that their attempts to outsmart the rules usually backfire. To the extent that moral disagreements are caused by learning from different experiences, we might be able to overcome our moral disagreements by sharing our experiences and learning from the experiences of others.

Although all of our experiments concerned moral decision-making, our finding that adaptive metacognitive learning from the consequences of past decisions shapes reliance on different decision strategies might also apply to judgement and decision-making more generally. Converging evidence for adaptive metacognitive learning in domains such as financial decision-making[18,33,34], cognitive control[37], planning[58–61], and problem-solving and mental arithmetic[18] seems to support this generalization.

Our results also challenge previous approaches to moral psychology that equated normative theories of morality with decision strategies. As we argued in the introduction of this Article, conceptually, deontology and utilitarianism are different from reliance on rules versus CBR, even though they are sometimes equated in the literature. The former are ethical theories that tell us what we should value, whereas the latter are decision strategies that can be used to achieve outcomes that are consistent with those values. In line with research showing that simpler heuristics can lead to better consequences in certain environments, in the real world, relying on rules may sometimes lead to better consequences than CBR for various reasons (for example, increased accuracy due to the bias–variance trade-off[62,63], lower cost of computation[18] and increasing trust[13]). Below, we discuss two findings showing that measures previously considered to measure reliance on different ethical theories may in fact measure reliance on different decision strategies.

First, existing self-report measures of deontology versus utilitarianism may actually measure reliance on the specific decision strategies of following rules versus CBR. Our results support this conclusion because (1) learning from consequences can increase people's scores on the Deontology Subscale of the DCS[55] and decrease their scores on the OUS Sacrificial Harm Subscale[53] and (2) interpersonal differences in these scales were unrelated to evidence of metacognitive moral learning from the consequences of past decisions. If the DCS Deontology Subscale actually measured deontology, we would expect that participants who score higher would be less driven by outcomes and more by the intrinsic rightness of the action and therefore show less learning. Instead, we found that participants' scores on this scale were unrelated to how much they engaged in metacognitive moral learning from consequences. Moreover, learning from consequences changed participants' scores on the DCS Deontology Subscale. This suggests that it measures reliance on rules rather than on deontology, and that reliance on rules is more receptive to learning than previously thought, given that these scales are often considered to measure stable traits[53]. A similar argument may apply to the OUS Sacrificial Harm Subscale,

although the case here is somewhat weaker because this scale claims to measure only one specific component of utilitarian psychology (sacrificial harm).

Second, a participant's 'deontological' or 'utilitarian' choices in moral dilemmas do not necessarily demonstrate that the participant is deontologist or utilitarian (see also ref. 64). Instead, those choices should be interpreted more cautiously as being consistent with following moral rules or CBR. If we had used participants' decisions in moral dilemmas to measure deontology and utilitarianism, we would have concluded that around 50% of people are deontologist (the proportion that chose the rule option in the first trial), rather than the much lower proportion of people that showed no learning from outcomes (around 5%). This Article thereby adds to the existing literature challenging the use of sacrificial dilemmas to measure utilitarian versus deontological decision-making[65,66] and offers an alternative interpretation of these choices in terms of decision strategies.

Our theory also raises the question of how to compare the learning signals from different ethical theories and whether it is possible to have a utility function that is agnostic about which theories people use. Our paradigm is able to capture learning broadly for ethical theories that take consequences into account. This is because we classified the outcomes in our paradigm as good or bad depending on participants' own evaluations and also use these evaluations for our computational models. This sidesteps the question of how people determine what is a morally good or bad outcome. In line with this, the modelling approach we used does not require any integration between the utility functions of different moral theories because we only model intra-individual learning on the basis of a participant's own utility function (rather than trade-offs between the utility functions of different participants). As for deontological theories, the intrinsic rightness of certain actions determines their goodness/badness regardless of their outcomes. We therefore would not expect that people who strictly follow deontology would learn in our paradigm, as they would not learn from outcomes. In line with this, our theory explicitly acknowledges that moral evaluations also depend on other factors, such as moral intuitions about the chosen action itself (see Fig. 1, particularly the arrow going directly from decision to moral evaluation). Consistent with these assumptions, we did indeed find that a small proportion of participants did not learn from the consequences of their decisions in our task.

Our finding that metacognitive learning was consistently more prevalent in the CBR Success condition than in the Rule Success condition raises the question of which experiences and situational factors trigger versus inhibit metacognitive moral learning. We investigated this question through a series of exploratory analyses reported in Section 3. These analyses identified several factors that predict increased versus decreased metacognitive moral learning, including taking the moral dilemmas seriously, the perceived plausibility of the outcomes, the emotional experience of the outcomes, the perceived informativeness of the outcomes and their relevance to the real world, and the perceived utility of learning. However, we also found that none of those factors differed significantly between the CBR Success and Rule Success conditions.

In principle, less learning in the Rule Success condition could have occurred because the majority of our participants already relied on rules in the first dilemma. However, we consistently found that for the first dilemma, around half of the participants chose the rule option, and the other half chose the CBR option in both conditions (Experiment 1: across both conditions, 50% chose the CBR option; Experiment 2: 55%; Experiment 3: 55%; Experiment 4: 54%). It therefore seems unlikely that they had a stronger prior that one option would work better over the other.

These results suggest that the difference in the amount of systematic change between the two conditions is unlikely to result from unintentional differences between conditions in the paradigm. Instead, it might reflect an inherent difference between CBR and following moral

rules: while there is only a single CBR strategy, there are a vast number of different rules one could learn to follow. Our paradigm reflected this reality: the pertinent moral rule(s) differed across the 13 dilemmas. In the CBR Success condition, learning was easier because participants only had to learn about the high effectiveness of CBR, and the decision strategy of CBR is more generally applicable in different contexts than any particular moral rule. By contrast, in the Rule Success condition, metacognitive learning could guide participants either to rely more on rules in general (that is, relying on rules leads to better outcomes) or to rely more on specific rules (for example, 'tell the truth' and 'do not kill'). In our experiments, the pertinent rule differed across dilemmas. Participants who learned about specific rules therefore observed less evidence for the effectiveness of any one rule. Moreover, even when those participants learned to rely more on one of the rules that led to good outcomes, this learning did not necessarily show in subsequent dilemmas where the pertinent moral rules were different. Future research could test this interpretation by conducting experiments in which a salient moral rule is held constant.

Our findings also raise the question of why we found a more consistent effect on participants' choices (consistent across all four experiments) than on their moral judgements (for which the effect was significant in only some of the experiments). We conducted a cross-study analysis that showed evidence for an overall effect on judgements and no statistically significant evidence for moderation by experiment or condition (Rule Success versus CBR Success; Section 3). However, this still raises the question of why the effect on judgements was weaker than that on choices. One possible explanation for the stronger effect of experience on choices is that when giving a judgement of moral rightness, (some) participants might have interpreted the question ("How morally right is it for you to [action under consideration]?") as asking solely about the intrinsic rightness of the action regardless of its consequences in the specific situation. Prior research shows that these types of moral judgements often involve different considerations from decisions that would imply reduced learning. For instance, moral judgements tend to be driven more by reputational concerns[67]. Given that social incentives strongly favour the expression of deontological over utilitarian convictions[68], one might expect that people's ratings are always biased towards deontological principles independent of the anticipated consequences. Future work could explore this by developing a scale that includes different questions, some of which are focused more on the action and others that are focused more on the outcomes, and validating those questions against behavioural measures.

In addition to these future directions, our findings open up several other avenues for future research. Our new paradigm enables rigorous experiments on moral learning from consequences and lays the groundwork for these follow-up studies. First, our demonstration of metacognitive moral learning raises the question of what the underlying mechanisms are. We have taken a first step towards developing and comparing models of model-free and model-based metacognitive moral learning. Our observation that metacognitive moral learning appears to be more model-based than model-free is consistent with a long series of findings suggesting that model-based learning contributes to many instances of learning that were once assumed to be purely driven by simple model-free RL (for example, refs. [69–72]). However, our experiments were not optimized for this comparison, and our models also differed along another dimension. That is, the model-free model learns from continuous moral evaluations, whereas the model-based model learns only about the probabilities of binary events (good versus bad). The main goal of our modelling was to assess the transfer for metacognitive learners, which is why we estimated evidence for metacognitive learning via Bayesian model-averaging over model-based and model-free models; our findings are robust to the specific learning style being assumed (Supplementary Methods).

Furthermore, in terms of the behavioural measure of model-based learning, which used ratings of the probabilities of the different outcomes, there is a possibility of rationalization: when asked to reflect on the probability of different outcomes, purely model-free learners may derive the judgement that negative consequences are more likely from their negative model-free evaluation of the action, even though they did not learn in terms of the probabilities of the outcomes during the task[73]. While the fact that we did not observe an effect on the measure of model-free learning provides some evidence against this account, a definitive answer will require an experimental paradigm where model-based and model-free mechanisms produce qualitatively different behaviours[72]. To address this limitation, we are developing an extension of the two-step task to moral decisions contrasting different decision strategies[74].

Second, it remains unclear which types of people are more likely to engage in metacognitive moral learning. Although several aspects of people's perception of our task predicted metacognitive learning, we did not find any relationships with stable individual differences, except for a barely statistically significant association with open-minded thinking about evidence (Supplementary Results).

Third, follow-up research could test how stable the moral learning induced by our paradigm is over time. Experiment 4 showed that the effects of learning are not fleeting, as the transfer effects were observed after an average time delay of about two hours. However, considering that the two experiments were still conducted relatively close together in time, it would be necessary to implement these studies with a larger time delay to draw stronger conclusions about how long these effects last.

Fourth, future research could explore the effects of variations in the reinforcement schedule. In the current experiment, we focused on a simple reinforcement schedule, where the rule and CBR options always or never led to success. The reasons for this choice were mostly pragmatic: our task has fewer trials than other RL tasks, and it is difficult to increase the number of trials much more without making the task too long. The current schedule therefore achieves the strongest learning signal, given the small number of trials in our experiment. One straightforward modification is to introduce probabilistic rewards (for example, CBR leads to success 80% rather than 100% of the time). Our preliminary results from a new moral learning paradigm with probabilistic outcomes suggest that metacognitive moral learning from the consequences of past decisions probably also occurs when the CBR/rule option leads to better outcomes only 70% of the time[74]. In line with research on intermittent conditioning[75], probabilistic reinforcement may lead to stronger behaviour maintenance once a given level of behaviour is reached and would therefore also be valuable for future work aimed at probing the temporal stability of moral learning.

Fifth, future research should explore different learning signals. Although our experiments focused on learning from consequences, in real-world situations where consequences are unobserved, delayed or ambiguous, other factors, such as social considerations[25,43,44,76], might have a stronger influence on the moral evaluations people learn from. Our paradigm could be adapted to use different kinds of outcomes.

Finally, future research should investigate what role metacognitive moral learning plays in moral development and moral learning in the real world. The real-world context that is most similar to our experimental paradigm is learning from stories. The stories we tell our children often teach moral lessons via the fictitious consequences of the protagonists' moral decisions, and so do some of the novels and movies we read and watch. Some teach us that overriding moral rules for anticipated benefits (CBR) leads to good consequences (for example, *Robin Hood* and *The Imitation Game*). Many others dwell on the tragic consequences of being swayed by the anticipated benefits (CBR) of breaking a moral rule (for example, *Les Misérables*, *The Mist* and *Minority Report*). The learning we demonstrated in our experiments

probably occurs when people encounter such stories. In our experiment, the evidence for metacognitive moral learning was strongest when participants perceived the outcomes to be highly realistic (Supplementary Results, Experiment 4, Other Exploratory Measures). This suggests that metacognitive moral learning might be even more powerful for real moral decisions with real consequences.

It has often been argued that human morality is fallible and that people are often swayed by morally irrelevant details[11]. While this may be true of people's decisions in traditional philosophy thought experiments, our experiments offer a more optimistic perspective using realistic moral dilemmas: when people experience the outcomes of their moral decisions, they can learn to adopt decision strategies that are more likely to yield outcomes they consider to be morally good. Moreover, when people's moral judgements of the consequences are sufficiently impartial, as they were in our experiments, the lessons they learn from those consequences can benefit the greater good. Thus, with sufficient experience, people's morality can, in principle, become more adaptive. From the perspective of ecological rationality[77], there is hope that this learning mechanism might tailor human morality to the demands of everyday life[13]. While we do not know whether following rules or CBR would lead to better outcomes in real life (and there is probably no domain-general answer to this question), our research suggests that in situations where people receive frequent, prompt and accurate feedback about the consequences of past decisions, their moral decision-making might be less inconsistent than their responses to thought experiments suggest (compare ref. 78).

The human capacity for moral learning demonstrated by our experiments is a crucial prerequisite for moral progress[79,80] Unlike social learning, which can propagate bias and prejudice[81], moral learning from the consequences of past decisions can ground people's subjective sense of right and wrong in the objective reality of what alleviates versus causes suffering and what promotes versus reduces well-being[26]. Some argue that moral progress has been too slow, leaving common morality unprepared for some of the biggest moral problems of the twenty-first century[82]. As an optimistic counterpoint, our findings suggest that when people observe the consequences of their decisions, moral learning can be fast and adaptive.

## Methods

All experiments were carried out in accordance with approved ethics protocols and complied with pertinent laws and regulations. We obtained informed consent from all participants. Our experiments received ethical approval from the Independent Ethics Commission of the Medical Faculty of the University of Tübingen under protocol number 429/2024BO2; the Office of the Human Research Protection Program (OHRPP) at the University of California, Los Angeles (UCLA), under protocol number IRB#23-001436; and the University College London (UCL) Psychology Ethics Committee under code EP/2018/005. Information about the specific ethics boards and payments are available in the 'Participants' section of each individual study.

All studies were preregistered. The analysis code, materials and preregistration for all experiments are available at https://osf.io/4up5z.

### The moral learning paradigm

The moral learning paradigm comprises a series of 13 moral dilemmas in which participants have to choose between two options. After each choice, they are shown one of the four possible outcomes before moving on to the next decision in a new moral dilemma. Which outcome they see is fully determined by their choice (yes versus no) and the condition they were in (CBR Success versus Rule Success). The following two sections explain the nature of the moral dilemmas and the possible outcomes of the participant's decision, and illustrate them using a concrete example.

The full text of all 13 moral dilemmas, the action choices and their consequences are available in the online repository.

**Realistic moral dilemmas.** To develop our moral learning paradigm, we built on the work by Bennis et al.[12] and Bauman et al.[52] to create vignettes describing realistic moral dilemmas based on historical events[83]. These include dilemmas that some individuals have faced in real life, such as whether to quit their job in a research lab that tests on animals, whether to buy stolen financial records to convict tax evaders and whether to legalize physician-assisted suicide. We adapted those dilemmas to ensure that the consequences of each action are uncertain and sometimes unexpected. Following Körner and Deutsch[83], we addressed the issue that the trolley problem confounds the distinction between CBR and rules with differences between action and omission by including vignettes where the action under consideration is endorsed by a moral rule and CBR advises against it as well as vignettes where this association is reversed.

We adapted 13 realistic moral dilemmas from Körner and Deutsch[83], which were selected on the basis of the feasibility of augmenting each scenario with plausible positive and negative outcomes for both actions and ethical considerations. Our dilemmas covered a range of scenarios involving different rule violations (killing, committing fraud, endorsing crime, animal suffering and disrespect for crime victims) and different contexts (accidents, war/terrorism, medicine, crime, animal rights and justice). This ensured that the moral rule(s) conflicting with the action recommended by CBR varied across dilemmas and allowed us to assess the generality of moral learning from consequences.

We edited the vignettes for clarity and to better suit the purpose of our study. We also added a fake moral dilemma in which the participants were instructed to take a clearly inferior action as an attention check.

The following is an example of a dilemma used in the study:

It is 1987 and you are on a ferry from Belgium to England. Suddenly, the ferry starts tilting and water begins to pour in. You and some other passengers are trying to get to the deck by a rope ladder. You are currently halfway up the ladder. Directly below you, a man who seems frozen into immobility by fear or cold is blocking the ladder. You try to speak to him, but he does not react. People behind you are jostling. The ship seems to be sinking fast and the man is still blocking the ladder. From the crowd below, someone shouts that you should push the man off. If you push the man off the ladder, he will probably die, but the other people will be able to climb on deck. If you do not push the man off the ladder, he will probably continue blocking the way so that many of the people behind you will not be able to get on deck and therefore will drown.

(0 = "Not at all morally right", 100 = "Completely morally right")
Do you push the man off the ladder? (Yes/No)

In this example, the CBR option would be to push the man, whereas the rule option would be to not push the man. Note that, like most other rules, many moral rules or norms prescribe what one ought not to do. Therefore, in our paradigm, the rule option is the choice that involves not committing a moral violation.

Importantly, we deconfounded between action/omission and decision strategies by randomizing whether the CBR or rule option was framed as the action under consideration. Eight vignettes asked the participants if they would perform the CBR-based action, and five vignettes asked if they would perform the rule action. As an example, in one vignette where the CBR choice coincides with action, one must decide whether to push a man off a ship to save many more passengers. Here, the action (pushing the man) would be the option recommended by CBR, but it violates a moral rule. In one vignette where the rule option coincides with action, one must decide whether to quit one's job as a veterinarian who uses animal experiments to develop vaccines.

Here, the action (quitting the job) is the option recommended by a moral rule (do not kill animals), but not by CBR (continuing the job could save many more animals than are harmed in the research).

**Outcomes of decisions.** After making a decision in the dilemma, the participants were shown the outcomes of that choice. Positive outcomes were always shown in green and the negative outcomes in red (note that in Experiment 1, only 6.25% of participants mentioned that these colours played a role in their decision strategy). In this example, participants in the CBR Success condition would see one of the following outcomes depending on their choice:

> Yes Success: You push the man off the ladder. He falls off the boat and you hear a loud splash as he enters the water. The people behind you start to climb on deck. In the end, your decision saves all of the remaining passengers—but the man dies in the process.
> No Failure: You do not push the man off the ladder. He remains frozen and continues to block the way for all the other passengers. In the end, your decision does not save anyone: you watch as the man and all of the remaining passengers die.

Participants in the Rule Success condition would see one of the following outcomes:

> No Success: You do not push the man off the ladder. Shortly after, he attempts to move, but is not physically able, so he stumbles and falls off. You hear a loud splash as he enters the water. The people behind you start to climb on deck. In the end, your decision saves all of the remaining passengers—but the man dies in the process.
> Yes Failure: You push the man off the ladder. However, his foot catches onto the ladder and as he falls, the ladder also falls down with him. You hear a loud splash as the man and ladder enter the water. Without the ladder, the remaining passengers have no way of making their way up to the deck. In the end, your decision does not save anyone: you watch as the man and all of the remaining passengers die.

After reading the outcome, the participants gave a moral evaluation of the outcome by answering the following question:

> How good or bad is this outcome? (−100 = "Extremely bad", 0 = "Neutral", 100 = "Extremely good")

Many of the overall good outcomes also include a small negative consequence in addition to the larger positive consequence. We ensured that participants evaluated the overall good outcomes as positive and the overall bad outcomes as negative by pre-testing the materials in a pilot study ($N = 27$). On average, all positive outcomes were evaluated as good (>0) and all the negative outcomes as bad (<0). A figure depicting the ratings for all outcomes is shown at https://osf.io/q6jr4. On the basis of the results of the pilot, we then modified all outcomes that were evaluated as relatively neutral to ensure that the vignettes and outcomes would be interpreted as intended in the main experiment and that the manipulation would be effective.

## Experiment 1
Experiment 1 received ethical approval from the Independent Ethics Commission of the Medical Faculty of the University of Tübingen under protocol number 429/2024BO2. The experiment and data analysis were preregistered at https://osf.io/jtwvs on 22 March 2023.

**Participants.** We recruited 421 participants from Prolific on 22 March 2023 on the basis of an a priori power analysis. The participants were primarily from the UK (we initially selected US and UK participants, but because we had posted the study in early morning US time, we had only four participants from the USA). and were pre-screened for age (18–70 years), fluency in English, and how frequently and how attentively they had participated in previous studies on the platform (that is, approval rate of ≥95% and had previously participated in at least five studies). We paid participants £2.70 for the 22-minute study. As preregistered, we excluded participants who were too fast ($N = 0$) and those who did not pass the attention check ($N = 34$). Our final sample size was $N = 387$ (mean age, 42.2; s.d. of age, 13.3; $N_{female} = 194$; $N_{male} = 191$; 2 participants did not report their age and gender).

**Materials and procedure.** The participants completed the experiment on the survey platform Qualtrics. They were randomly assigned to one of two conditions: CBR Success ($N = 195$) and Rule Success ($N = 192$).

After reading through the instructions (and passing a comprehension test), the participants completed 13 trials of the moral learning paradigm described in the previous section. After the last trial, the participants completed the Sacrificial Harm Subscale of the OUS[53] (note that we did not include the Impartial Benevolence Subscale because all dilemmas were related to trade-offs between CBR-based and rule-based decision-making, so we only expected an effect on the Sacrificial Harm Subscale). The participants then completed the Cognitive Reflection Test[84]. While our hypothesis about the former was preregistered, our analysis of the latter was exploratory.

Finally, the participants answered three open questions about their experience during the experiment: (1) "Did you change how you made decisions in later compared to earlier scenarios? Why or why not?"; (2) "Did you use information about the outcomes of your choices when making decisions throughout the experiment? If so, how?"; and (3) "Is there anything else you would like to tell the experimenters about how you went through this task?" The last question was optional. We only coded and analysed responses to the first question, as we realized that the word 'outcomes' in the second question was ambiguous: participants might understand it as referring to either the expected outcomes of their choices or the actual outcomes that they saw in the paradigm.

Details of the power analysis and full materials are available in the online repository.

**Data analysis.** We followed the preregistered data analysis plan unless specifically noted. To test the effect of trial number on choices, we preregistered a logistic mixed-effects model predicting the probability of the utilitarian choice from trial number, action framing and the appropriateness ratings in Körner and Deutsch[83]. We had to deviate from the original analysis plan by not using these appropriateness ratings as a covariate in the model for choices because the appropriateness ratings were always given for the action under consideration (that is, how appropriate is it for you to do this action), whereas our outcome variable was framed as the CBR choice (which may also be omission). This made the moral judgements unsuitable for predicting people's actions with the model we had preregistered. For analysing the moral judgements, we preregistered to test the interaction of trial number and framing within a linear mixed-effects model that also included the appropriateness ratings from Körner and Deutsch[83] as a covariate.

More details about the data analysis can be found in Supplementary Information section 1 and the preregistration.

## Experiment 2
Experiment 2 received ethical approval from the UCLA OHRPP under protocol number IRB#23-001436. The experiment and data analysis were preregistered at https://osf.io/7ds8a on 27 November 2023, except for the cognitive modelling; we present a preregistered replication of these modelling results in Experiments 3 and 4.

**Participants.** We recruited 420 UK participants from Prolific on 27 November 2023 on the basis of an a priori power analysis. The participants were pre-screened for fluency in English and their past approval rate (≥95%) and experience (having previously participated in at least ten studies) on the platform. We paid the participants £4.76 (US$6) for the 36-minute study (base rate of US$5, and to encourage careful reading, a bonus payment of US$1 for passing the attention check). As preregistered, we excluded participants who did not pass the attention check ($N = 40$). Our final sample size was $N = 380$ (mean age, 42.9; s.d. of age, 13.2; $N_{female} = 192$; $N_{male} = 188$).

**Materials and procedure.** As in Experiment 1, the participants were randomly assigned to be in either the CBR Success condition ($N = 196$) or the Rule Success condition ($N = 184$). The participants completed the moral learning paradigm from Experiment 1 with a minor change to the instructions (we replaced the word 'feedback' with 'outcomes' in one comprehension check question when referring to the consequences of decisions to prevent participants from misconstruing the experiment as a social learning task and reduce demand characteristics). The moral dilemmas and outcomes were the same as in Experiment 1. As in Experiment 1, we measured participants' moral judgements, decisions and moral evaluations of how good or bad the consequences of each decision were.

Following the moral learning task, we assessed the participants' learning and decision-making using the following behavioural measures of model-based learning, model-free learning, and metacognitive arbitration.

*Model-based learning.* To measure model-based learning, we showed the participants two vignettes after the main task (developed by Cheung et al.[85]), one with a CBR action and one with a rule action choice framing (that is, either the CBR option or the rule option was framed as the action under consideration). Each participant saw one CBR action and one rule action vignette from a set of six possible vignettes. We then showed the participants descriptions of one good and one bad outcome for each of the two choice options (that is, action versus omission). After reading the potential outcomes of either action, the participants were asked to estimate the probability of the good outcome and the probability of the bad outcome.

As an example, participants first read a dilemma about whether to proceed with an attack on an enemy base at the cost of potentially sacrificing the lives of nearby civilians (CBR action). As shown below, they were then asked to estimate the probability that proceeding with the attack would have good versus bad outcomes overall.

Assume you proceed with the attack. Which of the following outcomes is more likely? (0 = "Bad outcome much more likely", 50 = "Good and bad outcomes equally likely", 100 = "Good outcome much more likely")

1. You proceed with the attack and this leads to a good outcome overall. The attack weakens the enemy's capabilities, but it also kills a group of innocent civilians living in the vicinity. The war ends sooner than expected, saving many more innocent people who would have otherwise continued to suffer because of the ongoing war. In the end, your actions helped save these people—but at the cost of some civilian lives.

2. You proceed with the attack and this leads to a bad outcome overall. The enemy forces are strong and the outpost is well-defended, so it only suffered insignificant damages. As a result of the attack, a group of nearby civilians die as collateral damage. In the end, your actions do not save anyone: a group of innocent civilians die and the war continues.

This was followed by the same question for the outcomes of not proceeding with the action. This question and its outcomes, as well as full materials for the other vignettes, can be found in the online repository. Supplementary Information section 1 provides more details on the data analysis.

*Model-free learning.* Model-free moral learning assigns negative values to actions that previously produced bad outcomes, even when the person knows that their consequences would be benign in the current situation[29]. As such, simulated harmful actions have been found to elicit feelings of aversion even when they do not cause a negative outcome[54]. Therefore, to measure the effects of model-free learning, we asked the participants to imagine performing actions that have a conditioned aversive response in situations where they cannot cause any (real or imagined) harm (for example, shooting someone with a fake gun). If people engaged in model-free moral learning, we would expect participants in the Rule Success condition—where violating rules had bad consequences—to feel more negative affect from imagining performing a harmless norm violation than participants in the CBR Success condition, where breaking moral rules had positive consequences.

We used two vignettes (inspired by Cushman et al.[54] and Miller et al.[86]) where participants imagined themselves as actors in a realistic play where they had to perform certain actions on another actor. Below is an example vignette ('Gun'). The other vignette can be found in the online repository.

In your hand, you hold a prop—a fake handgun, a weighty metal replica of a revolver. You note that the gun looks and feels very real, even though you know it is fake. You attempt to shoot at the wall several times, but nothing happens, as there is no ammunition inside. You move towards the person standing in front of you. You take the gun and point it at their face. You curl your finger around the trigger and slowly pull it as if you were firing a real gun. The gun clicks, and nothing happens.

The participants then completed a modified version of the short-form Positive and Negative Affect Schedule[87], which only included only the following items: Afraid, Upset, Excited, Nervous, Scared, Alert, Distressed. We measured the strength of participants' model-free moral values as the mean over the five negative items for both scenarios (the two positive items were added only to reduce demand characteristics). We used the following prompt to emphasize that people should report even slight emotional responses:

How do you feel about having pulled the trigger? Note that you might not experience strong emotions after reading the text, so you are encouraged to report even the slightest changes. (0 = "Not at all", 25 = "Slightly", 50 = "Moderately", 75 = "Quite a bit", 100 = "Very much")

*Metacognitive arbitration—self-report measures.* To measure how people resolved the conflict between the preferences of alternative decision mechanisms (that is, metacognitive arbitration) depending on their learning history through self-report, we again showed participants two vignettes: one CBR action and one rule action vignette from a set of six possible vignettes from Cheung et al.[85]. (This was the same set of vignettes as those used in the model-based measures, but participants did not see the same vignette more than once.) After each decision, the participants answered two sets of questions about their decision-making style using measures also developed by Cheung et al.[85]. The first set of questions measured their reliance on intuition versus deliberation. The second set of questions measured their reliance on CBR versus rules.

The participants rated their agreement with each of the 12 statements about their decision-making style on a scale from 0 ("Not at all")

to 100 ("Entirely"). We used the mean scores of these items to derive four measures of the metacognitive preference for making moral decisions on the basis of (1) deliberation, (2) intuition, (3) rule-based reasoning and (4) CBR.

*Transfer.* Next, the participants completed the OUS Sacrificial Harm Subscale and a four-item questionnaire about deontological decision-making taken from the DCS[55] (adapted from refs. [88,89]).

To investigate whether participants' learning would generalize to incentivized choices, we also measured their donation decisions. We gave the participants a donation task where they had to decide how to allocate £200 between two charities: 1Day Sooner, an advocacy organization for human challenge trials, versus the Medical Research Foundation, a charity that supports medical research using more conventional methods. Critically, we informed the participants that although human challenge trials violate the moral norm to do no harm by infecting healthy volunteers, they are probably more effective than conventional medical trials because they greatly accelerate the development of life-saving medicines and potentially save more lives (that is, it is the CBR option). In contrast, the more conventional medical charity is probably less effective but does not violate any moral rules (that is, it is the rule option).

The donation task was incentive-compatible. That is, we incentivized the participants to make decisions according to their true preferences by informing them that we would execute the decision of one randomly selected participant. We later randomly selected a participant and executed their decision to donate £160 to 1Day Sooner and £40 to the Medical Research Foundation on 16 February 2024.

*Exploratory measures.* We also included the following exploratory measures.

The participants completed the six-item Need for Cognition Scale[90] and the short-form Self-Reflection Scale[91].

To measure awareness of metacognitive learning, we asked the participants some questions on what moral decision-making strategy (or strategies) they used throughout the study. The participants first defined the decision-making strategy they used in an open response. We then showed them seven statements about awareness of having changed their strategy during the study.

We used five questions to measure self-reported metacognitive learning (for example, whether participants reported thinking about choosing to rely on rules versus CBR).

We also included three additional questions from Cheung et al.[85] that asked the participants whether they explicitly traded off the benefits of the CBR option against the cost of breaking the rule (that is, integrated the strategies).

For details about these and other exploratory measures, see Supplementary Information section 1.

**Data analysis.** We used the same analysis as in Experiment 1 with an additional analysis focused on the effect on metacognitive learners. To test moderation effects and the effects for those participants who showed evidence for metacognitive learning, we first calculated the evidence for metacognitive learning for each participant. We did this by calculating an inclusion BF comparing the posterior odds of models that describe strategy learning (MF-M and MB-M) with models that describe behavioural learning from action/omission (MF-B and MB-B) or no learning (C-M and C-B). The code for calculating the inclusion BFs is available in the online repository and the preregistration. We estimated marginal likelihoods using the bridgesampling package[92]. For more information on inclusion BFs, see Hinne et al.[93] and Maier et al.[94,95].

To identify transfer effects only for metacognitive learners, we tested the effect of condition in a model that uses evidence for

metacognitive learning (in terms of an inclusion BF contrasting MB-M and MF-M with the other four models) as a covariate. We then recentred the inclusion BF covariate so that 0 indicates strong evidence for metacognitive learning. This allowed us to test the effect of the experimental condition for those participants who showed evidence for metacognitive learning. We also replicated our results focusing only on model-free or model-based learners (R Markdown files with these results are provided in the online repository).

### Experiment 3

Experiment 3 received ethical approval from the UCLA OHRPP under protocol number IRB#23-001436. The experiment and data analysis were preregistered at https://osf.io/7guj6 on 13 January 2024.

**Participants.** We recruited 900 UK-based participants from Prolific on 13 January 2024 on the basis of an a priori power analysis, which indicated sufficient power with 429 participants per condition for $d = 0.17$ on the transfer measures (on the basis of Experiment 2). The participants were pre-screened using the same criteria and paid the same amount as in Experiment 2. As preregistered, we excluded participants who failed the attention check ($N = 66$). Our final sample size was $N = 834$ (mean age, 43.4; s.d. of age, 14.2; $N_{female} = 424$; $N_{male} = 410$).

**Materials.** We showed the participants the same moral learning paradigm as in Experiments 1 and 2 (CBR Success: $N = 420$; Rule Success: $N = 414$). After the paradigm, the participants completed self-report measures of moral convictions (the same as Experiment 2) and a donation task in a randomized order.

For the donation task, the participants made a series of three donation decisions. Each time, they were asked to allocate £200 between two charities. The three pairs of charities used in the three allocation decisions were:

- 'Human Challenge Trials': Choice between 1Day Sooner (CBR option) and the Medical Research Foundation (rule option).
- 'Animal Testing': Choice between Breast Cancer Now, an organization funding animal research to combat breast cancer (CBR option), and Breast Cancer UK, an organization that does not fund animal research (rule option).
- 'Doctors': Choice between UK-Med, a charity providing humanitarian and medical aid by deploying health-care professionals to conflict or disaster zones that are inherently risky (CBR option), and Pathway, a charity that focuses on health care for homeless people in the UK (rule option).

Participants made these allocations knowing that the experimenters would execute the allocation decision of one randomly chosen participant for one randomly chosen pair of charities. We later executed one participant's decision to donate £150 to 1Day Sooner and £50 to the Medical Research Foundation on 16 February 2024. The full vignettes are available in the online repository.

At the end of the study, as exploratory measures, we included the Empathic Concern and Emotional Empathy scales[96]. We also included six items from a Matrix Reasoning task[97] to measure cognitive ability.

**Data analysis.** To test donation behaviour, we tested the main effect of donations to CBR charities between conditions. As preregistered, we used an analysis of variance model with random intercepts, main effects of condition and donation type, and an interaction between condition and donation.

We used the same analytical approach as in Experiment 2 to analyse the effect of learning, transfer and transfer on metacognitive learners.

## Experiment 4

Experiment 4 received ethical approval from the UCLA OHRPP under protocol number IRB#23-001436 and from the UCL Psychology Ethics Committee under code EP/2018/005. The experiment and data analysis were preregistered at https://osf.io/dgsfb on 19 July 2024.

**Participants.** We recruited 1,100 UK-based participants from Prolific on 19 July 2024. The participants were pre-screened using the same criteria as in Experiments 2 and 3. For the first study of this two-study paradigm, we paid the participants £3.78 (US$4.80) for the 28-minute study (base rate, US$4; bonus payment for passing the attention check, US$0.80). For the second study, which took five minutes, we paid the participants £1 (US$1.27).

A total of 1,100 people took part in the first study, and 811 fully completed both studies. As preregistered, of those who took part in both studies, we excluded some for failing the attention check ($N = 66$) and those of the remaining participants who indicated that they had participated in a study by any of the experimenters before ($N = 18$). Supplementary Fig. 10 visualizes the average time difference between finishing the first part of the study and starting the second part. While some participants started the second part directly after the first (23% of participants had a time gap of less than ten minutes), the majority of our participants had a considerable gap between the two studies (62% of participants had a time gap of more than one hour, and 45% had a time gap of more than two hours).

Our final sample size was $N = 727$ (mean age, 39.43; s.d. of age, 13.15; $N_{female} = 366$; $N_{male} = 361$).

**Materials and procedure.** The experiment comprised two separate tasks on Prolific, which we will call Experiments 4a and 4b. Experiment 4a comprised the moral learning paradigm from Experiments 1–3 followed by the exploratory measures, the dilemmas to measure model-based versus model-free learning and the dilemmas to measure risk aversion described below. Experiment 4b ($N = 727$) comprised only measures of transfer—namely, the measures of people's utilitarian and deontological moral convictions from Experiments 2 and 3 and the donation decisions described below. For these participants, $N = 371$ were in the CBR Success condition and $N = 356$ in the Rule Success condition in Experiment 4a.

*Recruitment procedure.* Experiment 4a and Experiment 4b were two separate online studies run by different researchers from different institutions. Experiment 4a was posted from F.L.'s Prolific account as a task called 'Moral decision-making study'. This task was described as a study run by a researcher at UCLA; it used the consent form of an IRB protocol issued by UCLA. Experiment 4b was posted from V.C.'s Prolific account as a task called 'Donation decisions'. This task was described as a study run by a researcher from UCL and used the consent form from an IRB protocol issued by UCL. Experiment 4 can therefore be viewed as two independent studies that were later joined together for a cross-study analysis.

The recruitment for Experiment 4b started about half an hour after the start of Experiment 4a and remained open for about 12 hours. Experiment 4b was visible only to workers on Prolific who had completed Experiment 4a, but it was impossible for them to know this. Workers who had completed Experiment 4a simply received an email from Prolific inviting them to Experiment 4b (as is standard recruitment procedure) without any further information.

Thus, from the participants' perspective, Experiment 4a and Experiment 4b were unrelated. We confirmed this assumption by explicitly asking the participants at the end of the second study whether they had ever participated in a study "conducted by any of the same experimenters before". We added the following clarification below the question: "Note that this refers to whether you have participated in other studies by the same specific researcher. Do not tick yes if you

have only participated in other studies from UCL run by different researchers. This is also not an attention check and your response would not affect your pay, so please answer honestly." Only 2.42% said yes, and we excluded these participants from the analysis.

*Dilemmas to measure risk aversion.* Instead of learning about CBR versus rules, participants might learn to become more versus less averse to the risk of doing harm. In the CBR Success condition, participants might learn that risking some harm (by choosing the CBR option) always turns out well. This would increase their preference for the CBR option in future decisions by making them less risk-averse. The opposite might occur in the Rule Success condition. To test this alternative hypothesis, we added a set of two vignettes ('Firefighter') measuring participants' risk aversion after the moral learning paradigm.

In the Firefighter vignette, participants decide which of two groups to save and which group to sacrifice. In the first dilemma, one option is to save eight people (with 100% certainty that three will die), and the other is to save three people (with a 75% chance that eight will die). In the second dilemma, one option is to save five people (with a 50% chance that 20 will die), and the other is to save 20 people (with 100% certainty that five will die). We measured participants' choices and how morally right they considered the first option to be. We calibrated the numbers of people in each dilemma using pilot studies and on the basis of prospect theory probability and utility weighting functions to avoid ceiling or floor effects[98].

If participants are learning about risk aversion, then we would expect that those in the CBR Success condition are less risk-averse than those in the Rule Success condition. Furthermore, if learning about risk aversion is the only learning mechanism, we would expect the effect of the experimental condition on rules versus CBR to disappear when controlling for risk aversion. For the full vignettes, see Supplementary Information section 1.

*Exploratory measures.* Directly after completion of the learning task, we included questions about perceived engagement with the task, perceived utility of learning from outcomes and perceiving the scenarios as informative about the real world (Supplementary Information section 1). At the end of the study, we included the Actively Open-Minded Thinking About Evidence Scale[99] and the Certainty of Knowledge Subscale of the Epistemic Belief Inventory[100].

*Transfer to new study.* Experiment 4b included the 'Human Challenge Trials' and 'Animal Research' donation vignettes from Experiment 3. Participants made these allocations knowing that the experimenters would execute the decision of one randomly chosen participant for one randomly chosen pair of charities. We later donated £160 to 1Day Sooner and £40 to the Medical Research Foundation on 28 June 2024.

We excluded the 'Doctor' vignette from this study because a pilot study conducted after Experiment 3 revealed that, contrary to our assumptions, participants neither considered sending doctors to crisis areas to be the option endorsed by CBR, nor did they think that donating to the homeless health-care charity was the option endorsed by moral rules (Section 3).

**Data analysis.** We use the same analytical approach as in Experiment 3.

## Computational models of moral learning from consequences

We implemented all models using the probabilistic programming language Stan[101] and estimated the marginal likelihoods of the data under each model using bridge sampling[92]. The Stan code for the models and the R code for model fitting are in the online repository.

**Model-free metacognitive learning ($Q$-learning).** We formalized model-free metacognitive learning via a $Q$-learning model of how people solve the meta-control problem of deciding when to rely on moral

rules versus CBR (Fig. 1). This model learns to predict the anticipated moral value $Q^{\text{meta}}(s, \text{CBR})$ of relying on CBR in the current situation $s$ and the moral value $Q^{\text{meta}}(s, \text{rules})$ of relying on moral rules. Assuming that in trial $t$ control was allocated to CBR, then, once the consequences of the resulting action are observed, the model calculates the moral prediction error (MPE):

$$\text{MPE}_t = Q_t^{\text{meta}}(s, \text{CBR}) - \text{MJ}_t, \tag{1}$$

which is the difference between the model's prediction of the moral value of relying on CBR ($Q_t^{\text{meta}}(s, \text{CBR})$) and the person's moral evaluation of how good their decision was after they observed its consequences ($\text{MJ}_t$). The participant provided a rating on a scale from −100 to +100. To obtain $\text{MJ}_t$, we divided that rating by 100.

The model then uses the MPE to update its estimate of the moral value of using CBR according to equation (2). How strongly this prediction is updated depends on the learning rate ($\alpha$):

$$Q_{t+1}^{\text{meta}}(s, \text{CBR}) = Q_t^{\text{meta}}(s, \text{CBR}) - \alpha \times \text{MPE}_t, \tag{2}$$

Conversely, when the decision was made by applying a moral rule, then the equivalent update was applied to the estimated value of rule-following (that is, $Q_t^{\text{meta}}(s, \text{rules})$).

In each new decision situation, the learned values of relying on CBR or rules determine the probability that the decision maker will engage in CBR or apply moral rules according to the softmax decision rule specified in equation (3):

$$p_t(s, \text{CBR}) = \frac{e^{\tau \times Q_t^{\text{meta}}(s, \text{CBR})}}{e^{\tau \times Q_t^{\text{meta}}(s, \text{CBR})} + e^{\tau \times Q_t^{\text{meta}}(s, \text{rules})}} \tag{3}$$

The parameter $\tau$ (inverse decision temperature) controls how deterministically the meta-controller allocates control to the decision mechanisms that it expects to produce morally better outcomes. Larger values of $\tau$ imply more deterministic meta-control, whereas lower values of $\tau$ imply more random meta-control.

As the prior distribution on the temperature parameter $\tau$, we used the log-normal distribution lognormal(0, 1.4). We chose this prior distribution because it assigns 90% of the prior probability mass to values of $\tau$ between $\frac{1}{10}$ and 10. The prior distribution on the learning rate is a uniform distribution on the interval [0, 1]. This prior reflects the belief that learning rates larger than 1 (that is, changing your belief by more than the prediction error) and learning rates smaller than 0 (that is, learning the opposite of what the prediction error suggests) are impossible.

**Model-based metacognitive learning (beta-Bernoulli updating).** For the model-based learning model, we assumed that the meta-controller learns the probabilities that a decision made by CBR versus rules will lead to a good versus bad state (for example, $P(s' \in \mathcal{G}|s, \text{CBR})$, where $\mathcal{G}$ is the set of good states). We modelled this as Bayesian learning with conjugate priors. For each decision mechanism, the prior is a beta distribution on the probability $\theta_t^{\text{CBR}}$ that the outcomes will be good overall. The likelihood function is a Bernoulli distribution over two possible outcomes: +1, meaning that the outcome was good overall, and −1, meaning the outcome was bad overall.

Thus, after learning from the person's moral evaluation $\text{MJ}_1, \ldots, \text{MJ}_t$ of the decisions made using the selected decision mechanisms $M_1, \ldots, M_t$ in trials 1, …, $t$, the model's posterior distribution ($P(\theta_t^{\text{CBR}} | \text{MJ}_1, \ldots, \text{MJ}_t, M_1, \ldots, M_t)$) on the probability of good outcomes resulting from CBR is

$$\text{Beta}\left(\alpha + \sum_{i=1}^{t} \mathbb{1}(C_i = \text{CBR and } R_i > 0), \alpha + \sum_{i=1}^{t} \mathbb{1}(C_i = \text{CBR and } R_i < 0)\right), \tag{4}$$

where $\mathbb{1}(\cdot)$ is the indicator function, which is one if and only if its argument is a true statement, and $\alpha$ determines the strength of the prior belief that positive and negative outcomes are equally likely. For higher values of $\alpha$, learning is slower. $\alpha$ therefore serves a similar function as the learning rate of the $Q$-learning model.

Conversely, after the first $t$ trials, the model's posterior distribution ($P(\theta_t^{\text{rules}} | \text{MJ}_1, \ldots, \text{MJ}_t, M_1, \ldots, M_t)$) over the probability that relying on rules will produce a good outcome is

$$\text{Beta}\left(\alpha + \sum_{i=1}^{t} \mathbb{1}(C_i = \text{rules and } R_i > 0), \alpha + \sum_{i=1}^{t} \mathbb{1}(C_i = \text{rules and } R_i < 0)\right). \tag{5}$$

The decision mechanism is again selected using a softmax decision rule based on the learned posterior probabilities of each decision mechanism producing morally good versus morally bad outcomes, as specified in equation (6):

$$p_t(s, \text{CBR}) = \frac{e^{\tau \times \theta_t^{\text{CBR}}}}{e^{\tau \times \theta_t^{\text{CBR}}} + e^{\tau \times \theta_t^{\text{rules}}}}. \tag{6}$$

For the prior distribution on $\tau$, we again use the log-normal distribution lognormal(0, 1.4). For the prior distribution on the prior precision $\alpha$, we use the gamma distribution Gamma(shape = 2.57, rate = 0.54) because this distribution assigns 90% of the probability mass to values between 1 and 10, which is equivalent to having seen between 1 and 10 instances of a positive outcome and between 1 and 10 instances of a negative outcome before starting the experiment.

**Models of model-free and model-based behavioural learning.** Our models of model-free and model-based behavioural learning were exactly analogous to the models of model-free and model-based metacognitive learning described above. The only difference was that the learning rules that the metacognitive models use to learn the value or transition probabilities associated with relying on alternative decision mechanisms (CBR versus following moral rules) are applied to the value or transition probabilities associated with the person's behaviour (action versus omission).

Our model of model-based behavioural learning estimates one single state-transition probability for all behaviours that the vignettes framed as the action under consideration and one single state-transition probability for not performing those behaviours. That is, model-based behavioural learning computes two posterior distributions: one for the probability that taking the action under consideration will lead to a good state (that is, $\theta_t^{\text{action}}$) and one for the probability that not taking that action will lead to a good state (that is, $\theta_t^{\text{omission}}$).

**Constant probability models: no learning.** As a baseline for our models of learning, we formulated equivalent models of what decisions people would make if there was no learning. The baseline for models of metacognitive learning assumes that the probability of relying on CBR versus rules is constant over time; that is

$$p_t(s, \text{CBR}) = \theta_{\text{CBR}}, \tag{7}$$

where $\theta_{\text{CBR}}$ is a free parameter with a uniform prior; that is

$$\theta_{\text{CBR}} \sim \text{Uniform}([0, 1]). \tag{8}$$

The baseline for models of behavioural learning assumes that the probability of performing the behaviour under consideration is constant over time; that is

$$p_t(s, \text{action}) = \theta_{\text{action}}, \tag{9}$$

where $\theta_{action}$ is a free parameter with a uniform prior; that is

$$\theta_{action} \sim \text{Uniform}([0,1]). \qquad (10)$$

**Model implementation.** We implemented all models using Stan and RStan[101,102], fitted them separately for each participant, and evaluated the marginal likelihoods using the bridgesampling R package[92]. We then used bmsR[103] to conduct Bayesian model selection. Since we developed multiple models of metacognitive learning, multiple models of behavioural learning and multiple models of moral decision-making without learning, we compared the proportions of participants best explained by each model family using family-level inference using the MATLAB function spm_compare_families[104,105] with 100,000 samples. We used the same approach to compare the proportions of people best explained by model-based versus model-free learning.

**Model recovery simulations.** We verified that all models could in principle be recovered by simulating from each of the six models, fitting all models on the simulated dataset and checking whether the marginal likelihood of the model that we simulated from was indeed the largest. This indicated that the model comparison consistently recovered the model that was being simulated from. We share the code in the online repository.

### Reporting summary
Further information on research design is available in the Nature Portfolio Reporting Summary linked to this article.

## Data availability
All the data that support the findings of this study are available via OSF at https://osf.io/4up5z/.

## Code availability
All code used is available via OSF at https://osf.io/4up5z/.

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

## Acknowledgements

This work was supported by start-up funds from UCLA and grant no. GR1400 from the Max Planck Institute for Intelligent Systems to F.L., and a grant from the Forethought Foundation for Global Priorities Research to M.M. The funders had no role in study design, data collection and analysis, decision to publish or preparation of the manuscript. We thank F. Bartoš for helpful discussions about computational modelling and for data visualization in Experiment 1, and M. Oh for coding qualitative responses in Experiment 1. We also thank D. Lagnado and the Causal Cognition Lab at UCL, the Crockett Lab at Princeton University, L. Caviola and M. Prentice for their feedback, and F. Cushman for comments on an earlier version of this manuscript.

## Author contributions

All authors were involved in the conceptualization and design of the experiments. V.C. and M.M. created the materials with feedback from F.L., programmed the experiments and collected the data. M.M. preregistered the experiments and conducted the data analysis and visualization. V.C. checked the analysis code. All authors contributed to writing, reviewing and editing the manuscript. F.L. supervised the project.

## Competing interests
The authors declare no competing interests.

## Additional information

**Correspondence and requests for materials** should be addressed to Maximilian Maier.

# Reporting Summary

## Statistics

For all statistical analyses, confirm that the following items are present in the figure legend, table legend, main text, or Methods section.

| n/a | Confirmed | |
|---|---|---|
| ☐ | ☒ | The exact sample size (*n*) for each experimental group/condition, given as a discrete number and unit of measurement |
| ☐ | ☒ | A statement on whether measurements were taken from distinct samples or whether the same sample was measured repeatedly |
| ☐ | ☒ | The statistical test(s) used AND whether they are one- or two-sided *Only common tests should be described solely by name; describe more complex techniques in the Methods section.* |
| ☐ | ☒ | A description of all covariates tested |
| ☐ | ☒ | A description of any assumptions or corrections, such as tests of normality and adjustment for multiple comparisons |
| ☐ | ☒ | A full description of the statistical parameters including central tendency (e.g. means) or other basic estimates (e.g. regression coefficient) AND variation (e.g. standard deviation) or associated estimates of uncertainty (e.g. confidence intervals) |
| ☐ | ☒ | For null hypothesis testing, the test statistic (e.g. *F*, *t*, *r*) with confidence intervals, effect sizes, degrees of freedom and *P* value noted *Give P values as exact values whenever suitable.* |
| ☐ | ☒ | For Bayesian analysis, information on the choice of priors and Markov chain Monte Carlo settings |
| ☐ | ☒ | For hierarchical and complex designs, identification of the appropriate level for tests and full reporting of outcomes |
| ☐ | ☒ | Estimates of effect sizes (e.g. Cohen's *d*, Pearson's *r*), indicating how they were calculated |

*Our web collection on statistics for biologists contains articles on many of the points above.*

## Software and code

Policy information about availability of computer code

| Data collection | Prolific and Qualtrics were used for online data collection. |
|---|---|
| Data analysis | We used R version 4.3.2 and the following R packages and versions. We also share this information in the online repository.<br>- rstan version 2.32.3<br>- stan version 2.26.1.<br>- tidyr version 1.3.0<br>- afex version 1.3.0<br>- stringr version 1.5.1<br>- emmeans version 1.9.0<br>- sjplot version 2.8.15<br>- ggplot2 version 3.4.4<br>- lme4 version 1.1-35.1<br>- bmsR version 0.0.1<br><br>For the BMS analysis, we used MATLAB version R2022a Update 6 with SPM version 8. |

For manuscripts utilizing custom algorithms or software that are central to the research but not yet described in published literature, software must be made available to editors and reviewers. We strongly encourage code deposition in a community repository (e.g. GitHub). See the Nature Portfolio guidelines for submitting code & software for further information.

## Data

Policy information about availability of data

All manuscripts must include a data availability statement. This statement should provide the following information, where applicable:

- Accession codes, unique identifiers, or web links for publicly available datasets
- A description of any restrictions on data availability
- For clinical datasets or third party data, please ensure that the statement adheres to our policy

> All data, materials, and code are available at https://osf.io/4up5z/

## Research involving human participants, their data, or biological material

Policy information about studies with human participants or human data. See also policy information about sex, gender (identity/presentation), and sexual orientation and race, ethnicity and racism.

| | |
|---|---|
| Reporting on sex and gender | We did not include any analyses on the participants' sex or gender, and this was not considered in the study design as it was not relevant to our research questions. We collected only the information that was automatically available to researchers as part of the sample demographic information on Prolific. We share this data on the online repository and consent was obtained for individual-level anonymized data. |
| Reporting on race, ethnicity, or other socially relevant groupings | We did not report on or include any analyses on race, ethnicity, or any other socially relevant groupings, as this was not relevant to our research questions. We collected only the information that was automatically available to researchers as part of the sample demographic information on Prolific. |
| Population characteristics | We used Prolific to recruit online participants from a UK sample with a balanced number of male and female participants. Participants were pre-screened for fluency in English, as we required them to have good comprehension of the task materials which were in English. |
| Recruitment | We used exclusion criteria as described above, with the additional criteria of participants' past approval rate (95%+) and experience (having previously participated in at least ten studies) on Prolific to ensure good data quality. Although the recruitment on Prolific is not entirely representative of the UK population, we did not impose any exclusion criteria based on other demographic or social groups. For more information, see: https://www.pnas.org/doi/abs/10.1073/pnas.2306281121#:~:text=The%20framework's%20factors%20 |
| Ethics oversight | Our experiments received ethical approval from the Independent Ethics Commission of the Medical Faculty of the University of Tübingen, the Office of the Human Research Protection Program, The University of California, Los Angeles (UCLA OHRPP), and the UCL Psychology Ethics Committee. We state the relevant ethics review board for each experiment in the manuscript. |

Note that full information on the approval of the study protocol must also be provided in the manuscript.

# Field-specific reporting

Please select the one below that is the best fit for your research. If you are not sure, read the appropriate sections before making your selection.

☐ Life sciences  ☒ Behavioural & social sciences  ☐ Ecological, evolutionary & environmental sciences

For a reference copy of the document with all sections, see nature.com/documents/nr-reporting-summary-flat.pdf

# Behavioural & social sciences study design

All studies must disclose on these points even when the disclosure is negative.

| | |
|---|---|
| Study description | This was an experimental study where we collected quantitative data. We also collected some qualitative responses from participants, but only included a brief analysis of this data in the supplementary materials. |
| Research sample | We recruited participants on Prolific. We report participant demographic information for each experiment in the Methods section of the paper. |
| Sampling strategy | We used random sampling for all studies. Sample size was based on a predetermined power analysis. We preregistered the planned sample size for all studies. |
| Data collection | Data collection was conducted online. Participants from the recruitment platform Prolific were redirected to complete an online survey on Qualtrics. |
| Timing | The recruited participants on the following dates. Experiment 1: 22nd of March 2023; Experiment 2: 27th of November 2023; Experiment 3: 13th of Jan 2024; Experiment 4: 19th of July 2024 |

| Data exclusions | We used several preregistered exclusion criteria, which we provide in the manuscript and the preregistrations (e.g., excluding participants who failed the attention check). |
|---|---|
| Non-participation | Response rate for each experiment: Experiment 1: 92%; Experiment 2: 94%; Experiment 3: 96%; Experiment 4: 74% (Experiment 4 was a cross-study analysis between two studies; 74% reflects the proportion of participants who participated in Experiment 4b after participating in Experiment 4a. See Methods for more details.) |
| Randomization | Participants were randomized into the between-subject conditions using the randomizer on Qualtrics. |

# Reporting for specific materials, systems and methods

We require information from authors about some types of materials, experimental systems and methods used in many studies. Here, indicate whether each material, system or method listed is relevant to your study. If you are not sure if a list item applies to your research, read the appropriate section before selecting a response.

## Materials & experimental systems

| n/a | Involved in the study |
|---|---|
| ☐ | ☐ Antibodies |
| ☐ | ☐ Eukaryotic cell lines |
| ☐ | ☐ Palaeontology and archaeology |
| ☐ | ☐ Animals and other organisms |
| ☐ | ☐ Clinical data |
| ☐ | ☐ Dual use research of concern |
| ☐ | ☐ Plants |

## Methods

| n/a | Involved in the study |
|---|---|
| ☐ | ☐ ChIP-seq |
| ☐ | ☐ Flow cytometry |
| ☐ | ☐ MRI-based neuroimaging |

## Antibodies

| Antibodies used | *Describe all antibodies used in the study; as applicable, provide supplier name, catalog number, clone name, and lot number.* |
|---|---|
| Validation | *Describe the validation of each primary antibody for the species and application, noting any validation statements on the manufacturer's website, relevant citations, antibody profiles in online databases, or data provided in the manuscript.* |

## Eukaryotic cell lines

Policy information about cell lines and Sex and Gender in Research

| Cell line source(s) | *State the source of each cell line used and the sex of all primary cell lines and cells derived from human participants or vertebrate models.* |
|---|---|
| Authentication | *Describe the authentication procedures for each cell line used OR declare that none of the cell lines used were authenticated.* |
| Mycoplasma contamination | *Confirm that all cell lines tested negative for mycoplasma contamination OR describe the results of the testing for mycoplasma contamination OR declare that the cell lines were not tested for mycoplasma contamination.* |
| Commonly misidentified lines (See ICLAC register) | *Name any commonly misidentified cell lines used in the study and provide a rationale for their use.* |

## Palaeontology and Archaeology

| Specimen provenance | *Provide provenance information for specimens and describe permits that were obtained for the work (including the name of the issuing authority, the date of issue, and any identifying information). Permits should encompass collection and, where applicable, export.* |
|---|---|
| Specimen deposition | *Indicate where the specimens have been deposited to permit free access by other researchers.* |
| Dating methods | *If new dates are provided, describe how they were obtained (e.g. collection, storage, sample pretreatment and measurement), where they were obtained (i.e. lab name), the calibration program and the protocol for quality assurance OR state that no new dates are provided.* |

☐ Tick this box to confirm that the raw and calibrated dates are available in the paper or in Supplementary Information.

| Ethics oversight | *Identify the organization(s) that approved or provided guidance on the study protocol, OR state that no ethical approval or guidance was required and explain why not.* |
|---|---|

Note that full information on the approval of the study protocol must also be provided in the manuscript.

# Animals and other research organisms

Policy information about studies involving animals; ARRIVE guidelines recommended for reporting animal research, and Sex and Gender in Research

| | |
|---|---|
| Laboratory animals | *For laboratory animals, report species, strain and age OR state that the study did not involve laboratory animals.* |
| Wild animals | *Provide details on animals observed in or captured in the field; report species and age where possible. Describe how animals were caught and transported and what happened to captive animals after the study (if killed, explain why and describe method; if released, say where and when) OR state that the study did not involve wild animals.* |
| Reporting on sex | *Indicate if findings apply to only one sex; describe whether sex was considered in study design, methods used for assigning sex. Provide data disaggregated for sex where this information has been collected in the source data as appropriate; provide overall numbers in this Reporting Summary. Please state if this information has not been collected. Report sex-based analyses where performed, justify reasons for lack of sex-based analysis.* |
| Field-collected samples | *For laboratory work with field-collected samples, describe all relevant parameters such as housing, maintenance, temperature, photoperiod and end-of-experiment protocol OR state that the study did not involve samples collected from the field.* |
| Ethics oversight | *Identify the organization(s) that approved or provided guidance on the study protocol, OR state that no ethical approval or guidance was required and explain why not.* |

Note that full information on the approval of the study protocol must also be provided in the manuscript.

# Clinical data

Policy information about clinical studies
All manuscripts should comply with the ICMJE guidelines for publication of clinical research and a completed CONSORT checklist must be included with all submissions.

| | |
|---|---|
| Clinical trial registration | *Provide the trial registration number from ClinicalTrials.gov or an equivalent agency.* |
| Study protocol | *Note where the full trial protocol can be accessed OR if not available, explain why.* |
| Data collection | *Describe the settings and locales of data collection, noting the time periods of recruitment and data collection.* |
| Outcomes | *Describe how you pre-defined primary and secondary outcome measures and how you assessed these measures.* |

# Dual use research of concern

Policy information about dual use research of concern

## Hazards

Could the accidental, deliberate or reckless misuse of agents or technologies generated in the work, or the application of information presented in the manuscript, pose a threat to:

No | Yes

☐ ☐ Public health

☐ ☐ National security

☐ ☐ Crops and/or livestock

☐ ☐ Ecosystems

☐ ☐ Any other significant area

## Experiments of concern

Does the work involve any of these experiments of concern:

No  Yes

☐ ☐ Demonstrate how to render a vaccine ineffective

☐ ☐ Confer resistance to therapeutically useful antibiotics or antiviral agents

☐ ☐ Enhance the virulence of a pathogen or render a nonpathogen virulent

☐ ☐ Increase transmissibility of a pathogen

☐ ☐ Alter the host range of a pathogen

☐ ☐ Enable evasion of diagnostic/detection modalities

☐ ☐ Enable the weaponization of a biological agent or toxin

☐ ☐ Any other potentially harmful combination of experiments and agents

# Plants

Seed stocks | *Report on the source of all seed stocks or other plant material used. If applicable, state the seed stock centre and catalogue number. If plant specimens were collected from the field, describe the collection location, date and sampling procedures.*

Novel plant genotypes | *Describe the methods by which all novel plant genotypes were produced. This includes those generated by transgenic approaches, gene editing, chemical/radiation-based mutagenesis and hybridization. For transgenic lines, describe the transformation method, the number of independent lines analyzed and the generation upon which experiments were performed. For gene-edited lines, describe the editor used, the endogenous sequence targeted for editing, the targeting guide RNA sequence (if applicable) and how the editor was applied.*

Authentication | *Describe any authentication procedures for each seed stock used or novel genotype generated. Describe any experiments used to assess the effect of a mutation and, where applicable, how potential secondary effects (e.g. second site T-DNA insertions, mosiacism, off-target gene editing) were examined.*

# ChIP-seq

## Data deposition

☐ Confirm that both raw and final processed data have been deposited in a public database such as GEO.

☐ Confirm that you have deposited or provided access to graph files (e.g. BED files) for the called peaks.

Data access links
*May remain private before publication.* | *For "Initial submission" or "Revised version" documents, provide reviewer access links. For your "Final submission" document, provide a link to the deposited data.*

Files in database submission | *Provide a list of all files available in the database submission.*

Genome browser session
(e.g. UCSC) | *Provide a link to an anonymized genome browser session for "Initial submission" and "Revised version" documents only, to enable peer review. Write "no longer applicable" for "Final submission" documents.*

## Methodology

Replicates | *Describe the experimental replicates, specifying number, type and replicate agreement.*

Sequencing depth | *Describe the sequencing depth for each experiment, providing the total number of reads, uniquely mapped reads, length of reads and whether they were paired- or single-end.*

Antibodies | *Describe the antibodies used for the ChIP-seq experiments; as applicable, provide supplier name, catalog number, clone name, and lot number.*

Peak calling parameters | *Specify the command line program and parameters used for read mapping and peak calling, including the ChIP, control and index files used.*

Data quality | *Describe the methods used to ensure data quality in full detail, including how many peaks are at FDR 5% and above 5-fold enrichment.*

Software | *Describe the software used to collect and analyze the ChIP-seq data. For custom code that has been deposited into a community repository, provide accession details.*

# Flow Cytometry

## Plots

Confirm that:

☐ The axis labels state the marker and fluorochrome used (e.g. CD4-FITC).

☐ The axis scales are clearly visible. Include numbers along axes only for bottom left plot of group (a 'group' is an analysis of identical markers).

☐ All plots are contour plots with outliers or pseudocolor plots.

☐ A numerical value for number of cells or percentage (with statistics) is provided.

## Methodology

| | |
|---|---|
| Sample preparation | *Describe the sample preparation, detailing the biological source of the cells and any tissue processing steps used.* |
| Instrument | *Identify the instrument used for data collection, specifying make and model number.* |
| Software | *Describe the software used to collect and analyze the flow cytometry data. For custom code that has been deposited into a community repository, provide accession details.* |
| Cell population abundance | *Describe the abundance of the relevant cell populations within post-sort fractions, providing details on the purity of the samples and how it was determined.* |
| Gating strategy | *Describe the gating strategy used for all relevant experiments, specifying the preliminary FSC/SSC gates of the starting cell population, indicating where boundaries between "positive" and "negative" staining cell populations are defined.* |

☐ Tick this box to confirm that a figure exemplifying the gating strategy is provided in the Supplementary Information.

# Magnetic resonance imaging

## Experimental design

| | |
|---|---|
| Design type | *Indicate task or resting state; event-related or block design.* |
| Design specifications | *Specify the number of blocks, trials or experimental units per session and/or subject, and specify the length of each trial or block (if trials are blocked) and interval between trials.* |
| Behavioral performance measures | *State number and/or type of variables recorded (e.g. correct button press, response time) and what statistics were used to establish that the subjects were performing the task as expected (e.g. mean, range, and/or standard deviation across subjects).* |

## Acquisition

| | |
|---|---|
| Imaging type(s) | *Specify: functional, structural, diffusion, perfusion.* |
| Field strength | *Specify in Tesla* |
| Sequence & imaging parameters | *Specify the pulse sequence type (gradient echo, spin echo, etc.), imaging type (EPI, spiral, etc.), field of view, matrix size, slice thickness, orientation and TE/TR/flip angle.* |
| Area of acquisition | *State whether a whole brain scan was used OR define the area of acquisition, describing how the region was determined.* |

Diffusion MRI    ☐ Used    ☐ Not used

## Preprocessing

| | |
|---|---|
| Preprocessing software | *Provide detail on software version and revision number and on specific parameters (model/functions, brain extraction, segmentation, smoothing kernel size, etc.).* |
| Normalization | *If data were normalized/standardized, describe the approach(es): specify linear or non-linear and define image types used for transformation OR indicate that data were not normalized and explain rationale for lack of normalization.* |
| Normalization template | *Describe the template used for normalization/transformation, specifying subject space or group standardized space (e.g. original Talairach, MNI305, ICBM152) OR indicate that the data were not normalized.* |
| Noise and artifact removal | *Describe your procedure(s) for artifact and structured noise removal, specifying motion parameters, tissue signals and physiological signals (heart rate, respiration).* |

| Volume censoring | *Define your software and/or method and criteria for volume censoring, and state the extent of such censoring.* |
| --- | --- |

## Statistical modeling & inference

| Model type and settings | *Specify type (mass univariate, multivariate, RSA, predictive, etc.) and describe essential details of the model at the first and second levels (e.g. fixed, random or mixed effects; drift or auto-correlation).* |
| --- | --- |
| Effect(s) tested | *Define precise effect in terms of the task or stimulus conditions instead of psychological concepts and indicate whether ANOVA or factorial designs were used.* |

Specify type of analysis: ☐ Whole brain  ☐ ROI-based  ☐ Both

| Statistic type for inference | *Specify voxel-wise or cluster-wise and report all relevant parameters for cluster-wise methods.* |
| --- | --- |

(See Eklund et al. 2016)

| Correction | *Describe the type of correction and how it is obtained for multiple comparisons (e.g. FWE, FDR, permutation or Monte Carlo).* |
| --- | --- |

## Models & analysis

| n/a | Involved in the study |
| --- | --- |
| ☐ | ☐ Functional and/or effective connectivity |
| ☐ | ☐ Graph analysis |
| ☐ | ☐ Multivariate modeling or predictive analysis |

| Functional and/or effective connectivity | *Report the measures of dependence used and the model details (e.g. Pearson correlation, partial correlation, mutual information).* |
| --- | --- |
| Graph analysis | *Report the dependent variable and connectivity measure, specifying weighted graph or binarized graph, subject- or group-level, and the global and/or node summaries used (e.g. clustering coefficient, efficiency, etc.).* |
| Multivariate modeling and predictive analysis | *Specify independent variables, features extraction and dimension reduction, model, training and evaluation metrics.* |

