## [Peer Review file · Nature Human Behaviour]

Learning From Outcomes Shapes Reliance on Moral Rules versus Cost-Benefit Reasoning

Corresponding Author: Mr Maximilian Maier

Version 0:

Decision Letter:

9th October 2024

Dear Mr Maier,

Thank you once again for your manuscript, entitled "Metacognitive Learning from Consequences of Past Choices Shapes Moral Decision-Making," and for your patience during the peer review process.

Your manuscript has now been evaluated by 3 reviewers, whose comments are included at the end of this letter. Although the reviewers find your work to be of interest, they also raise some important concerns. We are interested in the possibility of publishing your study in Nature Human Behaviour, but would like to consider your response to these concerns in the form of a revised manuscript before we make a decision on publication.

In sum, we invite you to revise your manuscript taking into account all reviewer and editor comments. We are committed to providing a fair and constructive peer-review process. Do not hesitate to contact us if there are specific requests from the reviewers that you believe are technically impossible or unlikely to yield a meaningful outcome.

We hope to receive your revised manuscript within two months. I would be grateful if you could contact us as soon as possible if you foresee difficulties with meeting this target resubmission date.

- Include a "Response to the editors and reviewers" document detailing, point-by-point, how you addressed each editor and referee comment. If no action was taken to address a point, you must provide a compelling argument. When formatting this document, please respond to each reviewer comment individually, including the full text of the reviewer comment verbatim followed by your response to the individual point. This response will be used by the editors to evaluate your revision and sent back to the reviewers along with the revised manuscript.
- Highlight all changes made to your manuscript or provide us with a version that tracks changes.

Link Redacted

We look forward to seeing the revised manuscript and thank you for the opportunity to review your work. Please do not hesitate to contact me if you have any questions or would like to discuss these revisions further.

Sincerely,

REVIEWER COMMENTS:

Reviewer #1 (Remarks to the Author):

Review Manuscript 30421

The authors report on 4 experiments that highlight the possibility that individuals consider the consequences of their past decisions when making moral decisions. Through these 4 experimental studies, the authors find that learning from the consequences of past decisions leads individuals to choose strategies that benefit the greater good. The authors also show that metacognitive moral learning involves generalization: its effects transfer from a classical utilitarian paradigm involving fictitious dilemmas to incentive-compatible, real-world donation decisions in a different context.

I am generally positive about this high-level research. For their study, the authors propose to mobilize clear and well-described computational models, one targeting model-free metacognitive learning and the other targeting model-based metacognitive learning. Although this is not my area of expertise, the authors describe these models precisely, which seems to me to capture the phenomena the authors are interested in.

I also appreciate the fact that the authors diversify their VDs, using particularly fictional moral dilemmas and exploring a possible transfer to more ecologically viable tasks (e.g. donation).

The authors also follow good research practice: each study is carefully pre-registered and the material, data and scripts are made publicly available, always on an open access basis, in line with the principles of open science.

In general, my only criticism would be a lack of clarity, especially for non-experts at several places. The technical reading is sometimes difficult to follow. Although the supplementary material provides a considerable amount of detail, I feel that the understanding should at least be clear without recourse to this supplementary material. I would highlight these few passages and advise the authors to work on simplifying them or providing additional explanations to make reading easier. The implications of the present results should be more fully developed. I very much appreciate that the authors do not over-interpret or over-generalize, but it seems to me that the conclusion drawn from the results should be given more prominence. What should we learn from these findings? What does it add to our understanding of morality? Do these results offer a break with what we previously knew (it seems to me that they do)? Finally, do these results teach us anything additional, beyond the realm of morality (again, it seems to me that they do)? I suggest that the authors develop these aspects.

The difference between metacognitive learning computations and more traditional reinforcement learning models is difficult to understand. This is especially true for those new to the field. It seems to me that special attention should be paid to this, and the explanations in the main article should be restructured or expanded, as this is one of the keys to the research reported. The authors indicate "Put simply, the mechanisms of metacognitive moral learning differ from standard RL in two key ways: (1) learning occurs in the meta-control system whose "actions" are our decision strategies (e.g., CBR and rule-following), and (2) the reward signal is the person's moral judgment of how good or bad their decision was. These moral judgments are partly based on consequences of the decision [40, 41]. Therefore, learning from the consequences of past decisions could, in principle, adaptively increase people's reliance on decision strategies that produce good consequences, and decrease reliance on those that produce bad consequences. » Is it also possible to indicate the characteristics or features of standard RL to help make the understanding easier?

P7. The authors indicate « In line with this, we found a significant interaction between trial number and whether action coincided with rules or CBR on moral judgments, $b = -1.68$, $t(2340.73) = -2.20$, $p = .028$, two-sided. » The interaction effect should be clarified. Intuitively, we might think that morality differs more between rule-compliant action and CBR on moral judgments of the latter (compared to the former). Whether this is the case should be explained to facilitate understanding.

P7. The authors indicate: However, the interaction between trial number and whether action coincided with rules or CBR was not statistically significant, $b = -1.45$, $t(2239.64) = -1.91$, $p = .057$, two-sided." This would therefore diminish the claim that it is learning from past events, but that individuals are directly considering the first consequence of their decision. Would we still call this learning? This could be discussed.

P8. The authors indicate: « Compared to the CBR Success condition, participants in the Rule Success condition predicted that the rule-based action would be more likely to produce good outcomes, whereas the CBR-based action would be less likely to produce good outcomes, $F(1, 756) = 17.28$, $p < .001$. ». I'm not sure which IVs are included in the model here. Are you reporting a main effect of an interaction? Further clarification would be welcome (even if the methodology section is more fleshed out).

P15. The authors suggest - and I would agree with them - that « everyone might be a consequentialist learner, regardless of

which moral principles they endorse. » This is particularly interesting in that the research proposed by the authors and the results seem to provide an 'impactful' answer to the need to stop considering a utilitarian response as consequentialist and aiming at the greater good, and to consider a deontic response which does not have this aim'. This could be suggested, if the authors agree.

I wish the authors the best of luck with their very high-quality research, which has the potential to offer a significant advance in the field of moral judgement and possibly much more.

Reviewer #2 (Remarks to the Author):

In this paper, Maier and colleagues present a series of experimental studies, augmented by computational modeling and personality surveys, which demonstrate that humans engage in metacognitive moral strategy learning. In particular, subjects came to adopt cost-benefit reasoning when trained on scenarios where this form of reasoning is beneficial. They then generalized this strategy to completely different contexts, and the generalization lasted at least a few hours.

Overall, I thought this was a really nice paper: clearly written, computationally sophisticated, and potentially impactful. I was particularly impressed by the generalization results. My comments below are relatively minor.

I want to raise two high-level issues. One is that they only saw evidence for metacognitive moral learning in the CBR Success condition, not in the Rule Success condition. In the Discussion, the authors pose this as raising a question about "which experiences and situational factors trigger versus inhibit metacognitive moral learning." My concern is that this is essentially post-hoc. An alternative is that the phenomenon is simply not robust. The fact that they see the effect across several experiments, and that it generalizes beyond the experimental paradigm, is reassuring, but it does seem to be a problem for the theory, which lacks any predictions about "triggering" conditions.

The other high-level issue is the characterization of success vs. failure in the vignettes. It seems to me that this takes as given that subjects have a particular utility function for the outcomes of moral dilemmas. But I thought (and here I may be betraying my ignorance of moral philosophy) the different moral frameworks (utilitarianism, consequentialism, etc.) essentially define different utility functions. Is it really possible to have a framework-agnostic utility function for learning at the metacognitive level? I understand that in the model implementation, you're actually using the subjective ratings, but this just pushes the question back further.

Minor comments:

I think the paper would benefit from some schematic diagrams of the experimental tasks, so that readers can get a better idea of what exactly the participants are seeing/doing.

I'm confused about why some tests are two-sided and some are one-sided.

Fig 4: "Metacognitive" -> "Metacognitive"

Somewhere there needs to be an explanation of how the Bayes Factors were obtained for Fig 4 and Fig 5. I think this comes from the Bayesian model selection described in the methods, but this isn't explicit.

There are a few places where p-values are reported without specifying what the statistical test was.

Reviewer #3 (Remarks to the Author):

The current study investigates how people learn to adjust their moral decision-making based on the consequences of their past choices. Across four experiments, the authors demonstrate that individuals adaptively modify their reliance on moral rules versus cost-benefit reasoning (CBR) in response to experienced outcomes. The researchers show that when CBR leads to better consequences, participants gravitate towards those types of choices, whereas if the inverse is true, the inverse happens. Interestingly, they find that these effects transfer to other moral judgments such as real-world donation decisions.

This work exhibits several strengths. The authors introduce a novel moral learning paradigm. The transparency of the work is commendable, with the authors reporting all procedures, analyses, and results in detail. They preregistered all experiments and these appear comprehensive and align with the reported analyses. As far as I can tell, they had sufficient statistical power to detect the reported effects. The core findings are replicated across studies. The inclusion of the incentive-compatible donation decision, adds ecological validity to the study.

All in all, I think this is an interesting piece of work. I will however admit that it took me a couple of reads to understand the manuscript sufficiently for me to be able to provide a review. Part of that is undoubtedly caused because I have never done any computational modeling, but still, my primary recommendation to the authors would be to revise the manuscript with non-technical readers in mind as I do think that several parts of the manuscript could benefit from making it a tad more accessible. I believe that doing so will help them reach a wider audience.

Below, I list some considerations that occurred to me while reading through the manuscript:

1.

One issue I was wondering about while reading the manuscript had to do with the author's framing of "Deontological" decisions as "rule-based" decisions. How to best label decisions in sacrificial dilemmas has been a point of contention and much of that debate has focused on whether "utilitarian" decisions should really be labeled "utilitarian". I appreciate that the authors steer clear of that debate by framing those decisions as "CBR-based". That seems like a fairly neutral descriptor that avoids much of the baggage associated with a term like "utilitarian".

I am however not entirely convinced whether "deontological" decisions are best labeled as "rule-based" decisions either. I realize that type of framing is not uncommon, but to the best of my knowledge, the empirical support for such a view is still somewhat sparse. Maybe the authors are aware of some convincing evidence in that regard, but I am not sure much evidence in that regard exists. In any case, I think it would be good if the authors could provide some more (empirical) support for labeling those decisions as such.

As an alternative, I was wondering whether the authors have considered conceptualizing non-CBR decisions as resulting from "an evaluation of the moral value of an action itself". In my mind, that is a bit more neutral, still fits clearly with the idea of "deontological" (as in the idea that actions have inherent moral value), and it contrasts nicely with CBR decisions as those result from an evaluation of outcomes. I also think this still aligns with the conceptualization suggested by Cushman in his work on model-free / model-based moral cognition. I don't intend this remark as a point of contention, it is certainly possible the authors considered such a framing and decided that the current "rule-based" framing fits better with the operationalization used in their paradigm but I was wondering whether the authors have reflected on that issue and decided against that alternative.

2.

While I like the paradigm the authors have used, and appreciate this novel approach to studying moral judgment, the authors decided to implement somewhat extreme learning conditions where either CBR or rule-based decisions always lead to good outcomes. I can see the methodological benefits, but it does make one wonder to what extent all of the findings represent learning or are the result of task demands. How would the authors counter such a charge? I will say I believe Study 4 is certainly helpful in that regard, but I am not entirely convinced it fully halts such a charge. I would suggest a discussion of that issue might benefit the paper.

Relatedly, I was also wondering whether the authors considered using a more nuanced approach, such as having CBR or rule-based decisions lead to positive outcomes in, say, 80% of cases. I think it might be valuable for the authors to discuss whether they considered less extreme paradigms and to reflect on how such approaches might affect the observed learning processes.

3.

On pages 5-6, the authors write: "Our experimental paradigm varied all of these salient features independently of which option each strategy recommended (see Methods section for more details). Therefore, it is not immediately obvious to participants what is being reinforced in our paradigm. This was also reflected in their responses to an open question in Experiment 1 (see Section S1.1)"

I actually think this is a very important bit of information. I think it might be good to include it a bit more explicitly in the results section because I think it gives more credibility to their findings. As a reader, the manipulation the authors used seemed a bit like a sledgehammer, but if participants don't feel it as such that counters some of my worries.

4.

Generally speaking, I think the authors' introduction of the different types of learning could benefit from a slightly more accessible explanation. I am familiar with the key papers on this issue but I still find the distinction between model-based and model-free learning hard to interpret and the authors could do more to help readers unfamiliar with this to understand the key points. For instance, the authors state that model-based learning results in "two beta distributions: one for the consequences of CBR and one for the consequences of rule-following," while model-free learning "assigns intrinsic moral values to using different decision strategies." An explanation of what these outcomes mean in practical terms and how they differ functionally would enhance the paper's accessibility.

Relatedly, the paper would benefit from a non-technical explanation of the difference between "metacognitive learning" and "behavioral learning" and what the practical difference between these different learning types reflect. The difference is never really explained (perhaps the authors consider it self-explanatory) but I must admit to being somewhat confused by the sudden introduction of "behavioral learning" with the comparison between metacognitive and behavioral learning on page 9. Based on the method sections, I ended up concluding that metacognitive learning reflects participants' increasing reliance on either CBR or Rule-based moral cognition and that behavioral reflects participants learning that either "Actions" or "Omissions" are good/bad, though I am not entirely certain of this interpretation.

5.

Similarly, I struggled a little bit to understand why the specific measure for model-based learning reflects a measure of model-based learning (and actually, the same is somewhat true for the measure of model-free learning). In regards, with the measure for model-based learning in Study 2, participants were basically confronted with a moral dilemma scenario after the learning task. The crucial difference seems to be that whereas, in the learning task, participants pick a dilemma option and receive feedback (it went great vs. it went poorly), in the measure for model-based learning, they are to estimate how likely the two options in the dilemma would lead to a good or bad outcome.

I was struck by the similarity and I struggle a little bit in understanding how such a measure is informative above and beyond the other type of measure. If participants are gradually more likely to pick a certain option, does not also follow that participants believe that option to be more likely to have a better outcome? What is the added informational value here? To me the two measures: a) the option that participants pick and b) what participants consider the option that yields a better outcome, are so similar that they might as well be the same thing.

6.
In regards, to that same measure. Could the authors explain why exactly that measure reflects model-based learning? Allow me to play devil's advocate, let's say we implement a learning procedure that increases the inherent value associated with one of two types of actions. If I understand correctly, that would be an implementation of model-free learning. Then assume that we ask participants who have undergone such a procedure the question "Which actions yield the best outcomes", would we expect to find no difference for participants' judgments of the two types of actions? Based on that measure, on page 8, the authors conclude: "This suggests that participants learned a probabilistic model of the consequences of relying on CBR versus rules, a mechanism we will refer to as model-based metacognitive learning." Why exactly is that? The authors reference model-free learning state the model-free system selects actions "based on their average consequences in the past". Could such a mechanism not also yield this outcome?

7.
Also near the same point in the manuscript, in regards with Experiment 2, p.8 I want to make sure I understand the results correctly. The authors note:
"Compared to the CBR Success condition, participants in the Rule Success condition predicted that the rule-based action would be more likely to produce good outcomes, whereas the CBR-based action would be less likely to produce good outcomes, $F(1, 756) = 17.28, p < .001$."
If I understand correctly, the authors confronted participants with two scenarios and asked on one scenario whether a Rule-based action would lead to positive/negative outcomes and on the other scenario whether a CBR-based action would lead to positive/negative outcomes (see page 21). In the preregistration, I read: "We will calculate a model-based learning score for the CBR action and a separate score for the rule action vignette." I am a little bit confused by the suggestion that two dependents are being compared (CBR-based action across both conditions and Rule-based action across both conditions) but that we are getting only a single statistical test. Are these analyses aggregated?

8.
Across the different Studies, the authors find that learning in the CBR-success condition appears to be primarily driven by model-based metacognitive learning, whereas learning in the Rules-success condition appears to be driven by model-based behavioral learning. This was a little surprising to me as I was (perhaps naively) expecting the latter to be driven by model-free metacognitive learning. I think the authors share that perspective and they do discuss it somewhat in their discussion but I think the manuscript could benefit from a deeper discussion of this issue.

9.
on page 9, the authors report the result of the incentive-compatible donation decision: "participants allocated £200 between a charity supporting conventional medical research and a highly effective charity promoting human challenge trials in which healthy volunteers are infected with a virus to speed up the development of vaccines. According to a preregistered one-sided t-test, participants in the CBR Success condition donated significantly more money ($M = 97.70$) to support human challenge trials than participants in the Rule Success condition ($M = 86.73$), $t(377.54) = 1.67, p = .047, d = 0.17$, one-sided. This indicates an increased reliance on CBR and a decreased reliance on moral rules because this charity breaks the moral rule "do no harm" to save many more lives in total"
Minor point but as far as I can tell this analysis does not allow one to distinguish whether the effect is driven by an increased reliance on CBR, a decreased a reliance on moral rules or a combination of the two.

10.
Page 11, all the figures have typos: Log(BF) for Metacognitive Learning

11.
In experiment 4, the authors test whether the learning effects transfer to a new context. Essentially, they replicate their previous paradigm, but rather than including the transfer measures in the same survey, these measures are included in a separate survey from a different set of researchers.
I personally feel that labeling this as a different context is a bit of a stretch and would recommend a more humble interpretation. Indeed, the measures are not included in the same survey, rather they are included in two separate surveys that participants take sequentially, but I am not sure this sufficient to demonstrate transfer to a new context. To me, this is similar to including a few intermediate tasks between the two parts of the study (i.e. accepting a new study, providing consent a second time, and possibly completing another survey in between).
I think this study works as an argument against the claim that the authors' findings reflect task demands, but I am not sure they show much more than that. I would have found this much more convincing if the two surveys had been separated in time (say a week apart) rather than how they were administered. If I read the methods correctly, the second survey launched approximately at the time the first participants of the first survey were expected to finish. Given that, I was actually wondering how many of the participants in this study completed both studies sequentially?

Reviewer #3 (Remarks on code availability):

I did read through the preregistration but did not check the code.

Version 1:

Decision Letter:

Our ref: NATHUMBEHAV-24072785A

25th February 2025

Dear Dr. Maier,

Thank you for submitting your revised manuscript "Consequentialist Learning Shapes Reliance on Moral Rules versus Cost-Benefit Reasoning" (NATHUMBEHAV-24072785A). It has now been seen by the original referees and their comments are below. As you can see, the reviewers find that the paper has improved in revision. We will therefore be happy in principle to publish it in Nature Human Behaviour, pending minor revisions to satisfy the referees' final requests and to comply with our editorial and formatting guidelines.

We are now performing detailed checks on your paper and will send you a checklist detailing our editorial and formatting requirements within two weeks. Please do not upload the final materials and make any revisions until you receive this additional information from us.

Sincerely,

[REDACTED]

[REDACTED]

[REDACTED]

Nature Human Behaviour

Reviewer #1 (Remarks to the Author):

Review of Manuscript 30421 entitled "Metacognitive Learning from Consequences of Past Choices Shapes Moral Decision-Making"

Generally speaking, the authors have responded in depth to each of my comments and misunderstandings. The addition of the discussion of what the results tell us is a real plus - and the authors present these contributions simply, which makes it possible to really appreciate the contribution. Although it wasn't my comment (but the reviewer 2's), the addition of a diagram explaining the paradigm definitely helps in understanding the protocol. At this point, I have no further comments to make, and am particularly in favor of seeing the article published as it stands.

Finally, I would like to offer the authors my sincere apologies for the delay in sending my report. I hope my very favorable opinion will be a consolation prize.

Again, I wish the authors good luck with their excellent research.

Reviewer #2 (Remarks to the Author):

I am happy with the changes made in response to my comments.

Reviewer #3 (Remarks to the Author):

I read through the revised manuscript and the authors (very thorough) reply letter and can only say that I was pleased to see how the authors approached this revision. As far as I can tell they responded to all comments with a clear and reasonable response and in doing so have clarified the issues I flagged in my review. I think the authors have written an interesting paper and I would be happy to see it in print.

I had one minor concern regarding the following passage. On page 23 in their conclusion the authors write: "It has often been argued that human morality is fallible and that people are often swayed by morally irrelevant details [11]. While this may be true of people's decisions in traditional philosophy thought experiments, our experiments offer a more optimistic perspective: When people experience the outcomes of their moral decisions, they learn to adopt decision strategies that benefit the greater good."

I do love happy endings but I was wondering whether this conclusion is fully appropriate. Just to push back a little: as far as I can tell, what the paper demonstrates is that people learn from the consequences of their decisions in such a way that they end up pursuing those decision-strategies that yield a higher proportion of the outcomes that they consider good. Within the context of these experiments, "the outcomes considered good" align with "the greater good" but that is because the dilemmas were set up in such a way. It is not something that I think would hold true in general as I can imagine many types of dilemmas where the outcomes that people consider "good" might not necessarily align with "the greater good". For instance, if the dilemmas used would involve harm to one's family vs. greater harm to strangers, I could imagine that many people will focus on whatever decision-strategy yields minimal harm to one's family even if that is to the detriment of the greater good. Accordingly, the conclusion "When people experience the outcomes of their moral decisions, they learn to adopt strategies that benefit the greater good" seems overly broad to me.

Additionally on On page 20, the authors write:

"Second, a participant's "deontological" or "utilitarian" choices in moral dilemmas do not necessarily demonstrate that the participant is deontologist or utilitarian. Instead, those choices should be interpreted more cautiously as being consistent with following moral rules or CBR."

A relevant reference here might be: Conway, P., Goldstein-Greenwood, J., Polacek, D., & Greene, J. D. (2018). Sacrificial utilitarian judgments do reflect concern for the greater good: Clarification via process dissociation and the judgments of philosophers. *Cognition*, 179, 241-265. I actually think the authors' perspective aligns with the view of several prominent researchers in the field. Judgments can be said to align with utilitarians/deontology, however labelling people as "utilitarians" "deontologists" is a stretch.

Beyond this, I had one reflection. This don't really warrant any change to the manuscript. It doesn't even warrant a reply from the authors. It is just a reflection I wanted to share with the authors.

On page 19 they write:

"Across all experiments, the overwhelming majority of participants showed some form of learning from consequences (at least 95% show metacognitive or behavioral learning, see Table 2), suggesting that at most 5% of people are strictly deontological, in the sense that they would continue to base their decisions on moral rules even if the consequences of previously doing so had been predominantly bad."

I actually find the 5% number to be quite a high given what it represents (people that fully ignore outcomes when making moral decisions). It does make me wonder what the number on the opposite side would be: the percentage of people that are "pure utilitarians" and only weigh outcomes without weighing the value of actions themselves.

London, December 20th, 2024

Dear Dr. Horder,

We are happy to submit the revised version of our manuscript, titled “Metacognitive Learning from Consequences of Past Choices Shapes Moral Decision-Making” for publication in *Nature Human Behaviour*. We are grateful to you and the reviewers for the constructive suggestions for improvement, which we have all addressed.

The main changes to the manuscript are as follows:

- We have added more detail and rewrote some sections to clarify the paradigm, data analyses, and modelling throughout the manuscript to make the article more accessible.
- We have substantially extended the Discussion section by adding more implications of our results.
- To further highlight the connection of our article to extant discussions in moral psychology (beyond the more specific topic of metacognitive learning), we have also changed the title of the article to “Consequentialist Learning Shapes Reliance on Moral Rules versus Cost-Benefit Reasoning” (though, if there is any concern about the new title, we are open to keeping the old one).
- We have clarified the measure of model-based learning in the Results section of Experiment 2 and added more detail on its potential limitations to the Discussion.
- We have added some more explanation and interpretation of our results to the Discussion. Specifically, we discuss why we found a more consistent effect on participants’ choices than on their moral rightness judgments, and why we found a higher proportion of metacognitive learning in the CBR Success condition than in the Rule Success condition.

Note that the revised version of the manuscript exceeds the 5000 word limit as this was necessary to include the additional clarifications requested by the reviewers (we contacted you prior to submission to confirm that this is permitted). Below is a point-by-point response to all comments. Our response is in bold. For your convenience, we have highlighted changes in the manuscript in blue. Thank you and we look forward to your response.

Kind regards,
Maximilian Maier, Vanessa Cheung, and Falk Lieder

POINT-BY-POINT RESPONSE TO REVIEWER COMMENTS

Reviewer #1:

Review Manuscript 30421

R1.1) The authors report on 4 experiments that highlight the possibility that individuals consider the consequences of their past decisions when making moral decisions. Through these 4 experimental studies, the authors find that learning from the consequences of past decisions leads individuals to choose strategies that benefit the greater good. The authors also show that metacognitive moral learning involves generalization: its effects transfer from a classical utilitarian paradigm involving fictitious dilemmas to incentive-compatible, real-world donation decisions in a different context.

I am generally positive about this high-level research. For their study, the authors propose to mobilize clear and well-described computational models, one targeting model-free metacognitive learning and the other targeting model-based metacognitive learning. Although this is not my area of expertise, the authors describe these models precisely, which seems to me to capture the phenomena the authors are interested in.

I also appreciate the fact that the authors diversify their VDs, using particularly fictional moral dilemmas and exploring a possible transfer to more ecologically viable tasks (e.g. donation). The authors also follow good research practice: each study is carefully pre-registered and the material, data and scripts are made publicly available, always on an open access basis, in line with the principles of open science.

Thank you very much for the positive evaluation.

R1.1) In general, my only criticism would be a lack of clarity, especially for non-experts at several places. The technical reading is sometimes difficult to follow. Although the supplementary material provides a considerable amount of detail, I feel that the understanding should at least be clear without recourse to this supplementary material. I would highlight these few passages and advise the authors to work on simplifying them or providing additional explanations to make reading easier.

Thank you for pointing this out. This comment also reflects concerns raised by Reviewer 3. In the revised version of the manuscript, we have added additional clarification for all the specific sections that you highlighted (see response to the comments below). We also re-read the paper thoroughly to identify any sections that may be unclear. For more detail on additional changes, please also refer to R3.6, R3.7, & R3.9.

R1.2) The implications of the present results should be more fully developed. I very much appreciate that the authors do not over-interpret or over-generalize, but it seems to me that the conclusion drawn from the results should be given more prominence. What should we learn from these findings? What does it add to our understanding of morality? Do these results offer a break with what we previously knew (it seems to me that they do)? Finally, do these results teach us anything additional, beyond the realm of morality (again, it seems to me that they do)? I suggest that the authors develop these aspects.

Thank you for these helpful suggestions. We have added a discussion of the broader implications of our findings for understanding morality and judgment and decision-making more generally.

To highlight what our findings add to our understanding of human morality, we have added the following text in the Discussion section (p.19):

“These findings offer a new perspective on human morality that connects two fundamental debates: the debate about whether people make moral decisions based on (intuitive) moral rules (often equated with deontology) or cost-benefit reasoning (often equated with utilitarianism) (Greene, 2013; Haidt, 2001) and the debate about whether human morality is learned from experience (empiricism) versus innate (nativism) (Haidt & Joseph, 2004). Our findings suggest that the degree to which a person's moral decisions are driven by either utilitarian reasoning or intuitive moral rules depends partly on their learning history. Across all experiments, the overwhelming majority of participants showed some form of learning from consequences (at least 95% show metacognitive or behavioral learning, see Table 2), suggesting that at most 5% of people are strictly deontological, in the sense that they would continue to base their decisions on moral rules even if the consequences of previously doing so had been predominantly bad. Instead, many people even show metacognitive learning: They flexibly adapt the degree to which they rely on (intuitive) moral rules by learning from the consequences of previous decisions. This suggests that people's moral decisions are not inevitably controlled by potentially innate intuitions. Instead, we found that experience can teach people to update their decision-making strategies based on empirical observations. This supports the empiricist view that human morality is shaped by learning from experience. Consistent with this view, people might disagree about moral dilemmas partly because of differences in life experience. Some people may have experienced that blindly following the rules yields worse outcomes than occasionally overriding them by cost-benefit reasoning. Many others likely experienced that their attempts to outsmart the rules usually backfire. To the extent that moral disagreements are caused by learning from different experiences, we might be able to overcome our moral disagreements by sharing our experiences and learning from the experiences of others.”

In addition to these broader implications, we also added more detail on how our results challenge previous approaches that equate decision strategies (rules vs. CBR) with normative theories (utilitarianism vs. deontology) (pp. 19-20).

Our results also challenge previous approaches to moral psychology that equated normative theories of morality with decision strategies. As we argued in the introduction of this article, conceptually, deontology and utilitarianism are different from reliance on rules versus CBR, even though they are sometimes equated in the literature. The former are ethical theories that tell us what we should value, whereas the latter are decision strategies that can be used to achieve outcomes that are consistent with those values. In line with research that shows that simpler heuristics can lead to better consequences in certain environments, in the real world, relying on rules may sometimes lead to better consequences than CBR for various reasons (e.g., increased accuracy due to the bias-variance trade-off (Gigerenzer & Brighton, 2009; Brighton & Gigerenzer, 2012), lower cost of computation (Lieder & Griffiths, 2017), and increased trust in decision-makers using rules (Gigerenzer, 2008). Below, we discuss two findings showing that measures previously considered to measure reliance on different ethical theories may, in fact, measure reliance on different decision strategies.

First, existing self-report measures of deontology versus utilitarianism may actually measure reliance on the specific decision strategies of following rules versus CBR. Our results support this conclusion because (1) learning from consequences can increase people's scores on the Deontology Subscale of the Deontological-Consequentialist Scale (Mata et al., 2022) and decrease their scores on the Oxford Utilitarianism Scale Sacrificial Harm Subscale (Kahane et al., 2018) and (2) interpersonal differences in these scales were unrelated to evidence of metacognitive moral learning from the consequences of past decisions. If the DCS Deontology Subscale actually measured deontology, we would expect that participants who score higher would be less driven by outcomes and more by the intrinsic rightness of the action and, therefore, show less learning. Instead, we found that participants' scores on this scale were unrelated to how much they engaged in metacognitive moral learning from consequences. Moreover, learning from consequences changed participants' scores on the DCS Deontology Subscale. This suggests that it measures reliance on rules rather than on deontology, and that reliance on rules is more receptive to learning than previously thought, given that these scales are often considered to measure stable traits (Kahane et al., 2018). [Footnote: A similar argument may apply to the OUS Sacrificial Harm Subscale, although the case here is somewhat weaker because this scale claims to only measure one specific component of utilitarian psychology (sacrificial harm).]

Second, a participant's "deontological" or "utilitarian" choices in moral dilemmas do not necessarily demonstrate that the participant is deontologist or utilitarian.

Instead, those choices should be interpreted more cautiously as being consistent with following moral rules or CBR. If we had used participants' decisions in moral dilemmas to measure deontology and utilitarianism, we would have concluded that around 50% of people are deontologist (the proportion that chose the rule option in the first trial), rather than the much lower proportion of people that showed no learning from outcomes (around 5%). This article thereby adds to the existing literature challenging the use of sacrificial dilemmas to measure utilitarian versus deontological decision-making (Kahane 2015; Kahane et al., 2015), and offers an alternative interpretation of these choices in terms of decision strategies.

Further, we now also speculate about potential broader implications of our findings beyond the moral domain (p. 19):

“Although all of our experiments concerned moral decision-making, our finding that adaptive metacognitive learning from the consequences of past decisions shapes reliance on different decision strategies might also apply to judgment and decision-making more generally. Converging evidence for adaptive metacognitive learning in domains ranging from financial decision-making (Erev & Barron, 2005; Rieskamp & Otto, 2006; Lieder & Griffiths, 2017), and cognitive control (Lieder et al., 2018), and planning (Jain et al., 2019; He et al., 2023; Callaway et al., 2022), as well as problem-solving and mental arithmetic (Lieder & Griffiths, 2017) seem to support this generalization.”

R1.3) The difference between metacognitive learning computations and more traditional reinforcement learning models is difficult to understand. This is especially true for those new to the field. It seems to me that special attention should be paid to this, and the explanations in the main article should be restructured or expanded, as this is one of the keys to the research reported. The authors indicate “Put simply, the mechanisms of metacognitive moral learning differ from standard RL in two key ways: (1) learning occurs in the meta-control system whose “actions” are our decision strategies (e.g., CBR and rule-following), and (2) the reward signal is the person’s moral judgment of how good or bad their decision was. These moral judgments are partly based on consequences of the decision [40, 41]. Therefore, learning from the consequences of past decisions could, in principle, adaptively increase people’s reliance on decision strategies that produce good consequences, and decrease reliance on those that produce bad consequences.” Is it also possible to indicate the characteristics or features of standard RL to help make the understanding easier?

Thank you for alerting us of this issue. To address it, we now explain the characteristics of standard RL and how previous theories have applied it to moral learning in the first paragraph of Section 1.1. We then contrast our theory to it in the first two sentences of the following paragraph. Concretely, we have added the following text (p. 4):

“Prior work has identified several mechanisms of moral learning (Cushman et al., 2017). According to one of these mechanisms, reinforcement learning (RL), moral values are learned from the consequences of previous actions. Each time the consequences of an action are better than expected, the probability of repeating this action is increased, and each time the action’s consequences are worse than expected, the probability of repeating this action is reduced.

While prior theories of moral learning (Cushman, 2013; Crockett, 2013) proposed that people learn on the level of more specific behaviors (e.g., whether to punch someone), we propose that people also learn on the level of moral decision-making strategies (e.g., whether to engage in rules or CBR).”

We also added additional detail on the models for these types of learning in the computational modelling section. For more detail, please see R3.6 and R3.7.

R1.4) P7. The authors indicate « In line with this, we found a significant interaction between trial number and whether action coincided with rules or CBR on moral judgments, $b = -1.68$, $t(2340.73) = -2.20$, $p = .028$, two-sided. » The interaction effect should be clarified. Intuitively, we might think that morality differs more between rule-compliant action and CBR on moral judgments of the latter (compared to the former). Whether this is the case should be explained to facilitate understanding.

Thank you for pointing this out. The idea of the interaction test was that if people endorse CBR actions more throughout the experiment and oppose rule actions more, this would be consistent with increased reliance on CBR during the task. We now added additional detail in two places of the corresponding section to clarify this (changes in blue text, p. 9):

“Participants judged the moral rightness of the action under consideration on a scale from 0 (“Not at all morally right”) to 100 (“Completely morally right”) before making a decision. In some vignettes, this action is consistent with the CBR option, and in others, it is consistent with the rule option. We predicted that within each condition, additional experience would increase (decrease) the perceived moral rightness of actions that are consistent (inconsistent) with the rewarded decision strategy (i.e., following CBR vs. rules). [...] In other words, people became more likely to endorse CBR actions and to oppose rule actions during the experiment, which is in line with increased reliance on CBR as participants proceeded through the task.”

R1.5) P7. The authors indicate: However, the interaction between trial number and whether action coincided with rules or CBR was not statistically significant, $b = -1.45$,

$t(2239.64) = -1.91, p = .057$, two-sided.” This would therefore diminish the claim that it is learning from past events, but that individuals are directly considering the first consequence of their decision. Would we still call this learning? This could be discussed.

We agree that the effect on participants’ moral judgments has not been discussed in sufficient detail in the previous version of the manuscript. While we did not find a consistent interaction effect between trial number and framing for moral rightness judgments across the four experiments, we did find a consistent effect of trial number on participants’ choices in all experiments, which was our primary outcome of interest. The interaction effect between trial number and framing on participants’ moral rightness judgments were weaker, reaching statistical significance in Experiment 1 and the Rule Success condition of Experiments 3 and 4, but not in Experiment 2 and the CBR Success condition of Experiments 3 and 4. To ascertain whether we see an overall effect on moral rightness judgments, we therefore conducted an additional analysis of the effect on these judgments across all experiments and conditions. This indicates a significant interaction effect between trial number and framing on judgments, $F(1, 6144.66) = 16.17, p < .001$, and no statistically significant evidence that this effect is moderated by experiment ($F(3, 26708.45) = 0.85, p = .465$) or condition ($F(1, 26704.34) = 3.01, p = .083$).

Even though this analysis shows that we find an overall effect on judgments, it still raises the question of why this effect was weaker than the one on participants’ choices. One explanation based on prior research is that people’s expression of moral judgments is driven by different motives and considerations than their behavior. For instance, the research surveyed by Batson and Thompson (2001) suggests that people’s moral judgments are driven by their desire to appear moral whereas their choices are driven by considerations of the action’s costs and benefits. Given that social incentives strongly favor the expression of deontological convictions over utilitarian reasoning (Everett et al., 2018), one might therefore expect that people’s ratings are always biased toward deontological principles relative to how people actually make their moral decisions. This was (inadvertently) also encouraged by the phrasing of our question, which asks about the action itself without mentioning the consequences (e.g., “How morally right is it for you to push the man off the ladder”). Unlike judgments, choices have real consequences. Thus, it is plausible that participants’ actual choices are more strongly informed by the decision mechanisms that generate the best consequences, according to their experience. These psychological differences between judgment versus choice further decrease the degree to which one can expect people’s judgment responses to accurately reflect learning-induced changes in their moral decision-making. This might be why, in our experiments, learning had a stronger effect on participants’ moral decisions than on their ratings.

We have added a paragraph to the Discussion section to discuss the different results for ratings versus choices (p. 22):

“Our findings also raise the question of why we found a more consistent effect on participants' choices than their moral judgments: While we found a consistent effect for the choices across all four experiments, the effect on the moral judgments of the action under consideration was only significant in some of the experiments. Therefore, we conducted a cross-study analysis which showed evidence for an overall effect on judgments and no statistically significant evidence for moderation by Experiment or Condition (Rule Success vs. CBR Success; see Section S1.6). However, this still raises the question of why the effect on judgments was weaker than on choices. One possible explanation for the stronger effect of experience on choices is that when giving a judgment of moral rightness, (some) participants might have interpreted the question (“How morally right is it for you to [action under consideration]?”) as asking solely about the intrinsic rightness of the action regardless of its consequences in the specific situation. Prior research shows that these types of moral judgments often involve different considerations compared to decisions that would imply reduced learning. For instance, moral judgments tend to be driven more by reputational concerns (Batson and Thompson, 2011). Given that social incentives strongly favor the expression of deontological over utilitarian convictions (Everett et al., 2018), one might expect that people’s ratings are always biased toward deontological principles independent of the anticipated consequences. Future work could explore this by developing a scale that includes different questions, some of which are focused more on the action and others that are focused more on the outcomes, and validating those questions against behavioral measures.”

R1.6) P8. The authors indicate: « Compared to the CBR Success condition, participants in the Rule Success condition predicted that the rule-based action would be more likely to produce good outcomes, whereas the CBR-based action would be less likely to produce good outcomes, $F(1, 756) = 17.28, p < .001$. ». I'm not sure which IVs are included in the model here. **Are you reporting a main effect of an interaction?** Further clarification would be welcome (even if the methodology section is more fleshed out).

Thank you very much for highlighting this important passage. We agree that the description of this analysis was somewhat difficult to follow. To address this, we have extended the Results section with clear information about what the corresponding dependent variables are, and report the statistical analysis which was originally in the SI. We give further detail in the Method section.

Below is a revised version of the Results section (changes in blue text, pp. 11-12):

“We developed self-report measures for model-free and model-based learning in line with previous literature on these two types of RL in the moral domain (Cushman, 2013; Cushman et al., 2012; Crockett, 2013). Because model-based learning involves learning a probabilistic model of the anticipated outcomes of

actions, we used a measure that asked participants to rate the probabilities of good versus bad outcomes of choosing the CBR option, and of choosing the rule option.

In contrast, model-free learning involves assigning values intrinsically to actions rather than building a model of their possible consequences. Therefore, to measure model-free learning, we adapted a task from (Cushman et al., 2012) and asked participants to imagine carrying out typically harmful actions, which would not cause negative consequences in this specific instance (e.g., shooting a prop gun). If people show an aversion to these actions even though they cannot produce negative consequences, this would suggest assigning values intrinsically to actions (i.e., model-free learning).

Evidence for Model-Based Metacognitive Learning

We showed participants two new moral dilemmas (see Method section). In one dilemma, the action under consideration was the rule option ("rule action"), and in the other dilemma, the action was the CBR option ("CBR action"). For both dilemmas, participants predicted the probability of an overall good versus overall bad outcome for each action and each omission on a scale of 0 ("Bad outcome much more likely") to 100 ("Good outcome much more likely"). To assess the effect of learning, we calculated the probability of an action versus an omission leading to good consequences for all participants in both the Rule Action and CBR Action vignette (i.e., ΔM , or the $P(+|action) - P(+|omission)$ score).

Participants in the Rule Success condition rated rule actions to be more likely to lead to good outcomes relative to omissions ($\Delta M = 6.03$, 95% CI [0.41, 11.65]) than participants in the CBR Success condition ($\Delta M = -3.41$, 95% CI [-8.81, 1.99]). In contrast, participants in the CBR Success condition rated CBR actions to be more likely to lead to good outcomes relative to omissions ($\Delta M = 8.21$, 95% CI [2.76, 13.67]) than participants in the Rule Success condition ($\Delta M = -5.47$, 95% CI [-10.94, -0.01]). This interaction was significant, $F(1, 756) = 17.28$, $p < .001$ (see Figure~ref{fig:MB_results} in the SI) .

In other words, participants in the CBR Success condition were more positive about the expected outcomes of CBR actions (i.e., they thought engaging in them would lead to better consequences relative to not doing anything) compared to rule actions, while this pattern was reversed in the Rule Success condition. This suggests that (some) participants learned the conditional probabilities of good versus bad outcomes, given that the decision is reached using CBR or rules, a mechanism we refer to as model-based metacognitive learning.

No Evidence for Model-Free Metacognitive Learning

By contrast, the experimental manipulation had no significant effect on people's emotional reactions to violations of the moral rule to do no harm ($t(377.08) = 0.62$,

p=.269, d = 0.06, one-sided), which would have been evidence for model-free learning (Cushman et al., 2012; see Section S1.2.4).

For more details on the methodology, analytic approach, and results for additional self-report measures, see Methods and Section S1.2.”

We provide a detailed description of our measures of model-based and model-free learning in the Method section (p. 29):

“To measure model-based learning, we showed participants two vignettes after the main task (developed by Cheung et al., 2024), one with a “CBR Action” and one with a “Rule Action” choice framing (i.e., either the CBR option or the rule option was framed as the action under consideration). Each participant saw one CBR Action and one Rule Action vignette from a set of six possible vignettes. Then, we showed participants descriptions of one good and one bad outcome for each of the two choice options (i.e., action vs. omission). After reading the potential outcomes of either action, participants were asked to estimate the probability of the good outcome and the probability of the bad outcome.

As an example, participants first read a dilemma about whether to proceed with an attack on an enemy base at the cost of potentially sacrificing the lives of nearby civilians (CBR Action). As shown below, they were then asked to estimate the probability that proceeding with the attack would have good versus bad outcomes overall.

Assume you proceed with the attack. Which of the following outcomes is more likely? (0 = “Bad outcome much more likely”, 50 = “Good and bad outcomes equally likely”, 100 = “Good outcome much more likely”)

- 1. You proceed with the attack, and this leads to a good outcome overall. The attack weakens the enemy’s capabilities, but it also kills a group of innocent civilians living in the vicinity. The war ends sooner than expected, saving many more innocent people who would have otherwise continued to suffer because of the ongoing war. In the end, your actions helped save these people—but at the cost of some civilian lives.*
- 2. You proceed with the attack, and this leads to a bad outcome overall. The enemy forces are strong and the outpost is well-defended, so it only suffered insignificant damages. As a result of the attack, a group of nearby civilians die as collateral damage. In the end, your actions do not save anyone: a group of innocent civilians die and the war continues.*

This was followed by the same question for the outcomes of not proceeding with the action. This question and its outcomes, as well as full materials for the other vignettes, can be found in the online repository.”

R1.7) P15. The authors suggest - and I would agree with them - that « everyone might be a consequentialist learner, regardless of which moral principles they endorse. » This is particularly interesting in that the research proposed by the authors and the results seem to provide an 'impactful' answer to the need to stop considering a utilitarian response as consequentialist and aiming at the greater good, and to consider a deontic response which does not have this aim'. This could be suggested, if the authors agree.

Thank you for the suggestion. We agree that the utilitarian response (which we refer to as the CBR option) would not necessarily lead to the greater good in real life. In the Discussion, we mentioned that our findings suggest that moral learning from consequences aligns people's decision-making with global consequentialism.

We agree that this is an interesting implication of our article that may have not been emphasized sufficiently in the previous version. To highlight the connection to this important question we also changed the title of the article to

“Consequentialist Learning Shapes Reliance on Moral Rules versus Cost-Benefit Reasoning”

We made a new addition to this section to highlight the implications you point out more strongly (changes in blue text, pp. 18-19):

“Even though moral learning was driven by consequences, it did not always direct people toward making their decisions by reasoning about consequences (CBR). Instead, learning from consequences increased reliance on moral rules when following them had previously led to better consequences. In other words, it is possible that people who prioritize moral rules do so in order to bring about good consequences, even though they may not necessarily be explicitly reasoning about this. Overall, our findings suggest that moral learning from consequences aligns people's decision-making with global consequentialism (Ord, 2009), according to which one should use whatever rule-based, reasoning-based, or virtue-based decision mechanism yields the best consequences. This suggests that, ironically, some people who insist on following moral rules regardless of the consequences may have reached this conviction by learning from consequences. In this sense, everyone might be a consequentialist learner, regardless of which moral principles they endorse.”

R1.8) I wish the authors the best of luck with their very high-quality research, which has the potential to offer a significant advance in the field of moral judgement and possibly much more.

Thank you again for your comments and for reviewing our manuscript.

Reviewer #2

R2.1) In this paper, Maier and colleagues present a series of experimental studies, augmented by computational modeling and personality surveys, which demonstrate that humans engage in metacognitive moral strategy learning. In particular, subjects came to adopt cost-benefit reasoning when trained on scenarios where this form of reasoning is beneficial. They then generalized this strategy to completely different contexts, and the generalization lasted at least a few hours.

Overall, I thought this was a really nice paper: clearly written, computationally sophisticated, and potentially impactful. I was particularly impressed by the generalization results. My comments below are relatively minor.

Thank you for the positive evaluation.

R2.2) I want to raise two high-level issues. One is that they only saw evidence for metacognitive moral learning in the CBR Success condition, not in the Rule Success condition. In the Discussion, the authors pose this as raising a question about "which experiences and situational factors trigger versus inhibit metacognitive moral learning." My concern is that this is essentially post-hoc. An alternative is that the phenomenon is simply not robust. The fact that they see the effect across several experiments, and that it generalizes beyond the experimental paradigm, is reassuring, but it does seem to be a problem for the theory, which lacks any predictions about "triggering" conditions.

Thank you for raising this important point. We agree that the lower proportion of metacognitive learners in the Rule Success condition compared to the CBR Success condition is a surprising finding. However, this doesn't mean that our finding of metacognitive moral learning from the consequences of past decisions isn't robust. To the contrary, across four experiments, we consistently find that about 30-40% of participants engage in metacognitive learning in the Rule Success condition, showing that it is in fact robust. The proportion of participants showing metacognitive moral learning is somewhat lower than in the CBR Success condition (usually about 60%). However, this is only a quantitative difference in the prevalence of metacognitive learning rather than a qualitative difference in the existence of this phenomenon. Therefore, it seems more plausible that these differences stem from factors specific to the two conditions.

Nevertheless, we agree that the lower proportion of participants that show metacognitive learning in the Rule Success condition as compared to the CBR Success condition merits more explanation and exploration. To investigate this question, we have conducted a series of exploratory analyses. These analyses screened the two experimental conditions for differences that might increase or decrease metacognitive

learning. We identified several factors that predict increased versus decreased metacognitive moral learning, including participants' self-report ratings of taking the moral dilemmas seriously, the perceived plausibility of the outcomes, the emotional experience of the outcomes, the perceived informativeness of the outcomes, their relevance to the real world, and the perceived utility of learning (all $p \leq .015$). However, we also found that none of those factors differed significantly between the CBR Success condition and the Rule Success condition (all $p \geq .135$). We report the results of these exploratory analyses in Section S1.4.4 of the Supplementary Information.

In principle, less learning in the Rule Success condition could have occurred because the majority of our participants already relied on rules in the first dilemma. However, we consistently found that for the first dilemma, around half of participants chose the rule option and the other half chose the CBR option in both conditions with only a slight bias towards CBR (Experiment 1: across both conditions, 50% chose the CBR option; Experiment 2: 55%; Experiment 3: 55%; Experiment 4: 54%). Therefore, it seems unlikely that a strong prior belief in the effectiveness of rules lead to larger updating in this condition.

These results suggest that it is unlikely that the difference between the amount of systematic change between the two conditions results from accidental differences between the two conditions. We speculate that, instead, the difference between the two conditions reflects an inherent difference between CBR versus following moral rules: while there is only a single CBR strategy, there are a vast number of different rules one could learn to follow. Our paradigm reflected this reality: the pertinent moral rule(s) differed across the 11 dilemmas. In the CBR Success condition, learning was easier in the sense that participants only had to learn about the surprisingly high effectiveness of CBR. By contrast, in the Rule Success condition, learning about the surprisingly low effectiveness of CBR was not enough. Participants also had to learn which moral rules, values, intuitions, or virtues to rely on instead. In this context, metacognitive learning could either guide participants to rely more on rules in general, or guide them to rely more on specific rules (e.g., do not kill). It is likely that some participants in our task learn about the effectiveness of relying on rules in general (as reflected by the high proportion of metacognitive learners and the fact that we see learning effects even on those scenarios with a rule that is not shared with other scenarios) whereas other participants may learn only about specific rules. Because the pertinent rule differed across dilemmas, participants engaging in metacognitive learning about specific rules observe less evidence for the effectiveness of any one rule. Moreover, even when those participants learned to rely more on one of the successful rules, this learning wouldn't necessarily show in subsequent dilemmas where the pertinent moral rules were different. Therefore, to the extent that some participants learn about the effectiveness of specific rules rather than the general effectiveness of rules, we would expect a smaller proportion of metacognitive learning in the Rule Success condition.

In any case, the main contribution of our paper was to demonstrate that metacognitive learning shapes moral decision making. Conducting a more detailed investigation and explanation of moderating conditions of metacognitive moral learning is an interesting direction for future work.

We added a discussion about the difference between the two experimental conditions in the Discussion section as follows (p. 21):

“Our finding that metacognitive learning was consistently more prevalent in the CBR Success condition than in the Rule Success condition raises the question of which experiences and situational factors trigger versus inhibit metacognitive moral learning. We investigated this question through a series of exploratory analyses reported in Section S1.4.4. These analyses identified several factors that predict increased versus decreased metacognitive moral learning, including taking the moral dilemmas seriously, the perceived plausibility of the outcomes, the emotional experience of the outcomes, the perceived informativeness of the outcomes and their relevance to the real world, and the perceived utility of learning. However, we also found that none of those factors differed significantly between the CBR Success and Rule Success conditions.

In principle, less learning in the Rule Success condition could have occurred because the majority of our participants already relied on rules in the first dilemma. However, we consistently found that for the first dilemma, around half of the participants chose the rule option, and the other half chose the CBR option in both conditions (Experiment 1: across both conditions, 50% chose the CBR option; Experiment 2: 55%; Experiment 3: 55%; Experiment 4: 54%). Therefore, it seems unlikely that they had a stronger prior that one option would work better over the other.

These results suggest that it is unlikely that the difference between the amount of systematic change between the two conditions results from unintentional differences between conditions in the paradigm. Instead, this difference might reflect an inherent difference between CBR versus following moral rules: while there is only a single CBR strategy, there are a vast number of different rules one could learn to follow. Our paradigm reflected this reality: the pertinent moral rule(s) differed across the 11 dilemmas. In the CBR Success condition, learning was easier because participants only had to learn about the high effectiveness of CBR, and the decision strategy of CBR is more generally applicable in different contexts than any particular moral rule. By contrast, in the Rule Success condition, metacognitive learning could either guide participants to rely more on rules in general (i.e., relying on rules leads to better outcomes), or to rely more on specific rules (e.g., “tell the truth” and “do not kill”). In our experiments, the pertinent rule differed across dilemmas. Therefore, participants who learned about specific rules observed less evidence for the effectiveness of any one rule.

Moreover, even when those participants learned to rely more on one of the successful rules, this learning did not necessarily show in subsequent dilemmas where the pertinent moral rules were different. Future research could test this interpretation by conducting experiments in which a salient moral rule is held constant.”

R2.3) The other high-level issue is the characterization of success vs. failure in the vignettes. It seems to me that this takes as given that subjects have a particular utility function for the outcomes of moral dilemmas. But I thought (and here I may be betraying my ignorance of moral philosophy) the different moral frameworks (utilitarianism, consequentialism, etc.) essentially define different utility functions. Is it really possible to have a framework-agnostic utility function for learning at the metacognitive level? I understand that in the model implementation, you're actually using the subjective ratings, but this just pushes the question back further.

Thank you for raising this important point. We agree that there are some difficulties with a framework-agnostic utility function. One difficulty is comparing different moral theories that assign utility to consequences but do so differently. For instance, a preference consequentialist and a utilitarian would assign very different utility to a situation in which someone's (irrational) preference was met, but they caused some harm to themselves as a consequence of this decision. We believe that this distinction can be overcome to some extent by using the value ratings that participants assign to the outcomes, especially considering that the predictions from our theory and the models do not depend much on the magnitude of the rating, but mostly on whether the outcome was rated as overall good or bad (i.e., whether they were positive or negative values on the slider). We believe it is a reasonable assumption that both a preference consequentialist and a utilitarian (or someone using any other moral decision-making framework that considers consequences) would be less likely to repeat an action if it leads to (in their view) worse consequences than expected, and more likely to repeat an action if it leads to better consequences than expected. Further, since we are not concerned with, for instance, consensus processes between the different frameworks used by different participants, but intra-individual learning, our framework does not require a utility function that can convert utilities between different people and their different ethical theories.

A more fundamental problem arises when comparing moral frameworks that assign utility to outcomes to those that assign utility to *actions*. In our theory, we accommodate the possibility of action-based views with a link directly from the action to the moral judgement (skipping the outcomes, Figure 1). In other words, while people who strictly follow deontology would still perceive the outcome of their actions as worse than expected, this would not result in a decreased probability of engaging in that action in the future, as only the action itself is relevant for strict deontologists and not the outcome. In our paradigm, we would, therefore, not expect learning for people solely relying on deontology. Consistent with this, we find that a small proportion of

participants (~5%) are best explained by the no-learning models across the three experiments. However, the fact that this proportion is so small suggests that many people who would be classified as deontologists with standard methods (i.e., methods that map the decision strategies of rules vs. CBR on the meta-ethical theories of consequentialism vs. deontology) are actually better described by rule utilitarianism or global consequentialism than deontology (i.e., they knowingly or unknowingly rely on rules because they lead to better consequences).

We added a paragraph about these issues to the Discussion section (p. 21):

“Our theory also raises the question of how to compare the learning signals from different ethical theories and whether it is possible to have a utility function that is agnostic about which theories people use. Our paradigm is able to capture learning broadly for ethical theories that take consequences into account. This is because we classified the outcomes in our paradigm as good or bad based on participants’ own evaluations and also use these evaluations for our computational models. This sidesteps the question of how people determine what is a morally good or bad outcome. [Footnote: The modeling approach we used does not require any integration between the utility functions of different moral theories because we only model intra-individual learning based on a participant’s own utility function (rather than trade-offs between the utility functions of different participants).] As for deontological theories, the intrinsic rightness of certain actions determines their goodness/badness regardless of their outcomes. Therefore, we would not expect that people who strictly follow deontology would learn in our paradigm, as they would not learn from outcomes. In line with this, our theory explicitly acknowledges that moral evaluations also depend on other factors, such as moral intuitions about the chosen action itself (see Figure 1, particularly the arrow going directly from decision to moral evaluation). Consistent with these assumptions, we did indeed find that a small proportion of participants did not learn from the consequences of their decisions in our task.”

Minor comments:

R2.4) I think the paper would benefit from some schematic diagrams of the experimental tasks, so that readers can get a better idea of what exactly the participants are seeing/doing.

Thank you for this suggestion. We have added a diagram to describe the experimental paradigm in the section “A New Experimental Paradigm Using Realistic Trolley-Type Dilemmas with Outcomes” of the Results section, where we first introduce the paradigm (p. 8).

We also added a few lines in the figure caption about what the task looks like from the participant's perspective (p. 8):

"Participants are randomized into one of two conditions: "Rule Success" and "CBR Success". The experimental condition determines whether choosing the CBR option or the rule option leads to good or bad outcomes. In some vignettes, the action under consideration is the CBR option (e.g., "Do you push the man?"), and in other vignettes, it is the rule option (e.g., "Do you quit your job?"). This means that the "yes" and "no" responses do not always correspond to the same decision strategy (rules vs. CBR) even in the same experimental condition.

Because of this, from the participant's perspective, choosing “yes” would sometimes lead to good outcomes and sometimes lead to bad outcomes. More details about the paradigm can be found in the Method section.”

R2.5) I'm confused about why some tests are two-sided and some are one-sided.

Thank you for pointing this out. We now clarify the approach that we took to decide between one-sided and two-sided tests in the main text of the manuscript (p. 9).

“Throughout this article, we use one-sided tests only when we preregistered a one-sided prediction. We use two-sided test either when we did not preregister a direction (mostly for interaction tests), or for tests that were not preregistered.”

R2.6) Fig 4: "Metacognitive" -> "Metacognitive"

Thank you for noticing this typo. We updated the axis label accordingly.

R2.7) Somewhere there needs to be an explanation of how the Bayes Factors were obtained for Fig 4 and Fig 5. I think this comes from the Bayesian model selection described in the methods, but this isn't explicit.

Thank you for pointing this out. Upon checking the manuscript, we noticed that the details on the inclusion Bayes factor calculation were only provided in the Supplementary Methods, and we agree that they should be in the main Method section. We moved the following text to the Method section of the manuscript (p. 32):

“To test moderation effects and the effects for those participants who show evidence for metacognitive learning, we first calculate the evidence for metacognitive learning for each participant. We do this by calculating an inclusion Bayes factor comparing the posterior odds of models that describe strategy learning (MF-M and MB-M) with models that describe behavioral learning from action/omission (MF-B and MB-B) or no learning (C-M and C-B). The code for calculating the inclusion Bayes factors is available in the online repository and the preregistration. We estimated marginal likelihoods using the bridgesampling package (Gronau et al., 2020a). For more information on inclusion Bayes factors, see Hinne et al. (2013), Maier et al. (2023, 2024).

To identify transfer effects only for metacognitive learners, we test the effect of condition in a model that uses evidence for metacognitive learning (in terms of an inclusion Bayes factor contrasting MB-M and MF-M with the other four models) as a covariate. We then recenter the inclusion Bayes factor covariate so that 0 indicates strong evidence for metacognitive learning. This allows us to test the

effect of the experimental condition for those participants who showed evidence for metacognitive learning.”

R2.8) There are a few places where p-values are reported without specifying what the statistical test was.

Thank you for pointing this out. It seems that this mainly occurred in the Results section of Experiment 3, which had been kept very brief as it was a replication of Experiment 2. We had included the full test statistics in the SI, but after considering your comment, we have now added them to the main text. We also added to the motivation for Experiment 3 for clarity. The section now reads (changes in blue text, p. 14):

“Experiment 3 provides a well-powered, preregistered ([url{https://osf.io/7guj6}](https://osf.io/7guj6)) replication and extension of these findings with two additional real-world donation decisions (see Method section) and twice as many participants (N = 834). All materials were identical to those from Experiment 2, except that we removed the self-report measures of metacognitive learning.

We found that the proportions of participants best explained by each model (see Table~ref{tab:prop}) and each type of learning mechanism (see Table~ref{table:meta_vs_behav}) were similar to those in Experiment~2. As predicted, metacognitive learners exhibited strong evidence of transfer to self-report measures of moral convictions (OUS Sacrificial Harm Subscale, $b = 0.84$, $t(830) = 8.39$, $p < .001$; DCS Deontology Subscale, $b = -0.79$, $t(830) = 5.67$, $p < .001$) and an overall main effect of the outcomes in the moral learning paradigm on the three incentive-compatible, real-world donation decisions ($b = 9.15$, $t(830) = 5.20$, $p < .001$).

When averaging across all participants, we found evidence for transfer to people's moral convictions (OUS Sacrificial Harm Subscale, $t(829.06) = 7.50$, $p < .001$, $d = 0.52$, one-sided; DCS Deontology Subscale, $t(825.3) = 2.81$, $p = .003$, $d = 0.20$, one-sided) but were unable to detect the effect on donations ($t(832) = 1.55$, $p = .061$, one-sided). This discrepancy underscores the importance of metacognitive learning for transfer. We report the full results, including effects on individual donation decisions, in Section S1.3.”

We also noticed an instance of this for one p-value in Study 2, which we revised as follows (changes in blue text, p. 12):

“The experimental manipulation had no significant effect on people's emotional reactions to violations of the moral rule to do no harm ($t(377.08) = 0.62$, $p = .269$, $d = 0.06$, one-sided), which would have been evidence for model-free learning (Cushman et al., 2012; see Section S1.2).”

Reviewer #3

R3.1) The current study investigates how people learn to adjust their moral decision-making based on the consequences of their past choices. Across four experiments, the authors demonstrate that individuals adaptively modify their reliance on moral rules versus cost-benefit reasoning (CBR) in response to experienced outcomes. The researchers show that when CBR leads to better consequences, participants gravitate towards those types of choices, whereas if the inverse is true, the inverse happens. Interestingly, they find that these effects transfer to other moral judgments such as real-world donation decisions.

This work exhibits several strengths. The authors introduce a novel moral learning paradigm. The transparency of the work is commendable, with the authors reporting all procedures, analyses, and results in detail. They preregistered all experiments and these appear comprehensive and align with the reported analyses. As far as I can tell, they had sufficient statistical power to detect the reported effects. The core findings are replicated across studies. The inclusion of the incentive-compatible donation decision, adds ecological validity to the study.

Thank you very much for the positive evaluation.

R3.2) All in all, I think this is an interesting piece of work. I will however admit that it took me a couple of reads to understand the manuscript sufficiently for me to be able to provide a review. Part of that is undoubtedly caused because I have never done any computational modeling, but still, my primary recommendation to the authors would be to revise the manuscript with non-technical readers in mind as I do think that several parts of the manuscript could benefit from making it a tad more accessible. I believe that doing so will help them reach a wider audience.

Thank you for raising this important point, which also echoes concerns raised by Reviewer 1. We have addressed each of the specific points outlined below and also reread the manuscript carefully to add more detail in various places that may have been unclear or too short in the previous version. For more details on additional steps that we took, please see our responses to Reviewer 1 (R1.1 – R1.4, R.1.6).

Below, I list some considerations that occurred to me while reading through the manuscript:

R3.3) One issue I was wondering about while reading the manuscript had to do with the author's framing of "Deontological" decisions as "rule-based" decisions. How to best label decisions in sacrificial dilemmas has been a point of contention and much of that debate has focused on whether "utilitarian" decisions should really be labeled "utilitarian". I appreciate that the authors steer clear of that debate by framing those decisions as "CBR-based". That seems like a fairly neutral descriptor that avoids much of the baggage associated with a term like "utilitarian".

I am however not entirely convinced whether "deontological" decisions are best labeled as "rule-based" decisions either. I realize that type of framing is not uncommon, but to the best of my knowledge, the empirical support for such a view is still somewhat sparse. Maybe the authors are aware of some convincing evidence in that regard, but I am not sure much evidence in that regard exists. In any case, I think it would be good if the authors could provide some more (empirical) support for labeling those decisions as such.

As an alternative, I was wondering whether the authors have considered conceptualizing **non-CBR decisions** as resulting from "an evaluation of the moral value of an action itself". In my mind, that is a bit more neutral, still fits clearly with the idea of "deontological" (as in the idea that actions have inherent moral value), and it contrasts nicely with CBR decisions as those result from an evaluation of outcomes. I also think this still aligns with the conceptualization suggested by Cushman in his work on model-free / model-based moral cognition. I don't intend this remark as a point of contention, it is certainly possible the authors considered such a framing and decided that the current "rule-based" framing fits better with the operationalization used in their paradigm but I was wondering whether the authors have reflected on that issue and decided against that alternative.

Thank you for the suggestion and we agree with this concern. When we refer to the "rule-based options" (i.e., non-CBR decisions), we do not necessarily mean that people use a cognitive process of thinking about a rule, just that the choice is aligned or consistent with a moral rule (or does not violate a moral rule). We considered your proposed solution ("conceptualizing non-CBR decisions as resulting from an evaluation of the moral value of an action itself"), but this also maps a decision process (evaluating the moral value of the action) on a choice, which we wish to avoid.

Therefore, we have changed the names of the two choices to the '*CBR option*' and '*rule option*' instead of 'CBR-based' and 'rule-based' options. We have gone through the manuscript and made these changes where appropriate. We added more clarification in the Introduction (p. 3):

"Previous research has conflated these decision mechanisms with the ethical theories of deontology and consequentialism by construing moral dilemmas as decisions between a utilitarian option and a deontological option. To avoid this conflation, we will analyze moral dilemmas as choices between an option that is consistent with a moral rule ("rule option") and an option that is inconsistent with that rule but appears preferable according to CBR ("CBR option"). Though we use these terms, it does not imply that people necessarily explicitly consider CBR or rules during the decision process. Some participants might, for instance, choose the rule option because of moral values or emotional reactions acquired through experiential learning."

This hopefully mitigates the issue of mapping a decision process to a choice, as we do not want to connect the choice options directly to an assumption about what decision processes people are actually engaging in.

R3.4) While I like the paradigm the authors have used, and appreciate this novel approach to studying moral judgment, the authors decided to implement somewhat extreme learning conditions where either CBR or rule-based decisions always lead to good outcomes. I can see the methodological benefits, but it does make one wonder to what extent all of the findings represent learning or are the result of task demands. How would the authors counter such a charge? I will say I believe Study 4 is certainly helpful in that regard, but I am not entirely convinced it fully halts such a charge. I would suggest a discussion of that issue might benefit the paper.

Relatedly, I was also wondering whether the authors considered using a more nuanced approach, such as having CBR or rule-based decisions lead to positive outcomes in, say, 80% of cases. I think it might be valuable for the authors to discuss whether they considered less extreme paradigms and to reflect on how such approaches might affect the observed learning processes.

Thank you for highlighting this uncertainty. Regarding your concern that the current implementation increases demand characteristics in comparison to a more probabilistic approach, we do not think this is very plausible as (1) Experiment 4, which is a two-part study where participants thought was implemented by two different experimenters, provides strong evidence against demand characteristics driving the effect, and (2) the dissociation of action vs. inaction and rules vs. CBR already makes the manipulation much less transparent to the participants than reading the manuscript might suggest. From the participant's perspective, answering "yes" would sometimes lead to good outcomes and sometimes lead to bad outcomes.

To more directly address your concern, we have recently developed a new moral learning paradigm with probabilistic outcomes. When we used this paradigm to reduce the strength of the outcome contingency from 100% to only 70%, we still observed evidence for metacognitive moral learning from the outcomes of previous decisions. Below we discuss the rationale for this new experimental paradigm, the follow-up study, and the preliminary results we obtained in this follow-up study.

When using probabilistic contingencies, according to our theory, we would expect to retain the qualitative effects as long as the rewarded decision strategy leads to good consequences more than 50% of the time. However, in a probabilistic setting, the learning signal is weaker than in the deterministic setting that we chose. Compared to standard reinforcement learning tasks, the paradigm used in Experiments 1-4 has very few trials (i.e., only 13 dilemmas). With this paradigm, it is difficult to present more trials within a reasonable time because each dilemma takes a significant amount of time to

read. This is why we used deterministic outcomes to create the most powerful learning signal possible. Retaining comparable statistical power with a probabilistic reward signal would require us to either further increase the sample size (which would be expensive given that Experiments 3 and 4 already cost around 8000 each), or increase the number of dilemmas, which would make the study even longer for participants (and also increase the cost).

One way to mitigate this issue is to design a paradigm which only uses picture-based dilemmas rather than text-based vignettes, which allows us to increase the number of trials and therefore implement probabilistic contingencies between choosing the CBR/rule option and the resulting outcomes. We have run such a follow-up experiment (Tahmasebi et al., 2024). Preliminary results of this follow-up experiment suggest that metacognitive moral learning from the consequences of past decisions also occurs when the CBR/rule option leads to better outcomes only 70% of the time. In line with research on intermittent reinforcement, we would expect that this type of schedule would actually lead to stronger behaviour maintenance beyond the task (assuming that one reinforces to the same level of response probabilities). We have incorporated this finding into the Discussion section (p. 23):

“Fourth, future research could explore the effects of variations in the reinforcement schedule. In the current experiment, we focused on a simple reinforcement schedule, where the rule/CBR options always/never lead to success. The reasons for this choice were mostly pragmatic: our task has relatively few trials compared to other reinforcement learning tasks, and it is difficult to increase the number of trials much more without making the task too long. Therefore, the current schedule achieves the strongest learning signal, given the small number of trials in our experiment. One straightforward modification is to introduce probabilistic rewards (e.g., CBR leads to success 80% rather than 100% of the time). Our preliminary results from a new moral learning paradigm with probabilistic outcomes suggest that metacognitive moral learning from the consequences of past decisions likely also occurs when the CBR/rule option leads to better outcomes only 70% of the time (Tahmasebi et al., 2024). In line with research on intermittent conditioning (Skinner, 1957), probabilistic reinforcement may lead to stronger behavior maintenance once a given level of behavior is reached and would, therefore, also be valuable for future work aimed at probing the temporal stability of moral learning.”

R3.5) On pages 5-6, the authors write: "Our experimental paradigm varied all of these salient features independently of which option each strategy recommended (see Methods section for more details). Therefore, it is not immediately obvious to participants what is being reinforced in our paradigm. This was also reflected in their responses to an open question in Experiment 1 (see Section S1.1)"

I actually think this is a very important bit of information. I think it might be good to include it a bit more explicitly in the results section because I think it gives more credibility to their findings. As a reader, the manipulation the authors used seemed a bit like a sledgehammer, but if participants don't feel it as such that counters some of my worries.

Thank you for the suggestion. We address a similar point in R2.4, where we have added in the new figure caption a description of what the task looks like from the participants' perspective. In the caption, we explain in more detail why the manipulation may not be immediately obvious to participants: *"In some vignettes, the action under consideration is the CBR option (e.g., "Do you push the man?"), and in other vignettes, it is the rule option (e.g., "Do you quit your job?"). This means that the "yes" and "no" responses do not always correspond with the same decision strategy (rules vs. CBR) even in the same experimental condition. Because of this, from the participant's perspective, choosing "yes" would sometimes lead to good outcomes and sometimes lead to bad outcomes."*

We also included some responses to an open question about the reward contingency that participants answered at the end of Experiment 1, where we asked them whether they used information about the outcomes of their choices when making decisions throughout the experiment, and if so, how. Most participants who said yes appeared to be unaware of the specific manipulation we used. We added this in a footnote in the section about the paradigm (in blue text, p. 7):

"Most participants are not trained in moral philosophy, meaning that the abstract moral theories of deontology and utilitarianism are likely less salient for them than the concrete choices between action versus omission, the specific behaviors, and the specific moral rules that recommend or oppose them. Our experimental paradigm varied all of these salient features independently of which option each strategy recommended (see Figure 1 and the Methods section for more details). Therefore, it is not immediately obvious to participants what is being reinforced in our paradigm. This was also reflected in their responses to an open question in Experiment 1. [Footnote: At the end of the study, we asked participants whether they "used information about the outcomes of [their] choices when making decisions throughout the experiment," and, if so, how. Of those participants who reported taking outcomes into account, most appeared to be unaware of the specific manipulation (e.g., "Yes I tried to worry more about the initial moral decision and less on the outcomes as it was clear the outcome could vary/was more unpredictable," and "I tried to anticipate what the likely outcome would be, but I wasn't right"; full responses are available in the online repository).]"

R3.6) Generally speaking, I think the authors' introduction of the different types of learning could benefit from a slightly more accessible explanation. I am familiar with the key papers on this issue but I still find the distinction between model-based and model-free learning hard to interpret and the authors could do more to help readers unfamiliar with this to understand the

key points. For instance, the authors state that model-based learning results in "two beta distributions: one for the consequences of CBR and one for the consequences of rule-following," while model-free learning "assigns intrinsic moral values to using different decision strategies." An explanation of what these outcomes mean in practical terms and how they differ functionally would enhance the paper's accessibility.

Thank you very much for raising this important point. We completely agree that the previous version was too condensed. We have added additional detail to address this. The new computational modelling section now reads as follows (with the new text in blue text). Concretely, the description of our model of model-based metacognitive moral learning now reads (changes in blue text, pp. 5-6):

“Model-based learning uses an explicit model of the world to estimate the conditional probabilities of different outcomes (Doll et al., 2012; Sutton et al., 2018, p.159). We modeled model-based metacognitive moral learning as Bayesian learning of the conditional probabilities of good versus bad outcomes of decisions made by CBR versus following moral rules (e.g., $P(\text{good outcome} | \text{CBR})$). This model learns these probabilities by updating the parameters of two beta distributions: one for the probability that CBR will yield a good outcome and one for the probability that following rules will yield a good outcome. The probability of a bad outcome is simply one minus the probability of a good outcome. In other words, this model estimates the likelihoods of four different outcomes: following CBR leads to good versus bad outcomes, and following rules leads to good versus bad outcomes.”

And the description of our model of model-free metacognitive moral learning now reads (p.6):

“Model-free learning assigns values to actions that are independent of the outcomes those actions will yield in the current situation. Those intrinsic values are based on the average reward each action produced in the past. To model model-free metacognitive moral learning, we adapted the most common model of model-free RL: Q-learning (Watkins, 1989; Watkins & Dayan, 1992; Dearden et al., 1998). Our model assigns values directly to using moral decision-making strategies (i.e., CBR vs. following moral rules); those values are represented as Q-values. After each decision, the model updates the Q-value of relying on the decision strategy that produced that decision. This update is proportional to the experienced moral prediction error. The moral prediction error is the difference between the decision-maker's judgment of how morally right or wrong the decision was and the current Q-value of the decision strategy that produced it. The higher the Q-value assigned to a decision strategy, the more likely the model is to rely on it.”

Moreover, we have added a paragraph that explicitly highlights the essential difference between these models (p. 6):

“Unlike the model-based beta-binomial model, the model-free Q-learning model does not learn about the probabilities of the four different outcomes, but instead learns two Q-values: one for CBR and one for rule following. Therefore, our two computational models capture the key distinction between model-based and model-free learning: model-based learning involves learning about the probabilities of the different outcomes of an action, whereas model-free learning assigns a value to the action itself.”

R3.7) Relatedly, the paper would benefit from a non-technical explanation of the difference between "metacognitive learning" and "behavioral learning" and what the practical difference between these different learning types reflect. The difference is never really explained (perhaps the authors consider it self-explanatory) but **I must admit to being somewhat confused by the sudden introduction of "behavioral learning" with the comparison between metacognitive and behavioral learning on page 9**. Based on the method sections, I ended up concluding that metacognitive learning reflects participants' increasing reliance on either CBR or Rule-based moral cognition and that behavioral reflects participants learning that either "Actions" or "Omissions" are good/bad, though I am not entirely certain of this interpretation.

Your interpretation is correct. We agree that this should have been stated more directly. We have therefore added the following paragraph to explicitly contrast our models of behavioral learning against our models of metacognitive learning (p.6):

“Our models of metacognitive moral learning attribute the outcome of each action to the decision strategy that selected it (i.e., applying CBR vs. moral rules). We compared these models to models of behavioral moral learning. Unlike metacognitive learning, behavioral learning attributes the outcome of each action to the action itself. For example, a child pushing their friend out of the sandbox may see that this action causes their friend to become upset, and learn not to repeat such actions. To model the generalization of behavioral learning across the different dilemmas of our experimental paradigm, we make the simplifying assumption that people generalize from the outcome of (not) taking the action under consideration in any one dilemma to the value of (not) taking the action under consideration in all other dilemmas. Our models of behavioral learning thus assume that each decision is represented as either performing the behavior under consideration (action) or not (omission). Actions were a very salient behavior level representation on which the learning signal could operate, given that in each trial, participants were asked whether to act (e.g., push the man) or not.

Behavioral learning can be either model-based or model-free. Apart from changing the learning signal to operate on the level of behaviors rather than strategies, our

models of model-based versus model-free behavioral learning are therefore equivalent to our models of model-based versus model-free metacognitive learning. We deconfound the action/omission learning from metacognitive learning as sometimes the action coincided with CBR and sometimes with rules.”

R3.8) Similarly, I struggled a little bit to understand why the specific measure for model-based learning reflects a measure of model-based learning (and actually, the same is somewhat true for the measure of model-free learning). In regards, with the measure for model-based learning in Study 2, participants were basically confronted with a moral dilemma scenario after the learning task. The crucial difference seems to be that whereas, in the learning task, participants pick a dilemma option and receive feedback (it went great vs. it went poorly), in the measure for model-based learning, they are to estimate how likely the two options in the dilemma would lead to a good or bad outcome.

I was struck by the similarity and I struggle a little bit in understanding how such a measure is informative above and beyond the other type of measure. If participants are gradually more likely to pick a certain option, does not also follow that participants believe that option to be more likely to have a better outcome? What is the added informational value here? To me the two measures: a) the option that participants pick and b) what participants consider the option that yields a better outcome, are so similar that they might as well be the same thing.

In regards, to that same measure. Could the authors explain why exactly that measure reflects model-based learning? Allow me to play devil's advocate, let's say we implement a learning procedure that increases the inherent value associated with one of two types of actions. If I understand correctly, that would be an implementation of model-free learning. Then assume that we ask participants who have undergone such a procedure the question "Which actions yield the best outcomes", would we expect to find no difference for participants' judgments of the two types of actions? Based on that measure, on page 8, the authors conclude: "This suggests that participants learned a probabilistic model of the consequences of relying on CBR versus rules, a mechanism we will refer to as model-based metacognitive learning." Why exactly is that? The authors reference model-free learning state the model-free system selects actions "based on their average consequences in the past". Could such a mechanism not also yield this outcome?

Thank you for raising this important point. Below we first give the justification for the measures as we implemented them and then proceed to discuss the issue of rationalisation from one type of learning to the other one, which is a limitation of our method.

Our justification for the measures was based on the distinction between assigning values to actions (model-free) versus reasoning about the outcomes of the consequences (model-based). Someone relying on model-free decision making would, through learning, assign a negative outcome to a certain action and be less likely to engage in that action, independent of the anticipated consequences. This is what the task based on Cushman

et al. (2012) tried to capture: If someone is still reluctant to engage in an action in a play scenario where clearly no negative consequences are possible, this would suggest model-free decision making, as the decision seems to rely on intrinsic values associated with the action itself rather than on reasoning about the consequences. This type of reasoning about the consequences is, crucially, not a part of model-free learning. In contrast, if someone is engaged in model-based learning, they would learn about the anticipated consequences of the action rather than assigning values to actions themselves. Therefore, we would expect that these types of learners do not show an effect in the realistic play scenario, but do show an effect in the task that asks them to rate the probabilities of different consequences. This is what we found in our data, which is why we concluded evidence for model-based metacognitive learning. We added more information to the section on the measures to clarify this point (p.11).

“We developed self-report measures for model-free and model-based learning in line with previous literature on these two types of RL in the moral domain (Cushman, 2013; Cushman et al., 2012; Crockett, 2013). Because model-based learning involves learning a probabilistic model of the anticipated outcomes of actions, we used a measure that asked participants to rate the probabilities of good versus bad outcomes of choosing the CBR option, and of choosing the rule option.

In contrast, model-free learning involves assigning values intrinsically to actions rather than building a model of their possible consequences. Therefore, to measure model-free learning, we adapted a task from Cushman et al. (2012): We asked participants to imagine carrying out typically harmful actions, which would not cause negative consequences in this specific instance (e.g., shooting a prop gun). If people show an aversion to these actions even though they cannot produce negative consequences, this would suggest assigning values intrinsically to actions (i.e., model-free learning).”

However, as you point out, as far as the model-based measure is concerned, this reasoning may not apply if people rationalize probabilities from their intrinsic valuations of actions. In other words, people may have a negative evaluation of a certain action rather than a probabilistic model of the outcomes; however, when asked to reflect on the probability of the outcomes, they may derive that bad outcomes are probably more likely based on the negative evaluation of the action that they had. One piece of evidence against such rationalizations is that we see no effect on the model-free measure, and we would expect to see an effect here if people rationalize their model-based responses from their model-free responses. Nonetheless, we completely agree that in terms of the model-based measure, we cannot rule out this type of rationalization (and there may be alternative explanations for why we don't see an effect on the model-free measures). In the previous version of the manuscript, we had already added a paragraph in the Discussion stating that future research is necessary to more definitively answer the

question of whether metacognitive learning is model-based or model-free, which we now extend (newly added text in blue, p. 22).

“First, our demonstration of metacognitive moral learning raises the question of what the underlying mechanisms are. We have taken a first step towards developing and comparing models of model-free and model-based metacognitive moral learning. Our observation that metacognitive moral learning appears to be more model-based than model-free is consistent with a long series of findings suggesting that model-based learning contributes to many instances of learning that were once assumed to be purely driven by simple model-free RL (Tolman, 1948; Courville et al., 2006; Huh et al., 2009; Daw et al., 2011). However, our experiments were not optimized for this comparison, and our models also differed along another dimension. That is, the model-free model learns from continuous moral judgments, whereas the model-based model learns only about the probabilities of binary events (good vs. bad). Further, in terms of the behavioral measure of model-based learning, which used ratings of the probabilities of the different outcomes, there is a possibility of rationalization: When asked to reflect on the probability of different outcomes, purely model-free learners may derive the judgment that negative consequences are more likely from their negative model-free evaluation of the action, even though they did not learn in terms of the probabilities of the outcomes during the task (Cushman, 2020). While the fact that we did not observe an effect on the measure of model-free learning provides some evidence against this account, a definitive answer will require an experimental paradigm where model-based and model-free mechanisms produce qualitatively different behaviors (Daw et al., 2011). To address this limitation, we are developing an extension of the two-step task to moral decisions contrasting different decision strategies (Tahmasebi et al., 2024).”

R3.9) Also near the same point in the manuscript, in regards with Experiment 2, p.8 I want to make sure I understand the results correctly. The authors note:

"Compared to the CBR Success condition, participants in the Rule Success condition predicted that the rule-based action would be more likely to produce good outcomes, whereas the CBR-based action would be less likely to produce good outcomes, $F(1, 756) = 17.28, p < .001$."

If I understand correctly, the authors confronted participants with two scenarios and asked on one scenario whether a Rule-based action would lead to positive/negative outcomes and on the other scenario whether a CBR-based action would lead to positive/negative outcomes (see page 21). In the preregistration, I read: "We will calculate a model-based learning score for the CBR action and a separate score for the rule action vignette." I am a little bit confused by the suggestion that two dependents are being compared (CBR-based action across both conditions and Rule-based action across both conditions) but that we are getting only a single statistical test. Are these analyses aggregated?

Thank you for pointing out this unclear section, which also echoes a comment by Reviewer 1. This section was indeed particularly difficult to understand. During the writing process we made multiple iterations over the draft to keep the section understandable for the reader without adding an extensive amount of information about methods to the results section. We believe that ultimately, we may have ended up with the unfortunate middle ground of adding enough information to confuse the reader but enough to make the section understandable.

The basic idea is that we calculate four model-based learning scores: for the rule action in the CBR Success condition, for the CBR action in the CBR Success condition, for the rule action in the Rule Success condition, and for the CBR action in the Rule Success condition. We then test the prediction that in the CBR Success condition CBR actions receive higher scores than in the Rule Success condition, while this difference is reversed in the other condition, using a single interaction test.

We provide much more detail on the analyses and changes that we made to improve the clarity of this section in response to R1.6.

R3.10) Across the different Studies, the authors find that learning in the CBR-success condition appears to be primarily driven by model-based metacognitive learning, whereas learning in the Rules-success condition appears to be driven by model-based behavioral learning. This was a little surprising to me as I was (perhaps naively) expecting the latter to be driven by model-free metacognitive learning. I think the authors share that perspective and they do discuss it somewhat in their discussion but I think the manuscript could benefit from a deeper discussion of this issue.

Thank you for pointing this out. We think this question is best answered in terms of the answers to two sub-questions:

- 1. Would we expect metacognitive or behavioural learning?**
- 2. Would we expect model-based or model-free learning?**

With regards to the first question, we agree that it is surprising that the proportion of people relying on metacognitive learning is lower in the Rule Success condition than in the CBR Success condition; however, we also point out that 30-40% of participants still rely on metacognitive learning in the Rule Success condition. In other words, this learning style still captures a large proportion of participants best. Moreover, having deconfounded CBR/rules from action/omission, allows us to infer that most of the learning-induced change in how often the CBR option is chosen results from metacognitive learning. We can draw this conclusion because behavioral learning does not reliably shift people's tendency to engage in CBR versus rules, but only changes their preference between action versus omission. This means that – even in the Rule Success condition – most of the systematic changes in response frequencies were driven by metacognitive learning. We discuss this comment, possible explanations, and

corresponding changes that we made to the manuscript in a response to Reviewer 2 who raised a similar point (please see R2.2).

The second question concerns whether we would expect model-free or model-based metacognitive learning. We are agnostic whether the learning should be model-based or model-free; however, we would expect that this should be the same between conditions. The reason for this can be found in the distinction between behavioral and metacognitive learning. Previous research has suggested that model-free behavioural learning and reliance on rules are connected; however, this concerns which actions to select in a given situation rather than learning when to rely on rules and when to rely on CBR in general (metacognitive learning). It is not clear *a priori* whether this type of metacognitive learning would be model-based (as was suggested for behavioral learning about CBR, Cushman, 2013; Crockett, 2013) or model-free (as was suggested for behavioral learning of rules, Cushman, 2013; Crockett, 2013); however, it seems plausible that it would work the same way independent of the direction of the update. That is, learning to rely more on rules vs. learning to rely more on CBR should rely on the same learning mechanism, in the same way that learning about the accuracy of a specific rule likely has the same learning mechanism as learning about its inaccuracy. In our study, we see that model-based metacognitive learning plays an important role in both conditions (albeit somewhat weaker in the Rule Success condition than the CBR success condition), suggesting that metacognitive learning might be model-based rather than model-free. This adds to a series of findings suggesting that model-based learning contributes to many instances of learning that were once assumed to be purely driven by simple model-free RL (e.g., Tolman, 1948; Courville et al., 2006; Huh et al., 2009, Daw et al., 2011).

We now discuss the above points in the Discussion (p. 23), please see our response to your earlier comment (R3.8) for the quoted text. Further, we clarified the distinction between metacognitive and behavioural learning in different places throughout the manuscript (see R1.3 & R3.7)

R3.11) on page 9, the authors report the result of the incentive-compatible donation decision: "participants allocated £200 between a charity supporting conventional medical research and a highly effective charity promoting human challenge trials in which healthy volunteers are infected with a virus to speed up the development of vaccines. According to a preregistered one-sided t-test, participants in the CBR Success condition donated significantly more money ($M = 97.70$) to support human challenge trials than participants in the Rule Success condition ($M = 86.73$), $t(377.54) = 1.67$, $p = .047$, $d = 0.17$, one-sided. This indicates an increased reliance on CBR and a decreased reliance on moral rules because this charity breaks the moral rule "do no harm" to save many more lives in total"

Minor point but as far as I can tell this analysis does not allow one to distinguish whether the effect is driven by an increased reliance on CBR, a decreased a reliance on moral rules or a combination of the two.

Thank you for pointing this out. We agree with your comment and have removed the relevant sentence from the manuscript.

R3.12) Page 11, all the figures have typos: Log(BF) for Metacognitive Learning

Thank you for noticing this typo. We updated the axis label accordingly.

R3.13) In experiment 4, the authors test whether the learning effects transfer to a new context. Essentially, they replicate their previous paradigm, but rather than including the transfer measures in the same survey, these measures are included in a separate survey from a different set of researchers.

I personally feel that labeling this as a different context is a bit of a stretch and would recommend a more humble interpretation. Indeed, the measures are not included in the same survey, rather they are included in two separate surveys that participants take sequentially, but I am not sure this sufficient to demonstrate transfer to a new context. To me, this is similar to including a few intermediate tasks between the two parts of the study (i.e. accepting a new study, providing consent a second time, and possibly completing another survey in between). I think this study works as an argument against the claim that the authors' findings reflect task demands, but I am not sure they show much more than that. I would have found this much more convincing if the two surveys had been separated in time (say a week apart) rather than how they were administered. If I read the methods correctly, the second survey launched approximately at the time the first participants of the first survey were expected to finish. Given that, I was actually wondering how many of the participants in this study completed both studies sequentially?

Thank you for your comments. We agree with your suggestions. We have gone through the manuscript and rephrased it to describe the transfer between *experiments* rather than between *contexts*. We also emphasize the aim of the transfer, which was to rule out the alternative explanation that effects are driven by demand characteristics, rather than showing transfer across a time delay, as we agree that the evidence is not so strong when the two surveys were administered quite close together in time.

We added the following note in the Discussion as a suggestion for future research (p. 23):

“Third, follow-up research could test how stable the moral learning induced by our paradigm is over time. Experiment 4 showed that the effects of learning are not fleeting, as the transfer effects were observed after an average time delay of about two hours. However, considering that the two experiments were conducted quite close together in time, it would be necessary to implement these studies with a larger time delay to draw stronger conclusions about how long these effects last.”

Regarding your question how many participants completed both tasks sequentially, we provide that information in the Methods section and added a histogram of the time gap between the studies to the Supplementary Results of Experiment 4 (p. 33):

“A total of 1100 people took part in the first study and 811 fully completed both studies. As preregistered, of those who took part in both studies, we excluded some for failing the attention check (N=66), and those of the remaining participants who indicated that they had participated in a study by any of the experimenters before (N = 18). Figure S10 visualizes the average time difference between finishing the first part of the study and starting the second part. While some participants started the second part directly after the first (23% of participants had a time gap of less than 10 minutes), the majority of our participants had considerable gap between the two studies (62% of participants had a time gap of more than one hour and 45% had a time gap of more than two hours).”

Our ref: NATHUMBEHAV-24072785A

25th February 2025

Dear Dr. Maier,

Thank you for submitting your revised manuscript "Consequentialist Learning Shapes Reliance on Moral Rules versus Cost-Benefit Reasoning" (NATHUMBEHAV-24072785A). It has now been seen by the original referees and their comments are below. As you can see, the reviewers find that the paper has improved in revision. We will therefore be happy in principle to publish it in Nature Human Behaviour, pending minor revisions to satisfy the referees' final requests and to comply with our editorial and formatting guidelines.

We are now performing detailed checks on your paper and will send you a checklist detailing our editorial and formatting requirements within two weeks. Please do not upload the final materials and make any revisions until you receive this additional information from us.

Sincerely,

[REDACTED]

[REDACTED]

[REDACTED]

Nature Human Behaviour

Dear Dr. Horder,

Thank you very much for the conditional acceptance. We are grateful for your and the reviewers' positive comments and constructive feedback for improvement. We have addressed the minor suggestions of Reviewer 3 by making several adjustments to the Discussion section. In particular, we now acknowledge that the benefits of moral learning from consequences are contingent on how people evaluate those consequences.

Regarding the queries raised by the editorial assistants and author checklist, we would be happy to have the reviewer comments, author rebuttal letters, and editorial decision letters published as a Supplementary item. Regarding advertisement on social media, the relevant handles of the authors are as follows: X/Twitter: @MaxMa1er, @vanessachg_, @FalkLieder, @RtnlAltruismLab; Bluesky: @maxmaier.bsky.social, @vanessachg.bsky.social

See below for a point-by-point reply to the remaining comments. We hope that the manuscript now meets your expectations for final acceptance.

Sincerely,
Maximilian Maier, Vanessa Cheung, and Falk Lieder

Point-by-Point Response

Reviewer 1

Comment 1.1: *Generally speaking, the authors have responded in depth to each of my comments and misunderstandings. The addition of the discussion of what the results tell us is a real plus - and the authors present these contributions simply, which makes it possible to really appreciate the contribution. Although it wasn't my comment (but the reviewer 2's), the addition of a diagram explaining the paradigm definitely helps in understanding the protocol. At this point, I have no further comments to make, and am particularly in favor of seeing the article published as it stands.*

Finally, I would like to offer the authors my sincere apologies for the delay in sending my report. I hope my very favorable opinion will be a consolation prize.

Again, I wish the authors good luck with their excellent research.

Response: Thank you very much for the feedback and for your positive comments.

Reviewer 2

Comment 2.1: I am happy with the changes made in response to my comments.

Response: Thank you very much for the feedback.

Reviewer 3

Comment 3.1: *I read through the revised manuscript and the authors (very thorough) reply letter and can only say that I was pleased to see how the authors approached this revision. As far as I can tell they responded to all comments with a clear and reasonable response and in doing so have clarified the issues I flagged in my review. I think the authors have written an interesting paper and I would be happy to see it in print.*

I had one minor concern regarding the following passage. On page 23 in their conclusion the authors write:

"It has often been argued that human morality is fallible and that people are often swayed by morally irrelevant details [11]. While this may be true of people's decisions in traditional

philosophy thought experiments, our experiments offer a more optimistic perspective: When people experience the outcomes of their moral decisions, they learn to adopt decision strategies that benefit the greater good."

I do love happy endings but I was wondering whether this conclusion is fully appropriate. Just to push back a little: as far as I can tell, what the paper demonstrates is that people learn from the consequences of their decisions in such a way that they end up pursuing those decision-strategies that yield a higher proportion of the outcomes that they consider good. Within the context of these experiments, "the outcomes considered good" align with "the greater good" but that is because the dilemmas were set up in such a way. It is not something that I think would hold true in general as I can imagine many types of dilemmas where the outcomes that people consider "good" might not necessarily align with "the greater good". For instance, if the dilemmas used would involve harm to one's family vs. greater harm to strangers, I could imagine that many people will focus on whatever decision-strategy yields minimal harm to one's family even if that is to the detriment of the greater good. Accordingly, the conclusion "When people experience the outcomes of their moral decisions, they learn to adopt strategies that benefit the greater good" seems overly broad to me.

Response: Thank you for raising this point. You are right that the benefits of moral learning from consequences are contingent on how people evaluate those consequences. We have rewritten the corresponding paragraphs of the Discussion section to acknowledge this contingency (changes highlighted in blue text):

"It has often been argued that human morality is fallible and that people are often swayed by morally irrelevant details [11]. While this may be true of people's decisions in traditional philosophy thought experiments, our experiments offer a more optimistic perspective: When people experience the outcomes of their moral decisions, they **can learn to adopt decision strategies that are more likely to yield outcomes they consider to be morally good. Moreover, when people's moral judgments of the consequences are sufficiently impartial, as they were in our experiments, the lessons they learn from those consequences can benefit the greater good.** Thus, with sufficient experience, people's morality can, **in principle, become more** adaptive.

[...]

our research **suggests** that in situations where people receive frequent, prompt, and accurate feedback about the consequences of past decisions, their moral decision-making might **be less inconsistent** than their responses to thought experiments suggest." (p. 24)

Comment 3.2: *Additionally on On page 20, the authors write: "Second, a participant's "deontological" or "utilitarian" choices in moral dilemmas do not necessarily demonstrate that the participant is deontologist or utilitarian. Instead, those choices should be interpreted more cautiously as being consistent with following moral rules or CBR."*

A relevant reference here might be: Conway, P., Goldstein-Greenwood, J., Polacek, D., & Greene, J. D. (2018). Sacrificial utilitarian judgments do reflect concern for the greater good: Clarification via process dissociation and the judgments of philosophers. Cognition, 179,

241-265. *I actually think the authors' perspective aligns with the view of several prominent researchers in the field. Judgments can be said to align with utilitarians/deontology, however labelling people as "utilitarians" "deontologists" is a stretch.*

Response: Thank you for this suggestion. We have added the reference to this sentence (changes highlighted in blue text):

"Second, a participant's "deontological" or "utilitarian" choices in moral dilemmas do not necessarily demonstrate that the participant is deontologist or utilitarian (see also Conway et al., 2018). Instead, those choices should be interpreted more cautiously as being consistent with following moral rules or CBR."

Comment 3.3: Beyond this, I had one reflection. This don't really warrant any change to the manuscript. It doesn't even warrant a reply from the authors. It is just a reflection I wanted to share with the authors.

On page 19 they write:

"Across all experiments, the overwhelming majority of participants showed some form of learning from consequences (at least 95% show metacognitive or behavioral learning, see Table 2), suggesting that at most 5% of people are strictly deontological, in the sense that they would continue to base their decisions on moral rules even if the consequences of previously doing so had been predominantly bad."

I actually find the 5% number to be quite a high given what it represents (people that fully ignore outcomes when making moral decisions). It does make me wonder what the number on the opposite side would be: the percentage of people that are "pure utilitarians" and only weigh outcomes without weighing the value of actions themselves.

Response: Thank you for sharing this interesting perspective! We agree that this doesn't warrant any change to the manuscript, but is nonetheless worth thinking about.